# Regret-Optimal Transfer Learning from Pretrained Kernel Regressors – with Applications to American Option Pricing

## Abstract

Several pre-trained machine learning models operate by aggregating, fine-tuning, and adapting a (set of) models which have been pre-trained on a set of relevant tasks to any given novel task. This paper exhibits a theoretically regret-optimal iterative scheme for performing this aggregation for pre-trained finite-rank kernel ridge regression models using a fixed and finite number of training iterations. We investigate the properties of our algorithm, including its computational complexity and its adversarial robustness. We show that an adversary which perturbs $q$ training pairs by at most $\varepsilon > 0$, across all training sets, cannot reduce the regret-optimal algorithm's regret by more than $\mathcal{O}(\varepsilon q \bar{N}^{1/2})$, where $\bar{N}$ is the aggregate number of training pairs. Additionally, by leveraging symmetries within the regret-optimal algorithm, we further develop a nearly regret-optimal heuristic that runs with $\mathcal{O}(Np^2)$ fewer elementary operations, where $p$ is the dimension of the parameter space and $N$ is the total number of training instances. To validate our theoretical findings, we conduct numerical experiments in the context of American option pricing, utilizing a randomly generated finite-rank kernel.

## 1 Introduction

A core challenge in machine learning is identifying a set of parameters that optimize the performance of a learning model on a single dataset $\mathcal{D}_1$ by empirical risk minimization. Often, especially in transfer learning, one does not have access to only a single dataset but *several relevant datasets* $\mathcal{D}_1, \ldots, \mathcal{D}_N$ whose data can be integrated into the training scheme. Comparable datasets naturally arise in instances of meta-learning (Finn et al., 2017; Pavasovic et al., 2022)), in multi-task learning (Kumar & Daumé III, 2012; Yu et al., 2020), in quantitative finance due to natural market cycles (Lowry & Schwert, 2002), or in robust finance approaches to model uncertainty resulting in several plausible stochastic time-series models wherewith data can be simulated (Hou & Obłój, 2018; Aksamit et al., 2019; Prashanth & Bhat, 2022).

This paper considers the problem of optimally selecting a deterministic training algorithm which maximizes the performance of any given finite-rank kernel ridge regressor on a primary dataset $\mathcal{D}_1$ while incorporating the information present in $N$ *pretrained* kernel ridge regressors, each trained on one of the datasets $\mathcal{D}_1, \ldots, \mathcal{D}_N$. The optimality of a training algorithm is quantified by minimizing a performance metric, the trajectory generated by the optimization algorithm after a fixed number ($T$) of iterations.

We focus on the rich class of finite-rank kernel ridge regressor (fKRR) since they subsume all deep neural networks whose hidden layers have been pre-trained (or randomly generated) and frozen, and only the final linear layer can be trained on the relevant novel task; see e.g. Herrera et al. (2023); Gonon (2023). Furthermore, fKRRs are also known to provide analytically tractable approximations to the training dynamics of deep neural networks under certain conditions Jacot et al. (2018; 2020); Arora et al. (2019); Bordelon et al. (2021).

### 1.1 Problem Formulation

Fix positive integers $d$, $N$, $p$, and $T$. A finite-rank kernel regressor is a map $f_\theta : \mathbb{R}^d \to \mathbb{R}$ which is linear in its trainable parameter[1] $\theta \in \mathbb{R}^p$ but typically *non-linear* in its inputs $x \in \mathbb{R}^d$ through a feature map $\phi : \mathbb{R}^d \to \mathbb{R}^p$

$$f_\theta(x) = \phi(x)^\top \theta; \tag{1}$$

typically with $p \gg d$. The kernel associated to $f$ is given by $\kappa(x, \tilde{x}) = \phi(x)^\top \phi(x)$. One example would be a pre-trained deep neural network where only the final linear layer $\theta$ is fine-tuned and the frozen hidden layers define the feature map $\phi$; see Section 3 for details.

We are provided with $N$ non-empty datasets $\mathcal{D}_1, \ldots, \mathcal{D}_N \subseteq \mathbb{R}^d \times \mathbb{R}$, of which $\mathcal{D}_1$ is distinguished as describing the principal (supervised) regression task. Each of the $i = 1, \ldots, N$ pre-trained fKRRs $f_{\theta^n}$ optimizes their respective penalized empirical minimization problem defined over their respective dataset $\mathcal{D}_i$

$$\inf_{\theta \in \Theta} \sum_{(x,y) \in \mathcal{D}_i} (f_\theta(x) - y)^2 + \kappa \|\theta\|_2; \tag{2}$$

where the hyperparameter $\kappa > 0$ controls the regularity of the model by modulating the magnitude of Euclidean norm $\|\theta\|_2$ of its weight vector. The minimizer $\theta_i^\star$ of (2) for each $i$ is called a *finite-rank kernel ridge regressor* (fKRR) and is available in closed-form, see Appendix C.

Fix a set of weights $w \in \Delta_N \overset{\text{def.}}{=} \{[0,1]^N : \sum_{i=1}^N w_i = 1\}$, to be optimized. We extend the summed squared error (SSE) loss function in (2) in the case $n = 1$ to incorporate the influence of other datasets into the primary regression task

$$l(\Theta, w; \mathcal{D}) \overset{\text{def.}}{=} \sum_{i=1}^N w_i \sum_{j=1}^{|\mathcal{D}_i|} (f_{\theta^w}(x_j^i) - y_j^i)^2, \tag{3}$$

where $\Theta \overset{\text{def.}}{=} (\theta_1^\top, \cdots, \theta_N^\top)^\top \in \mathbb{R}^{Np \times 1}$ and $\theta^w \overset{\text{def.}}{=} \sum_{i=1}^N w_i \theta_i$. The optimal weights $w$ are selected in a PAC-Bayesian fashion, see Alquier (2021) for an overview of PAC-Bayesian bounds, in such a way that they prioritize the influence of datasets which are similar to $\mathcal{D}_1$ while automatically removing sufficiently different datasets.

Since the optimizer of (3) need not be available in closed-form, we must optimize it iteratively. Upon fixing a maximal number of iterations $T$ and an initial parameter choice $\theta_0 \in \mathbb{R}^p$, we follow Casgrain & Kratsios (2021) and identify every deterministic training algorithm for (3) with a sequence in $\mathbb{R}^{Np}$ of length $1 + T$

$$\theta^{0 \cdots T} \overset{\text{def.}}{=} \{\theta(t)\}_{t=0}^T, \qquad \theta(0) = \theta_0. \tag{4}$$

**Regret Optimality**   We seek the algorithms which aim to minimize (3) while biasing their iterates towards the parameters $\theta_1^\star, \ldots, \theta_N^\star$ which are optimal for the pre-trained models, weighted by $w$. To this end, minimize the following *penalized regret functional* over the class of deterministic algorithms, i.e. of the form (4),

$$\mathcal{R}(\Theta^{0 \cdots T}) \overset{\text{def.}}{=} \underbrace{l(\Theta(T), w; \mathcal{D})}_{\text{Prediction}} - l^\star + \sum_{t=0}^{T-1} \big( \underbrace{\lambda \|\Theta(t+1) - \Theta^\star\|_2^2}_{\text{Transfer Learning}} + \underbrace{\beta \|\Delta\Theta(t)\|_2^2}_{\text{Algo. Stability}} \big), \tag{5}$$

where $\Theta^\star \overset{\text{def.}}{=} (\theta_1^{\star\top}, \cdots, \theta_N^{\star\top})^\top \in \mathbb{R}^{Np \times 1}$ records the optimal parameters of each pretrained fKRR, $\Delta\Theta(t) \overset{\text{def.}}{=} \Theta(t+1) - \Theta(t)$ quantifies the magnitude of each training update, and $l^\star \overset{\text{def.}}{=} \min_{\Xi \in \mathbb{R}^{Np \times 1}} l(\Xi, w; \mathcal{D})$ is the minimal value of the loss function (3). Thus, the formulation of regret in (5) is *specific to the setting of training a fKRR from multiple datasets*. The first term *(prediction)* in (5) emphasizes that any "good" optimization algorithm must minimize the regression loss function (3). The second term *(transfer learning)* encodes the degree to which an optimization algorithm encodes information from similar datasets into its iterates. Finally,

---

[1]We emphasize that all vectors are represented as column vectors, not row vectors.

the third (algo. stability) term biases towards algorithms whose iterative updates do not change too rapidly, similar to controlling the learning rate in gradient descent. A game theoretic interpretation of $\mathcal{R}$ is given in Appendix A.

The primary objective of this paper is to exhibit a *regret optimal optimization algorithm*, meaning a sequence $\Theta^{0\ldots T}$ minimizing $\mathcal{R}$ across all deterministic algorithms with $T$ iterations; i.e. all elements of $\mathbb{R}^{(1+T)Np}$. Further, our objective is to explicitly establish the existence of a regret-optimal algorithm, i.e. the algorithm minimizing (5), and to investigate its theoretical properties such as adversarial robustness and computational complexity.

## 1.2 Contributions

We exhibit a *regret optimal* algorithm, by which we mean an optimizer of (5). Aligned with the recent trends in optimization Casgrain (2019b); Li et al. (2018), we leverage techniques from optimal control theory to characterize the regret-optimal algorithm. We show that this is possible due to the observation that minimizers of (5) coincide with the minimizers of the energy functional

$$\mathcal{L}(\Theta^{0\ldots T}) \stackrel{\text{def.}}{=} \underbrace{\sum_{t=0}^{T-1}\left[\lambda\left\|\Theta(t+1)-\Theta^{\star}\right\|_2^2 + \beta\left\|\Delta\Theta(t)\right\|_2^2\right]}_{\text{Running-Cost}} + \underbrace{l(\Theta(T),w;\mathcal{D})}_{\text{Terminal Cost}}. \tag{6}$$

From this control-theoretic perspective, every algorithm (4) can be regarded as driven by a "control", and the functional $\mathcal{L}$ has the form of a "cost function" of an optimal control problem, decomposable into the sum of a "running/operational cost" and a "terminal/objective cost". This allows us to leverage tools from optimal control (Touzi & Tourin, 2013) to derive a closed-form expression of the unique regret-optimal algorithm driven by the unique "optimal control" as defined in Algorithm 2.

We prove that this procedure is adversarially robust, meaning that the value of (6) varies by $\mathcal{O}(\varepsilon\sqrt{\bar{N}}q)$ if a malicious adversary can perturb at most $1/q$ percent of all the data in each of the dataset $\mathcal{D}_i$ by at-most $\varepsilon \geq 0$, where $\bar{N} = \sum_{i=1}^{N}|\mathcal{D}_i|$. Furthermore, we show that our regret-optimal algorithm has a computational complexity of $\mathcal{O}(N^2p^3 + T(Np)^{2.373})$. Moreover, we characterize the optimal choice of the weights $w$ which the central planner must use to maximize the selected model's performance on the primary/focal dataset $\mathcal{D}_1$.

Finally, we show that if all the datasets $\mathcal{D}_1, \ldots, \mathcal{D}_N$ are equally dissimilar (but not necessarily similar) then the regret-optimal algorithm can be accelerated. This acceleration requires $\mathcal{O}(Np^2)$ fewer elementary operations. We also show that if this "accelerated" algorithm is used in situations when the datasets differ by a little, then it does not deviate far from the regret-optimal algorithm. Thus, it is an accelerated heuristic.

We emphasize that this problem has not previously been studied. Therefore, there are no available benchmarks for the proposed algorithm. Hence, our numerical experiments consist of ablation studies where the proposed models are benchmarked against various natural alternatives to the regret-optimal algorithm.

### Outline

The paper is organized as follows. In Section 2, we introduce the regret-optimal algorithm and demonstrate the optimality, computational complexity, and adversarial robustness of the algorithm. We further introduce an accelerated algorithm of lower computational complexity that can achieve near-regret optimality. In Section 3, we conduct a convergence analysis of the regret-optimal algorithm and compare it to the standard gradient descent approach. Then we show the transfer learning capabilities of the algorithm for American option pricing.

A game theoretic interpretation of $\mathcal{R}$ is given in Appendix A. Appendix B is devoted to proving the optimality, adversarial robustness, and computational complexity of the regret-optimal algorithm. The near-regret optimality and computational complexity of the accelerated algorithm are also proven. Appendix C briefly introduces the background of finite-rank kernel ridge regressors.

### 1.3 Related Work

Examples of $f_\theta$ of the form (1) include extreme learning machine (Gonon et al., 2020; 2023; Ghorbani et al., 2021; Mei & Montanari, 2022) or any finite-rank kernel ridge regressor (fKRR) (Amini, 2021; Amini et al., 2022). Random feature models, and finite-rank ridge regressors, are typical in contemporary quantitative finance (Gonon, 2023; Herrera et al., 2023) or reservoir computing (Cuchiero et al., 2021), where it is favourable to rapidly deploy a deep learning model due to time or computing-power limitations/constraints. Finite-rank kernel ridge regressors are also typical in the setting of standard transfer (Howard & Ruder, 2018) and multi-task learning (Ren & Lee, 2018; Jia et al., 2019) pipelines where one often has a pre-trained deep neural network model and, for any novel task, the user freezes the hidden layers in the deep neural network model and only fine-tunes/trains the deep neural network's final linear layer. This is useful, since the hidden layers of most deep neural network architectures are known to process general, cruder features and the final layer of any such model captures finer, task-specific features (Zeiler & Fergus, 2014; Yosinski et al., 2014).

The optimization of a model trained from multiple data sources lies at the heart of multi-task learning. In Sener & Koltun (2018), the authors cast the task of optimizing (3) as a multi-objective optimization problem. They propose to solve it via a multi-gradient descent algorithm which, under mild conditions, converges to a Pareto stationary point; i.e. which is essentially a critical point of each of the $i$-th player's penalized SSE. Though iterative schemes for multi-objective optimization can efficiently reach critical points (Fliege et al., 2019), this problem's critical point is not necessarily a minimum since the problem (3) is not convex when $w$ is shared amongst all $N$ players.

This proposed algorithm reflects the ideas of the federated stochastic gradient descent (FedSGD) algorithm of McMahan et al. (2017) and of its online counterpart FedOGD which is an online federated version of online gradient descent[2] and more efficient online versions thereof such as Kwon et al. (2022). From the federated learning perspective, the problem of training individual fKRR on each dataset, and then centralizing them by optimizing the mixing parameter $w$ in (3) and its origins date back to local gradient descent introduced in the optimization and control literature in Mangasarian (1995). Though several authors (e.g. Khaled et al. (2020)) have proposed various progressive improvements of federated gradient-descent-type algorithms for training a learner from several data sources, the problem of studying *the* most efficient iterative federated learning algorithm for regret minimization has not yet been tackled.

We note that (stochastic) optimal control tools have revealed new insights into machine learning. Examples include an optimal step-size control (Li et al., 2017b) and batch-size control (Zhao et al., 2022), developing online subgradient methods that adapt to the data's observed geometry (Duchi et al., 2011), developing new maximum principle-based training algorithms for deep neural networks (Li et al., 2018), or unifying (stochastic) optimization frameworks (Casgrain, 2019a).

### Notation

We briefly introduce some additional notation. Each dataset has a finite number of samples which we write as $\mathcal{D}_i = \{(x_j^i, y_j^i) \mid 1 \le j \le |\mathcal{D}_i|\}$. We assume that some feature map $\phi$ is fixed and denote the features of each sample of $\mathcal{D}_i$ as

$$u_j^i \stackrel{\text{def.}}{=} \phi(x_j^i).$$

Moreover, we denote the joint inputs and outputs for each dataset as

$$X^i \stackrel{\text{def.}}{=} \left(x_1^i, \cdots, x_{|\mathcal{D}_i|}^i\right)^\top \in \mathbb{R}^{|\mathcal{D}_i| \times d} \quad \text{and} \quad Y^i \stackrel{\text{def.}}{=} (y_1^i, \cdots, y_{|\mathcal{D}_i|}^i)^\top \in \mathbb{R}^{|\mathcal{D}_i|},$$

and the joint features as

$$U^i \stackrel{\text{def.}}{=} \Phi(X^i) \stackrel{\text{def.}}{=} \left(\phi(x_1^i), \cdots, \phi(x_{|\mathcal{D}_i|}^i)\right)^\top \in \mathbb{R}^{|\mathcal{D}_i| \times p}.$$

Finally, we use $I_p$ to denote a $p \times p$ identity matrix.

---

[2]See (Hazan et al., 2016, Chapter 3) for details on online gradient descent

## 2 Main Results

This section contains the description of the *unique* regret-optimal algorithm and an analysis of its theoretical properties. Throughout our manuscript, we maintain the following two assumptions.

**Assumption 1** ($\mathcal{D}$-Boundedness). *There are constants $K_x, K_y > 0$ such that all $(x_j^i, y_j^i) \in \mathcal{D}$ satisfy $\|u_j^i\|_2 \leq K_x$ and $|y_j^i|^2 \leq K_y$, where $u_j^i = \phi(x_j^i)$.*

**Assumption 2** (($w^\star, \mathcal{D}$)-Compatibility). *The data $\mathcal{D} = \cup_{i=1}^N \mathcal{D}_i$ and the weights $w^\star$ satisfy*

$$\sum_{i=1}^N \sum_{j=1}^{|\mathcal{D}_i|} u_j^i u_j^{i\top} \geq 0, \tag{7}$$

$$[w_1^\star I_p, \cdots, w_N^\star I_p]^\top \left( \sum_{i=1}^N w_i^\star \sum_{j=1}^{|\mathcal{D}_i|} u_j^i u_j^{i\top} \right) [w_1^\star I_p, \cdots, w_N^\star I_p] \geq 0. \tag{8}$$

With the notation introduced before, the object $[w_1^\star I_p, \cdots, w_N^\star I_p]$ above is a $p \times Np$ matrix.

### 2.1 The Algorithms

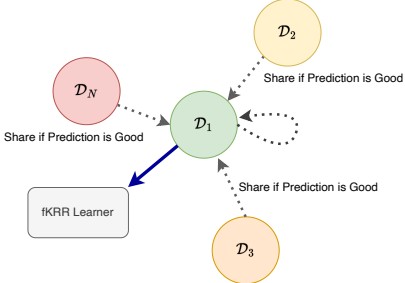

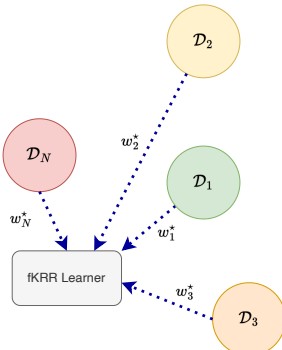

(a) Step 1: Adaptive "Prior" Sharing: The $i^{th}$ dataset $\mathcal{D}_i$ is shared with the main dataset $\mathcal{D}_1$ if the best model $f_{\theta_i^\star}$, optimizing (25) on dataset $\mathcal{D}_i$, can outperform the optimal model trained on the main dataset, up to a slack factor $\eta > 0$. The relative importance of each dataset is proportional to $f_{\theta_i^\star}$'s performance relative to the training performance of the best model trained on the main dataset. When $\eta$ is small, sharing is unlikely; when $\eta$ is large, nearly all datasets are pooled together with approximately the same relative importance.

(b) Step 2: Optimized "Posterior" Sharing: We then optimize the relative importance/weights given to the shared datasets from the naive adaptive sharing. The optimization is a closed-form update of the adaptive "prior" weights which optimize a certain maximum expected performance criterion subject to a relative entropy penalization against the weights obtained from adaptive prior sharing.

Figure 1: Visual Summary of Optimal Information Sharing (Algorithm 1).

We begin by describing our procedure for determining the optimal mixture weights defining the cooperative objective function (3) used in (5). This procedure is illustrated graphically in Figure 1 and detailed in Algorithm 1.

Suppose now that we have used Algorithm 1 to compute weights $w^\star = (w_1^\star, \cdots, w_N^\star)$ and the locally optimal fKRR parameters $\theta_1^\star, ..., \theta_N^\star$, and $w^\star$. The key novelty in the next algorithm follows a similar spirit to Li et al. (2017a) and Casgrain & Kratsios (2021), by casting the problem of constructing a regret-optimal algorithm as an optimal control problem.

As before, denote

$$\Theta(t) \stackrel{\text{def.}}{=} [\theta_1^\top(t), \cdots, \theta_N^\top(t)]^\top, \quad \Theta^\star \stackrel{\text{def.}}{=} [\theta_1^{\star\top}, \cdots, \theta_N^{\star\top}]^\top,$$

---

**Algorithm 1:** Optimal Information Sharing - Initialization to Algorithm 2

---

**Require:** Datasets $\mathcal{D}_1, \ldots, \mathcal{D}_N$, finite-rank kernel $\phi$, hyperparameters[3] $\kappa, \eta > 0$.

    `// Initialize Locally-Optimal Learners and Record Scores on Main Dataset`

    **For** $i : 1, \ldots, N$ **in parallel**

    |  $\theta_i^\star \leftarrow (\Phi^\top(X^i)\Phi(X^i) + \kappa I_p)^{-1}\Phi^\top(X^i)Y^i$                        `// Optimize each fKRR`

    |  $s^i \leftarrow \frac{1}{|\mathcal{D}_1|}\sum_{j=1}^{|\mathcal{D}_1|} \left\| f_{\theta_i^\star}(u_j^1) - y_j^1 \right\|_2^2$            `// Score predictive power of fKRR on main dataset`

        **end**

    `// Get Adaptive Prior`

    **For** $i : 1, \ldots, N$ **in parallel**

    |  $w_i \leftarrow \frac{[s^1 + \eta - s^i]_+}{\sum_{j=1}^N [s^1 + \eta - s^j]_+}$

        **end**

    `// Get Optimized Posterior`

    **For** $i : 1, \ldots, N$ **in parallel**

    |  $w_i^\star \leftarrow \frac{e^{s_i/\eta}[s^1 + \eta - s^i]_+}{\sum_{j=1}^N e^{s_j/\eta}[s^1 + \eta - s^j]_+}$

        **end**

    **return** Optimal Weights $w^\star$ and Locally-Optimal fKRR Parameters $\theta_1^\star, \ldots, \theta_N^\star$.

---

$$\boldsymbol{\alpha}(t) \overset{\text{def.}}{=} [\alpha_1^\top(t), \cdots, \alpha_N^\top(t)]^\top,$$

which are all elements of $\mathbb{R}^{Np}$, and introduce the dynamics that $\Theta$ follows

$$\Theta(0) = \Theta^\star, \quad \Theta(t+1) = \Theta(t) + \boldsymbol{\alpha}(t), \quad 0 \le t \le T - 1, \tag{9}$$

which is equivalently written as

$$\Theta(t+1) = \Theta^\star + \sum_{u=0}^t \boldsymbol{\alpha}(u), \quad 0 \le t \le T - 1. \tag{10}$$

Then, since the optimizers of the energy (6) coincide with the optimizers of the systemic regret functional (5), searching for a regret-optimal algorithm reduces to solving for an optimal control $\alpha$ to minimize the cost (6) subject to (9). The optimal control problem (6) and (9) is a discrete-time linear quadratic optimal control problem, and can be solved by dynamic programming methods that are standard in the literature. For instance, by (Başar & Olsder, 1998, Proposition 5.1), we obtain the solution for the regret-optimal algorithm in closed-form as solutions of a system of Riccati equations (15) and (16), given in Theorem 2.

Algorithm 2 below implements the optimal control approach in two steps. In Step 1, we solve the equation system (15) and (16) backwards for $t = T, T - 1, ..., 1$. Then in Step 2, we obtain the optimal control $\boldsymbol{\alpha}(t)$ and update the parameter $\Theta(t)$ for $t = 0, 1, ... T$.

In the next subsections, we state our primary results, whose proofs are presented in Appendix B. We first quantify the optimality of Algorithm 1. Next, we establish the optimality, complexity, and adversarial robustness of Algorithm 2.

## 2.2 Properties of Algorithm 1

By decoupling the optimization of the "dataset-relevant weights" $w$ in Algorithm 1 from the optimization of the parameter $\theta$ in Algorithm 2, we avoid open-loop forward-backward iterations, which iterate between optimizing $w$ and $\theta$. We now study the optimality of the weight $w^\star$ generated by Algorithm 1. The weights $w^\star$ represent the optimal probability distributions of randomly selecting amongst the $N$ optimized regressors $\{f_{\theta_i^\star}\}_{i=1}^N$, each optimized locally on their dataset $\mathcal{D}_i$ by solving (25). Here, optimality is quantified in terms of the best-expected prediction on the focal dataset $\mathcal{D}_1$ subject to a penalty for deviating too far from a

---

**Algorithm 2:** Regret-Optimal Optimization Algorithm

---

**Require:** Datasets $\mathcal{D}_1, \ldots, \mathcal{D}_N$, $N$. Iterations $T \in \mathbb{N}_+$, finite-rank kernel $\phi$ and hyperparameters $\lambda, \beta, \kappa, \eta > 0$.

// Get Initialize Weights and Locally-Optimal fKRR Parameters

$\theta_1^\star, \ldots, \theta_N^\star, w^\star \leftarrow$ Run: Algorithm 1 with $\mathcal{D}_1, \ldots, \mathcal{D}_N$, $\phi$, and $\kappa, \eta$.

// Initialize Updates

$P(T) = [w_1^\star I_p, \cdots, w_N^\star I_p]^\top \left( \sum_{i=1}^N w_i^\star \sum_{j=1}^{|\mathcal{D}_i|} u_j^i u_j^{i\top} \right) [w_1^\star I_p, \cdots, w_N^\star I_p]$

$S(T) = -[w_1^\star I_p, \cdots, w_N^\star I_p]^\top \sum_{i=1}^N w_i^\star \sum_{j=1}^{|\mathcal{D}_i|} u_j^i y_j^i$

$\Theta(0) = \Theta^\star \overset{\text{def.}}{=} (\theta_1^{\star\top}, \cdots, \theta_N^{\star\top})^\top$

// Generate Iterates

**for** $t = T - 1, \ldots, 1$ **do**

  // Update Driving Parameters

  $P(t) = \beta I_{Np} - \beta^2 [(\lambda + \beta) I_{Np} + P(t+1)]^{-1}$

  $S(t) = \beta [(\lambda + \beta) I_{Np} + P(t+1)]^{-1} (S(t+1) - \lambda \Theta^\star)$

  **end for**

**for** $t = 0, \ldots, T - 1$ **do**

  // Update Control

  $\boldsymbol{\alpha}(t) = -[(\lambda + \beta) I_{Np} + P(t+1)]^{-1} [(\lambda I_{Np} + P(t+1))\Theta(t) - \lambda \Theta^\star + S(t+1)]$

  $\Theta(t+1) = \Theta(t) + \boldsymbol{\alpha}(t)$

  **end for**

**return** Return the Optimized fKRR $f_{\theta^{w^\star}}$

---

reference distribution. This reference distribution serves to automatically delete models whose datasets are too dissimilar to $\mathcal{D}_1$. Inspired by most meta-learning problems, e.g. Pavasovic et al. (2022), similarity is quantified by extracting a "similarity score" from each learner.

Let $\mathcal{P}(\{\theta_i^\star\}_{i=1}^N)$ consist of all probability measures on the set $\{\theta_i^\star\}_{i=1}^N$. Any probability measure $\mathbb{P}_w$ in $\mathcal{P}(\{\theta_i^\star\}_{i=1}^N)$ is uniquely determined by a weight $w$ in the $N$-simplex; consisting of all $w \in [0,1]^N$ whose entries sum to 1. Let $\theta$ be a random variable with law $\mathbb{P}_w$, for some $\mathbb{P}_w \in \mathcal{P}(\{\theta_i^\star\}_{i=1}^N)$. The average performance of the random learner $f_\theta$ on the dataset $\mathcal{D}_1$ is

$$\mathbb{E}_{\theta \sim \mathbb{P}_w} \left[ \frac{1}{|\mathcal{D}_1|} \sum_{j=1}^{|\mathcal{D}_1|} \left\| f_\theta(u_j^1) - y_j^1 \right\|_2^2 \right] = \sum_{i=1}^N w_i \underbrace{\left[ \frac{1}{|\mathcal{D}_1|} \sum_{j=1}^{|\mathcal{D}_1|} \left\| f_{\theta_i^\star}(u_j^1) - y_j^1 \right\|_2^2 \right]}_{i^{th} \text{ learner's score } (s_i)}. \tag{11}$$

Fix an *information sharing level* $\eta > 0$. The information sharing level $\eta$ adaptively determines which of the datasets in $\mathcal{D}$ are implicitly shared with the principle dataset $\mathcal{D}_1$. The information sharing is formalized through the probability measure

$$\bar{\mathbb{P}}^\eta(\theta_i^\star) \overset{\text{def.}}{=} \frac{[s^1 + \eta - s^i]_+}{\sum_{i=1}^N [s^1 + \eta - s^i]_+}.$$

on the set of $N$ learners, or equivalently on the parameters $\theta_1^\star, \ldots, \theta_N^\star$. The measure $\bar{\mathbb{P}}^\eta$ thus allows us to minimize the regularized version of the loss in (11),

$$\min_{\mathbb{P}_w} \underbrace{\mathbb{E}_{\theta \sim \mathbb{P}_w} \left[ \frac{1}{|\mathcal{D}_1|} \sum_{j=1}^{|\mathcal{D}_1|} \left\| f_\theta(u_j^1) - y_j^1 \right\|_2^2 \right]}_{i^{th} \text{ learner's score } (s_i)} + \underbrace{\eta \, D_{\mathrm{KL}} \left( \mathbb{P}_w \| \bar{\mathbb{P}}^\eta \right)}_{\substack{\text{Inhomogeneity of} \\ \text{information sharing}}}, \tag{12}$$

where the infimum is taken over all $\mathbb{P}_w$ in $\mathcal{P}(\{\theta_i^\star\}_{i=1}^N)$ and $\mathrm{D}_{\mathrm{KL}}$ denotes the *Kulbeck-Leibler divergence with respect to* $\bar{P}^\eta$ given for any probability measure $\mathbb{P}_w$ on $\{\theta_i^\star\}_{i=1}^N$, namely

$$\mathrm{D}_{\mathrm{KL}}\left(\mathbb{P}_w\|\bar{\mathbb{P}}^\eta\right) \stackrel{\text{def.}}{=} \sum_{i=1}^N w_i \log\left(\frac{w_i}{[s^1+\eta-s^i]_+}C_\eta\right),$$

where $C_\eta \stackrel{\text{def.}}{=} \sum_{j=1}^N [s^1 + \eta - s^j]_+$, if $\mathbb{P}_w$ is absolutely continuous with respect to $\bar{\mathbb{P}}^\eta$ and otherwise $\mathrm{D}_{\mathrm{KL}}\left(\mathbb{P}_w\|\bar{\mathbb{P}}^\eta\right) = \infty$. There is a unique probability measure optimizing the penalized functional (12) determined by the weight $w^\star$ in Algorithm 1.

**Theorem 1** (Optimality of Weights $w^\star$ in Algorithm 1)**.** *Fix $\eta > 0$. The unique weights $w_i^\star$ on the $N$-simplex minimizing* (12) *are given by*

$$w_i^\star = \frac{e^{s^i/\eta}[s^1+\eta-s^i]_+}{\sum_{j=1}^N e^{s^j/\eta}[s^1+\eta-s^j]_+}, \qquad i = 1, \ldots, N. \tag{13}$$

### 2.3 Guarantees for Algorithm 2

Our next result shows that Algorithm 2, and its iterates, are regret-optimal.

**Theorem 2** (Algorithm 2 Computes the Unique Regret-Optimal Algorithm)**.** *Suppose that Assumptions 1 and 2 hold and fix $\Theta^\star \in \mathbb{R}^{Np}$ as in* (28)*. Then, the systemic regret functional $\mathcal{R}$ defined in* (5) *admits a unique minimizer $\hat{\Theta} \stackrel{\text{def.}}{=} (\hat{\Theta}(t))_{t=0}^T$ of the form* (29) *with $\hat{\Theta}(0) = \Theta^\star$ aand increments*

$$\Delta\hat{\Theta}(t) = -\left[(\lambda+\beta)I_{Np} + P(t+1)\right]^{-1}\left[(\lambda I_{Np} + P(t+1))\hat{\Theta}(t) - \lambda\Theta^\star + S(t+1)\right], \tag{14}$$

*where, for $t = 1, \ldots, T$, the matrices $P(t)$ and $S(t)$ in* (14) *are determined by*

$$\begin{cases} P(T) = [w_1^\star I_p, \cdots, w_N^\star I_p]^\top \left(\sum_{i=1}^N w_i^\star \sum_{j=1}^{|\mathcal{D}_i|} u_j^i u_j^{i\top}\right)[w_1^\star I_p, \cdots, w_N^\star I_p] \\ P(t) = \beta I_{Np} - \beta^2[(\lambda+\beta)I_{Np} + P(t+1)]^{-1} \end{cases}, \tag{15}$$

$$\begin{cases} S(T) = -[w_1^\star I_p, \cdots, w_N^\star I_p]^\top \sum_{i=1}^N w_i^\star \sum_{j=1}^{|\mathcal{D}_i|} u_j^i y_j^i \\ S(t) = \beta\left[(\lambda+\beta)I_{Np} + P(t+1)\right]^{-1}(S(t+1) - \lambda\Theta^\star) \end{cases}. \tag{16}$$

*In particular, for each $t = 0, \ldots, T-1$*

$$\Delta\hat{\Theta}(t) = \boldsymbol{\alpha}(t),$$

*where $\boldsymbol{\alpha}$ is as in Algorithm 2.*

**Remark 1.** *Instead of the sums we can equivalently use our matrix notation to write $\sum_{j=1}^{|\mathcal{D}_i|} u_j^i u_j^{i\top} = U^{i\top}U^i$ and $\sum_{j=1}^{|\mathcal{D}_i|} u_j^i y_j^i = U^{i\top}Y^i$.*

The explicit update rules for the regret-optimal optimization algorithm presented above permit us to compute its complexity. Since we will be assuming that we work with *floating-point arithmetic*, all elementary arithmetic operations (e.g. addition, subtraction, multiplication) are of constant (i.e. $\mathcal{O}(1)$) complexity. We are most interested in the regime where the number of datasets is much larger than the typical number of instances/data per dataset; i.e. when

$$\frac{1}{N}\sum_{i=1}^N |\mathcal{D}_i| \le N. \tag{17}$$

**Theorem 3** (Complexity of Regret-Optimal Algorithm)**.** *In the setting of Theorem 2, the computational complexity[4] of computing the sequence $(\hat{\Theta}(t))_{t=0}^T$ defining the regret-optimal algorithm is $\mathcal{O}\left(N^2 p^3 + T(Np)^{2.373}\right)$.*

---

[4]We emphasize that we have assumed that computational complexity of all elementary arithmetic operations (e.g. addition, subtraction, multiplication) is $\mathcal{O}(1)$.

Next, we quantify the sensitivity of Algorithm 2 to adversarial attacks. We study the robustness of Algorithm 2 with respect to a family of perturbations that switch a given percentage $0 \le q \le 1$ of all training data by selecting fake data points at a distance of at-most $\varepsilon \ge 0$ for any perturbed datapoint. Thus, for a fixed set of datasets $\{\mathcal{D}_i\}_{i=1}^N$, and $q, \varepsilon$ as above, we define the class $\mathbb{D}^{q,\varepsilon}$ of *adversarially generated datasets* of "magnitude" $(q, \varepsilon)$ as all sets of datasets $\{\widetilde{\mathcal{D}}_i\}_{i=1}^N$, $\widetilde{\mathcal{D}}_i = \{\{(\tilde{x}_j^i, \tilde{y}_j^i) \,|\, 1 \le j \le |\mathcal{D}_i|\} \subseteq \mathbb{R}^d \times \mathbb{R}$ satisfying Assumptions 1 and 2, as well as

$$\underbrace{\frac{\#\{(i,j):\, u_j^i \ne \tilde{u}_j^i \text{ or } y_j^i \ne \tilde{y}_j^i\}}{\sum_{i=1}^N |\mathcal{D}_i|} \le q}_{\text{Attack Persistence}} \text{ and } \underbrace{\max_{i,j} \big\{ \max \big( \|u_j^i - \tilde{u}_j^i\|_2, |y_j^i - \tilde{y}_j^i|^2 \big) \big\} \le \varepsilon,}_{\text{Attack Severity}}$$

where the indices $(i,j)$ run over $\{(i,j)\,|\,1 \le i \le N,\, 1 \le j \le |\mathcal{D}_i|\}$ and $\tilde{u}_j^i \stackrel{\text{def.}}{=} \phi(\tilde{x}_j^i)$.

In view of Theorem 2, the updates of the unique regret-optimal algorithm $\hat{\Theta}^{0 \cdots T}$ are determined by the sequence $\boldsymbol{\alpha} \stackrel{\text{def.}}{=} (\boldsymbol{\alpha}(t))_{t=0}^T$ computed by Algorithm 2. It is convenient to emphasize this connection on $\boldsymbol{\alpha}$. and the connection with the dataset $\mathcal{D}$ when expressing the following theorem, with the energy (6) determining the value of the systematic regret functional $\mathcal{R}$. We therefore write $\mathcal{L}(\boldsymbol{\alpha}; \mathcal{D}) \stackrel{\text{def.}}{=} \mathcal{L}(\hat{\Theta}^{0 \cdots T})$.

**Theorem 4** (Adversarial Robustness)**.** *Fix a $\mathcal{D}$ satisfying Assumptions 1 and 2. For any level of "attack persistence" $0 \le q \le 1$ and any degree of "attack severity" $\varepsilon \ge 0$ we have that*

$$\big| \mathcal{L}(\boldsymbol{\alpha}; \mathcal{D}) - \mathcal{L}(\widetilde{\boldsymbol{\alpha}}; \widetilde{\mathcal{D}}) \big| \le \mathcal{O}\Big( \varepsilon \sqrt{\bar{N} q} \Big),$$

*where $\widetilde{\boldsymbol{\alpha}}$ is the output of Algorithm 2 for any given adversarially-generated set of datasets $\{\widetilde{\mathcal{D}}_i\}_{i=1}^N \in \mathbb{D}^{q,\varepsilon}$ and $\bar{N} = \sum_{i=1}^N |\mathcal{D}_i|$. Here $\mathcal{O}$ hides a constant depending only on Assumption 1.*

### 2.4 An Accelerated Heuristic

We now discuss a situation in which the computational complexity of the regret-optimal algorithm can be reduced. This is achieved by exploiting symmetries in the optimal policy computed by Algorithm 2.

Examining the explicit update rules, we notice that *if* the weights $w$ defining the energy (6) were to be $\overline{1/N} \stackrel{\text{def.}}{=} (1/N, \dots, 1/N)$ then the matrices defining the optimal policy/control's updates would become highly symmetric Toeplitz matrices, depending on exactly three $p$-dimensional block sub-matrices. This allows us to parameterize the entire regret-optimal algorithm, for the sub-optimal weights $\overline{1/N}$, using only three low-dimensional systems whose dimension is *independent* of the number of datasets $N$. In line with this, Algorithm 3 implements the regret-optimal algorithm for the weight specification $\overline{1/N}$ by only keeping track of these three optimal low-dimensional systems, and then efficiently recombining them by leveraging the Toeplitz structure defining the optimal policy/control. These low-dimensional systems are defined using *helper functions*, listed in equations (18)-(20).

The approximate near-optimality of Algorithm 3 only depends on the stability of the optimal policy/control, defining the algorithm's updates as a function of the weights $w$. If the optimal weights defining the systemic regret functional (5) are close to $\overline{1/N}$, then Algorithm 3 is nearly regret-optimal. Otherwise, the sub-optimality it incurs in favor of computational efficiency is explicitly quantifiable and depends only on the difference between the optimal weights computed in Algorithm 1 and the symmetric weights $\overline{1/N}$.

Note now record, the matrix-valued functions $\gamma_1(\cdot)$ and $\gamma_2(\cdot)$ used to define the updates in Algorithm 3.

The matrix-valued functions $\gamma_1(\cdot)$ and $\gamma_2(\cdot)$ in Algorithm 3 are defined as

$$\gamma_1(\cdot) \stackrel{\text{def.}}{=} \big\{ (\lambda + \beta) I_p + \pi_1(\cdot)$$
$$- (N-1)\pi_2(\cdot) \big[ (\lambda + \beta) I_p + \pi_1(\cdot) + (N-2)\pi_2(\cdot) \big]^{-1} \pi_2(\cdot) \big\}^{-1}, \tag{18}$$

$$\gamma_2(\cdot) \stackrel{\text{def.}}{=} -\gamma_1(\cdot)\pi_2(\cdot) \big[ (\lambda + \beta) I_p + \pi_1(\cdot) + (N-2)\pi_2(\cdot) \big]^{-1}, \tag{19}$$

---

**Algorithm 3:** Accelerated (Nearly) Regret-Optimal Algorithm

---

**Require:** Datasets $\mathcal{D}_1, \ldots, \mathcal{D}_N$, $N$. Iterations $T \in \mathbb{N}_+$, finite-rank kernel $\phi$, hyperparameters $\lambda, \beta, \kappa, \eta > 0$.

   // Get Initialize Weights and Locally-Optimal fRKR Parameters

   $\theta_1^\star, \ldots, \theta_N^\star, w^\star \leftarrow$ Run: Algorithm 1 with $\mathcal{D}_1, \ldots, \mathcal{D}_N, \phi$, and $\kappa, \eta$.

   // Initialize Updates

   $\pi_1(T) = \pi_2(T) = (1/N^3) \sum_{i=1}^{N} \sum_{j=1}^{|\mathcal{D}_i|} u_j^i u_j^{i\top}$

   $\pi_3(T) = -(1/N^2) \sum_{i=1}^{N} \sum_{j=1}^{|\mathcal{D}_i|} u_j^i y_j^i$

   $\theta_1(0) = \theta_1^\star$, $\theta_2(0) = \theta_2^\star$, $\ldots$, $\theta_N(0) = \theta_N^\star$

   $\theta_{(N)}^\star = \frac{1}{N} \sum_{i=1}^{N} \theta_i^\star$.

   // Generate Iterates

   **for** $t = T - 1, \ldots, 0$ **do**

      // Update Low-Dimensional Driving Parameters

      $\pi_1(t) = \beta I_p - \beta^2 \gamma_1(t+1)$ // $\gamma_1(\cdot)$ is defined by (18)

      $\pi_2(t) = -\beta^2 \gamma_2(t+1)$ // $\gamma_2(\cdot)$ is defined by (19)

      $\pi_3(t) = \beta \big[\gamma_1(t+1) + (N-1)\gamma_2(t+1)\big](\pi_3(t+1) - \lambda\theta_{(N)}^\star)$

      **end for**

   **for** $t = 0, \ldots, T - 1$ **do**

      // Update Low-Dimensional Control

      **for** $i = 1, \ldots, N$ **do**

         $\widehat{\alpha}_i(t) = \widehat{\alpha}\big(\theta_i(t),\ (\theta_j(t))_{j=1, j\neq i}^{N},\ (\gamma_i(t+1))_{i=1}^{2},\ (\pi_i(t+1))_{i=1}^{3}\big)$

         $\theta_i(t+1) = \theta_i(t) + \widehat{\alpha}_i(t)$

      **end for**

      **end for**

   **return** Synchronized fKRR $f_{\theta w^\star}$

---

where

$$\pi_1(T) = \pi_2(T) = \frac{1}{N^3} \sum_{i=1}^{N} \sum_{j=1}^{|\mathcal{D}_i|} u_j^i u_j^{i\top} \qquad \text{and} \qquad \pi_3(T) = -\frac{1}{N^2} \sum_{i=1}^{N} \sum_{j=1}^{|\mathcal{D}_i|} u_j^i y_j^i$$

and, for $t = T - 1, \ldots, 0$,

$$\pi_1(t) = \beta I_p - \beta^2 \gamma_1(t+1)$$
$$\pi_2(t) = -\beta^2 \gamma_2(t+1)$$
$$\pi_3(t) = \beta\big[\gamma_1(t+1) + (N-1)\gamma_2(t+1)\big]\left(\pi_3(t+1) - \lambda\theta_{(N)}^\star\right)$$

where $\theta_{(N)}^\star = \frac{1}{N} \sum_{i=1}^{N} \theta_i^\star$.

The updates of the low-dimensional systems which Algorithm 3 exploits are implemented via the helper function $\hat{\alpha}$, defined as

$$\widehat{\alpha}\big(\theta_i, ((\theta_j)_{j=1, j\neq i}^N), (\gamma_i)_{i=1}^2, (\pi_i)_{i=1}^3\big) \stackrel{\text{def.}}{=} -(\gamma_1 - \gamma_2)(\lambda I_p + \pi_1 - \pi_2)\theta_i$$
$$- \big\{\gamma_1\pi_2 + \gamma_2[\lambda I_p + \pi_1 + (N-2)\pi_2]\big\} \sum_{j=1}^{N} \theta_j$$
$$- \big[\gamma_1 + (N-1)\gamma_2\big](\pi_3 - \lambda\theta_{(N)}^\star). \tag{20}$$

The degree to which Algorithm 3 is nearly regret-optimal depends on how far the weights $w$ determined by Algorithm 1 are from the equal weighting $\overline{1/N}$. This is precisely quantified by the following results.

**Theorem 5** (Near-Regret Optimality)**.** *Suppose that $\mathcal{D}$ satisfies Assumptions 1 and 2. and let $\hat{\Theta}^{0\cdots T}$ denote the unique regret-optimal algorithm[5] computed according to Algorithm 2. Then, the output $\vartheta^{0\cdots T}$ of Algorithm 3 satisfies*

*(i)* **Near-Optimality:** *The difference $\left| \mathcal{L}(\Theta^{0\cdots T}) - \mathcal{L}(\vartheta^{0\cdots T}) \right|$ is of the order*

$$\mathcal{O}\Big( \bar{N} N^3 \max_{1\le i,k,l\le N} \Big| \frac{1}{N^3} - w_k^\star w_l^\star w_i^\star \Big| + \bar{N} N^{3/2} \max_{1\le k,l\le N} \Big| \frac{1}{N^2} - w_k^\star w_l^\star \Big|$$
$$+ \bar{N} \max_{1\le i\le N} \Big| \frac{1}{N} - w_i^\star \Big| + N \big\| \Theta^\star - \Theta_{(N)}^\star \big\|_2 \Big),$$

*(ii)* **Near-Optimal Increments:** *The difference $\| \Delta\Theta^{0\cdots T-1} - \Delta\vartheta^{0\cdots T-1} \|_{\ell^1}$ is of the order*

$$\mathcal{O}\Big( \bar{N} N^{3/2} \max_{1\le i,k,l\le N} \Big| \frac{1}{N^3} - w_k^\star w_l^\star w_i^\star \Big|$$
$$+ \bar{N} N^{1/2} \max_{1\le k,l\le N} \Big| \frac{1}{N^2} - w_k^\star w_l^\star \Big| + \big\| \Theta^\star - \Theta_{(N)}^\star \big\|_2 \Big),$$

*where $\mathcal{O}$ hides a constant depending only on $\lambda$, $\beta$, $\Theta^\star$, $(w_1^\star, ..., w_N^\star)$, $T$, $K_x$, and in $K_y$.*

The next result shows that Algorithm 3 accelerates the regret-optimal algorithm by a linear factor in $N$ and by a quadratic factor in $p$, when $N$ is large.

**Theorem 6** ($\mathcal{O}(N p^2)$-Acceleration over Regret-Optimal Algorithm)**.** *With the convention that floating-point arithmetic operations are $\mathcal{O}(1)$, we have that:*

*(i)* **Complexity:** *Algorithm 3 has a complexity of $\mathcal{O}(TNp \max\{N, p\}))$*

*(ii)* **Acceleration over Regret-Optimal Algorithm:** *The regret-optimal algorithm has a complexity of $\mathcal{O}(TN^3 p^3)$.*

*When $N \ge p$, Algorithm 3 requires $\mathcal{O}(N p^2)$ fewer operations than the regret-optimal algorithm.*

Finally, we establish that Algorithm 3 is the regret-optimal algorithm, optimizing the systemic regret functional (6), in the "symmetric" case where the optimal weights $w^\star$ are all $(1/N, \ldots, 1/N)$.

**Theorem 7** (Regret-Optimality in the Symmetric Case)**.** *Suppose that $w_i^\star = 1/N$ for $i = 1, \ldots, N$. Then, $\hat{\Theta}^{0\cdots T} = \vartheta^{0\cdots T}$.*

## 3 Experiments

We now experimentally verify our theoretical results, with a focus on quantitative finance. In particular, we implemented the regret-optimal algorithm as well as its accelerated version and provide them together with the code for all experiments described in this section at https://github.com/???.

First, we conduct a convergence analysis of our regret-optimal algorithm, comparing it to standard gradient descent. We then demonstrate the transfer learning capabilities of our algorithm in the context of American option pricing. We use the randomized neural networks approach of Herrera et al. (2023) as our fKRR model, since it best balances computational efficiency and expressiveness amongst the available non-linear versions of the Longstaff & Schwartz (2001) algorithm for American option pricing.

---

[5]That is, the unique optimizer of (6)

### 3.1 Convergence of the Regret-Optimal Algorithm

We first study the evolution of the energy functional (6) and loss (3) achieved in each iteration by our regret-optimal Algorithm 2 with weights initialized by Algorithm 1. That is to say, we compute $\mathcal{L}(\Theta^{0\cdots t})$ as defined in (6) but for intermediate iterations $t$, as well as the loss $l(\Theta(t), w; \mathcal{D})$. We compare the regret-optimal algorithm with its accelerated version, given by Algorithm 3 with weights given by Algorithm 1, where we use $\Theta(0) = \Theta^\star_{(N)}$ and replace $\Theta^\star$ by $\Theta^\star_{(N)}$ in (6). Moreover, we compare to the results when using gradient descent to optimize the loss (3) directly. For all algorithms we use a ridge coefficient $\kappa = 0$, regret coefficients $\lambda = 0$, $\beta = 1$ and Algorithm 1 with information sharing level $\eta = 10$ to get the corresponding optimal weights $w^\star$ and to initialize $\Theta(0)$ with the locally optimal parameters. We note that the comparison of our algorithms to gradient descent optimizing the loss (3) only makes sense in the $\lambda = 0$ regime, since otherwise the objectives and therefore the behaviour are different. On the other hand, using $\beta > 0$ doesn't change the objective, but only forces our algorithm to make equally sized steps $\Delta\Theta(t)$, where the step size depends on $\beta$ and $T$.

For our convergence analysis we use $N = 5$ randomly generated datasets. In particular, each dataset $\mathcal{D}_i$ corresponds to a neural network $g_{\omega_i}$ with fixed randomly-generated hidden weights and biases. Here $\omega_i$ denotes the vector of all these hidden weights and bias parameters. The networks have $d = 5$ dimensional input, one hidden layer with 10 hidden nodes, ReLU activation function and map to a 1-dimensional output. The dataset is generated by sampling $S_i = 100$ inputs $x^i_j$ from a $d$-dimensional standard normal distribution and the corresponding target values are given by $g_{\omega_i}(x^i_j)$, i.e.,

$$\mathcal{D}_i = \{(x^i_j, g_{\omega_i}(x^i_j)) \,|\, 1 \leq i \leq S_i, x^i_j \sim N(0, I_d)\}.$$

Our feature map $\phi$ in (1) is a randomized neural network[6] with one hidden layer, 500 hidden nodes and ReLU activation as studied by Mei et al. (2021); Gonon et al. (2020); Gonon (2023). In particular, the weights of the hidden layers are randomly initialized and fixed and only the weights of the last layer, corresponding to $\theta$ in (1), are optimized. We note that by our choice for the number of hidden nodes to equal the total number of samples, the algorithms can in principle learn to have perfect replication on the training set, i.e., loss $l(\Theta(t), w; \mathcal{D}) = 0$. For a fair comparison between the algorithms, we use the same randomly generated datasets and the same randomized neural networks for each of the methods.

Figure 2 shows the averaged results for 10 runs each with a different set of randomly generated datasets and a different randomized neural network as fKRR. For the regret-optimal algorithm and for its accelerated version we use $T = 10^3$ iterations, while we use $10^5$ steps for gradient descent with learning rate $7 \cdot 10^{-5}$. We note that for learning rates $\geq 8 \cdot 10^{-5}$ the training becomes unstable such that the loss explodes. The gradient descent method gradually but slowly reduces the loss. Since the gradient descent method makes so small parameter updates (and since $\lambda = 0$), the regret is nearly 0 and the energy $\mathcal{L}$ nearly equal to the loss. Even though it runs 100 times the number of iterations of our algorithms, it would still need much more training to achieve the same loss. On the other hand, the regret-optimal algorithm and its accelerated version behave nearly indistinguishably, reducing the loss nearly to 0 and piling up a little bit of regret. We note that by increasing the number of iterations, our algorithms could reduce the loss further by making smaller parameter updates and therefore piling up less regret, allowing them to focus even more on reducing the loss.

In the regime where $\lambda > 0$ the behaviour of the algorithm changes drastically. In particular, since deviating from $\Theta(0) = \Theta^\star$ is penalised, the algorithm remains at the starting point in the initial steps and only starts to converge when approaching $T$. The exact behaviour depends on the choice of $\lambda$ (and is additionally influenced by the choice of $\beta$). In Figure 3 we compare the behaviour for $\lambda = 10^{-4}$ and $\lambda = 2$ while otherwise using the same setup as before. For the smaller value of $\lambda$, the algorithm starts the decent earlier and decreases the loss by more compared to the larger value for $\lambda$, where the algorithm deviates from $\Theta^\star$ only in the very last steps.

### 3.2 American Option Pricing

We test our algorithm in the context of the optimal stopping problem for American option pricing, a computationally highly challenging problem which has recently become tractable also in higher dimensional

---

[6]Also called a random feature model or an extreme learning machine

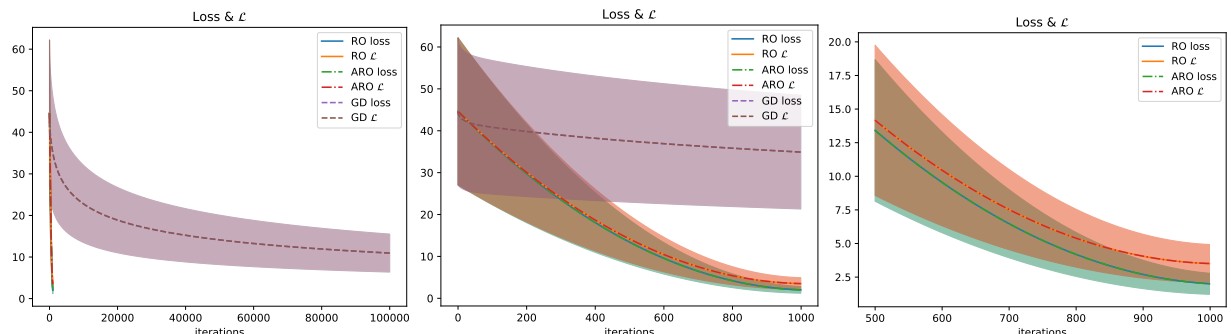

Figure 2: Loss and energy $\mathcal{L}$ for gradient descent (GD), the regret-optimal algorithm (RO) and the accelerated regret-optimal algorithm (ARO). Means and standard deviations over 10 runs are shown. Left: all iterations; middle: first 1000 iterations; right: iterations 500 to 1000 for our algorithms.

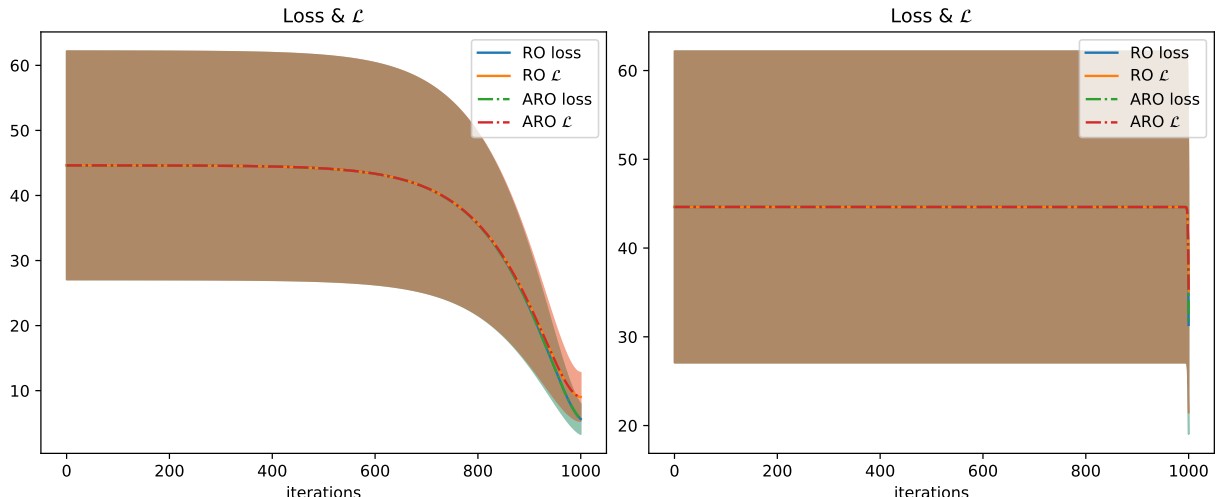

Figure 3: Loss and energy $\mathcal{L}$ for the regret-optimal algorithm (RO) and the accelerated regret-optimal algorithm (ARO). Means and standard deviations over 10 runs are shown. Left: $\lambda = 10^{-4}$; right: $\lambda = 2$.

settings through deep learning (Becker et al., 2019; Hu, 2020; Becker et al., 2021; Herrera et al., 2023). This is a good example for our purpose for two reasons. On the one hand, Herrera et al. (2023) did show that randomized neural networks, which are one class of fKRR, are well suited to (approximately) solve these optimal stopping problems. Moreover, it was shown that using randomized neural networks can heavily improve the computation time, which is also of importance to us since we need to solve several optimizations problems for a potentially large number of datasets. On the other hand, to solve the optimal stopping problem one needs to generate synthetic datasets. Hence, this problem allows us to generate multiple synthetic datasets of different sizes and varying similarity on which we can test the transfer-learning capabilities of our algorithm. The performance can be measured by comparing the achieved prices with those of several baseline methods.

We first provide a short recap of the optimal stopping problem in American option pricing and then present our transfer-learning experiments.

### 3.2.1 Background: American Option Pricing and Optimal Stopping

An American option is a financial derivative that gives its owner the right to execute a certain predetermined trade at any time between the start date and the maturity of the option. For example, a call (put) option gives its owner the right to buy (sell) a certain stock from the writer of the option for a fixed price $K$, called

the strike. If the current stock price $X_t$ is higher (lower) than $K$ then the option owner can make a profit of $(X_t - K)_+$ (or $(K - X_t)_+$ respectively) by directly selling (buying) the stock again in the open market after having exercised their right to buy (sell) from the option writer. While American options allow for continuous-in-time execution, it is common practice to approximate them via Bermudan options, which can be executed on a predetermined time grid $t_0 < \cdots t_M$. We briefly overview the American option pricing problem, details of which can be found in (Peskir & Shiryaev, 2006, Chapter 25).

In order to capture the computational challenges posed by the American option pricing problem in higher dimensions, we consider a stochastic process $(X_t)_{t=0}^M \in (\mathbb{R}^d)^{(M+1)}$ describing the price of $d$ stocks on the time grid and let $(Z_t)_{t=0}^M \in \mathbb{R}^{(M+1)}$ be the corresponding discounted payoff process of an American option written on a one-dimensional process obtained from $X_t$ (e.g. the largest price among the $d$ stocks). We define the filtration $\mathbb{F} = (\mathcal{F}_t)_{t=0}^M$ generated by $X$ via $\mathcal{F}_t = \sigma(X_s | s \leq t)$. Then the price of the American option is given by the starting value $U_0$ of the *Snell envelope* (Snell, 1952), given by

$$U_m = \sup_{\tau \in \mathcal{T}_m} \mathbb{E}[Z_\tau \,|\, \mathcal{F}_m], \tag{21}$$

where $\mathcal{T}_m$ is the set of all stopping times $\tau \geq m$. The smallest optimal stopping time is

$$\tau_M := M, \tag{22}$$
$$\tau_m := \begin{cases} m, & \text{if } Z_m \geq \mathbb{E}[U_{m+1} \,|\, \mathcal{F}_m], \\ \tau_{m+1}, & \text{otherwise,} \end{cases}$$

hence, to compute the price of the American option it suffices to compute the *continuation values* $\mathbb{E}[U_{m+1} \,|\, \mathcal{F}_m]$ for all $m$. In the following, we will use the *randomized least squares Monte Carlo* (RLSM) algorithm of Herrera et al. (2023), which utilises randomized neural networks to approximate the continuation values via backward induction (note the backward recursive scheme that (21) and (22) imply). In particular, this leads to $M$ backward-recursively defined regression problems that need to be solved sequentially. Since (21) is a maximisation problem and since we only approximate the optimal stopping time, we compute lower bounds of the price $U_0$. The standard procedure is to fix some law for the process $X$ and to sample a dataset of $\nu = \nu_1 + \nu_2$ i.i.d. paths of $X$, where $\nu_1$ of them are used for training the randomized neural networks (approximating the continuation value functions) and $\nu_2$ of them are used to evaluate the resulting price $U_0$ (using these trained networks). Through the independence of the two parts of the dataset it is clear that any over-fitting bias the networks might learn will *not* lead to prices that are too large. This is important, since the quality of a method can therefore be evaluated by the resulting price, using a higher price indicating a a better result.

### 3.2.2 Transfer Learning in American Option Pricing

In order to test the transfer learning capabilities of our algorithm, we do not generate a very large dataset of the most plausible distribtution of $X$ as is standard practice. Instead we generate many smaller, more or less similar datasets $\mathcal{D}_1, \ldots, \mathcal{D}_N$ out of which $\mathcal{D}_1$ is considered our main dataset on which we evaluate the performance.

We always use the RLSM algorithm to price the American options and only vary the optimization method used internally by RLSM to solve the $M$ recursive regression problems arising at the exercise dates. Depending on the chosen optimisation method only one, multiple or all datasets are taken into account to compute the parameters of the randomized neural network at the given exercise date. We compare our regret-optimal (RO) optimisation method of Theorem 2 to the following baselines.

- **Local optimizer (LO-$i$).** The local optimizers $\theta_i^\star$ solves the individual optimisation problem on the datasets $\mathcal{D}_i$ without considering the other datasets.

- **Mean Local optimizer (MLO).** The mean of the local optimizers $\bar{\theta}^\star = \frac{1}{N} \sum_{i=1}^N \theta_i^\star$ is the bagging aggregation.

- **Joint optimizer (JO).** The joint optimizer $\theta_\mathcal{N}^\star$ with $\mathcal{N} = \{1, \ldots, N\}$ solves the optimisation problem of the pooled regime $\mathcal{D} = \cup_{i=1}^N \mathcal{D}_i$.

- **Joint Subset optimizer.** This is the joint optimizer for any subset $\mathcal{I} \subset \mathcal{N}$ of the datasets $\mathcal{D}_\mathcal{I} = \cup_{i \in \mathcal{I}} \mathcal{D}_i$.

In this application we are only interested in the terminal parameters $\Theta(T)$. We know that our algorithm yields the parameters optimising (6) for any $T \geq 1$ since there exists a closed-form solution. Hence, we choose $T = 1$ which is most specialized on minimizing (3), while the influence of the running costs (26) can be controlled via the choice of $\lambda$ and $\beta$. Additionally, this minimizes the computation time for our algorithm.

As fKRR we use the same randomized neural networks as in Section 3.1. The random weights of the randomized neural networks, as well as the randomly sampled paths both introduce some randomness into the pricing problem. Therefore we follow Herrera et al. (2023) by always running the experiments $n_{runs} = 100$ times with different seeds and taking means over the prices. For easier comparison, we then consider the relative performance ($RP$) on our main dataset, computed as the mean price of any given method divided by the mean price of the local optimizer on the main dataset. Additionally, we compute asymptotically valid (by the central limit theorem) 95%-confidence-intervals as $RP \pm z_{0.975}\hat{\sigma}/\sqrt{n_{runs}}$, where $z_{0.975}$ is the 0.975-quantile of the standard normal distribution and $\hat{\sigma}$ is the standard deviation of $RP$ (computed as the sample standard deviation of the prices over the runs divided by the mean price of the local optimizer on the main dataset). To make the results comparable between the optimization methods, we use the same seeds for all of them, such that run $i$ always uses the exact same paths and the exact same random weights in the randomized neural network, independently of the chosen method. Moreover, we always use $\nu_2 \geq 50,000$ evaluation paths, such that the noise of the Monte Carlo approximation of the expectations is relatively small when evaluating the prices, even if we use much smaller training set sizes $\nu_1$.

In the first experiment we study the ability of our method to transfer knowledge of those datasets, out a large number of small equally-sized datasets, benefiting the main task.

**Experiment 1.** In this experiment we generate 13 different datasets, all of a Heston model with $\nu_1 = 100$ training samples but with different parameters. The Heston model is defined as

$$
\begin{aligned}
dX_t &= (r - \delta)X_t dt + \sqrt{v_t}X_t dW_t, \\
dv_t &= -k(v_t - v_\infty)dt + \sigma\sqrt{v_t}dB_t,
\end{aligned}
\tag{23}
$$

where $W$ and $B$ are two Brownian motions with correlation $\rho \in [-1, 1]$. The main dataset $\mathcal{D}_1$ has the parameters $r = 0.05$, $\delta = 0.1$, $k = 2$, $v_\infty = 0.01$, $\sigma = 0.2$, $\rho = -0.3$ and starting values $X_0 = 100$ and $v_0 = v_\infty$. The other datasets are define as all combinations of $r \in \{0.05, 0.5\}$, $\sigma \in \{0.15, 0.2, 0.25\}$, $v_\infty \in \{0.005, 0.015\}$ and otherwise the same parameters as the main dataset. For all datasets we use a max call option with strike $K = 100$ on $d = 2$ i.i.d. stocks of the given model with $M = 9$ equidistant exercise dates until the maturity $T_{mat} = 3$. The discounting factor is $e^{-r\,T_{mat}\,t/M}$ for $0 \leq t \leq M$. Moreover, we use a randomized neural network with one hidden layer with 300 nodes and all methods use the ridge coefficient $\kappa = 2$ and, where applicable, regret coefficients $\lambda = 2, \beta = 1$.

We report the results in Table 1. For our regret-optimal method (RO) we use information sharing levels $\eta \in \{10, 100, 500\}$ and see that $\eta = 100$ leads to the best result outperforming the local optimizer of the main dataset (LO-1) by 9%. It also outperforms all other local optimizers (LO-$i$), the mean local optimizer (MLO) and the joint optimizer (JO). For $\eta = 10$ we have similar performance as LO-1 and for $\eta = 500$ we also have significant outperformance, though smaller than for $\eta = 100$.

The 6 datasets with $r = 0.05$ (LO-2, ..., LO-7) are relatively similar to the main dataset (with $RP \geq 0.96$), while the other 6 (LO-8, ..., LO-13) are quite different (with $RP < 0.73$). Therefore, we claim that our algorithm should mainly transfer knowledge from those 6 datasets with $r = 0.05$ while avoiding to learn something wrong from the others. In line with this we provide results of the joint optimizer on the 7 datasets with $r = 0.05$, which performs better than the other baselines and outperforms LO-1 by 6.2%. In particular, this confirms that the datasets with $r = 0.05$ are those to transfer knowledge from, since the joint optimizer for all datasets performs worse than LO-1. The joint subset optimizer is significantly outperformed by our regret-optimal algorithm RO ($\eta = 100$) by about 3%, which shows that our method constitutes a better way for transfer learning than simply selecting the datasets most similar to $\mathcal{D}_1$.

We additionally compare our method to the local optimizer for the main dataset which uses either 700 (the number of training samples in the 7 similar datasets with $r = 0.05$) or $50,000$ training samples. Importantly, both are just references that would not be available in a real world setting where the datasets are limited. The first one only leads to a slight increase in $RP$ of about 1%, which shows that our method extracts nearly as much knowledge out of the (same amount of) samples of the similar datasets with $r = 0.05$. The second one (which is considered a good proxy for the true American option price on the main dataset) has about 10% better performance. This also shows the limitations of our method, in particular, it can only transfer as much knowledge as available in the other datasets.

Table 1: Relative performance (RP) and 95%-confidence-intervals for different optimization methods in Experiment 1. We compare the local optimizers on the different datasets (LO-$n$), with the mean local optimizer (MLO), the joint optimizer (JO) and our regret-optimal method (RO). The "oracle" local optimizer on the main dataset with additional training samples (standard: 100 samples per dataset) is included. When the information sharing parameter $\eta$ is set to 100, the regret-optimal (RO) algorithm outperforms all "non-oracle" baselines.

| method | $RP$ | 95%-CI |
|---|---|---|
| LO-1 | 1.000 | $[0.991; 1.009]$ |
| LO-2 | 0.985 | $[0.977; 0.994]$ |
| LO-3 | 0.966 | $[0.958; 0.973]$ |
| LO-4 | 0.975 | $[0.966; 0.984]$ |
| LO-5 | 0.966 | $[0.957; 0.975]$ |
| LO-6 | 0.979 | $[0.971; 0.988]$ |
| LO-7 | 0.978 | $[0.971; 0.986]$ |
| LO-8 | 0.721 | $[0.711; 0.730]$ |
| LO-9 | 0.720 | $[0.711; 0.729]$ |
| LO-10 | 0.706 | $[0.696; 0.716]$ |
| LO-11 | 0.725 | $[0.715; 0.736]$ |
| LO-12 | 0.707 | $[0.697; 0.716]$ |
| LO-13 | 0.718 | $[0.708; 0.727]$ |
| MLO | 0.823 | $[0.813; 0.833]$ |
| JO | 0.886 | $[0.878; 0.894]$ |
| JO (datasets 1-7) | 1.062 | $[1.057; 1.067]$ |
| RO ($\eta = 10$) | 1.000 | $[0.990; 1.010]$ |
| **RO** ($\eta = 100$) | **1.090** | **$[1.086; 1.095]$** |
| RO ($\eta = 500$) | 1.050 | $[1.043; 1.057]$ |
| LO-1 (700 tr. samp.) | 1.100 | $[1.097; 1.104]$ |
| LO-1 (50K tr. samp.) | 1.194 | $[1.192; 1.195]$ |

In the second experiment we test the ability of our method to transfer knowledge from a small dataset which is similar to the small main dataset, even if it is confronted with a very different and much larger (dominating in terms of samples) dataset.

**Experiment 2.** In this experiment we generate 3 different datasets. The main dataset is generated from a rough Heston model

$$dX_t = (r - \delta)X_t dt + \sqrt{v_t}X_t dW_t,$$

$$v_t = v_0 + \int_0^t \frac{(t-s)^{H-1/2}}{\Gamma(H+1/2)} \kappa(v_\infty - v_s)ds + \int_0^t \frac{(t-s)^{H-1/2}}{\Gamma(H+1/2)} \sigma\sqrt{v_s}dB_s,$$

which is similar to the Heston model except that the volatility is a rough process with Hurst parameter $H \in (0, 1/2]$, where the case $H = 1/2$ coincides with the standard Heston model. We use $H = 0.1$ and otherwise the same parameters as for the main dataset in Experiment 1. The second and third dataset generated from standard Heston model (i.e. $H = 1/2$), with the same parameters, except for the drift $r = 0.5$ for the second dataset. In particular, dataset 3 is the closest non-rough dataset to the main dataset, while dataset 2 is quite different. For datasets 1 and 3 we use $\nu_1 = 100$ training samples, while we use $\nu_1 = 50,000$ for dataset 2, such that it dominates the others in its size. All the other choices are the same as in Experiment 1.

The results are reported in Table 2. The baselines MLO and JO perform significantly worse than the LO on the main dataset. As expected, the JO suffers much more from the dominating dataset 2 than the MLO. Indeed, the MLO would suffer more from a large number of datasets that are quite different from the main dataset, no matter their sample size. In contrast to this, our regret optimal method with $\eta \in \{50, 100\}$ outperforms the LO-1 by 4%, showing that also a quite different dataset dominating in sample size is not a problem.

Table 2: Relative performance (RP) and 95%-confidence-intervals for different optimization methods in Experiment 2. We compare the local optimizers on the different datasets (LO-$n$), the mean local optimizer (MLO), the joint optimizer (JO) and our regret-optimal method (RO). With $\eta = 100$ RO outperforms all baselines.

| method | $RP$ | 95%-CI |
|---|---|---|
| LO-1 | 1.000 | $[0.993; 1.007]$ |
| LO-2 | 0.773 | $[0.765; 0.782]$ |
| LO-3 | 1.003 | $[0.994; 1.012]$ |
| MLO | 0.932 | $[0.920; 0.944]$ |
| JO | 0.763 | $[0.754; 0.772]$ |
| RO ($\eta = 10$) | 0.993 | $[0.985; 1.000]$ |
| RO ($\eta = 50$) | 1.040 | $[1.034; 1.046]$ |
| **RO ($\eta = 100$)** | **1.040** | $[\mathbf{1.034; 1.047}]$ |

Both of these experiments show that our regret optimal method is well suited for automatic (up to the hyper-parameter $\eta$) dataset selection in transfer learning tasks that have potentially large numbers of datasets and do not allow a (manual) pre-selection. In particular, neither a large number of "bad" datasets (of which the MLO suffers) nor some "bad" datasets that dominate in sample size (of which the JO suffers) constitute a problem for our regret-optimal method. Moreover, the sample efficiency when transferring knowledge from similar datasets to the main task is very high (outperforming the JO on the respective subset of similar datasets and nearly being on par with the LO on the main dataset using a larger training sample), as we saw in Experiment 1.

## 4 Conclusion

In this work we presented a regret-optimal algorithm for federated transfer learning based on finite-rank kernel ridge-regressors (fKRR). Besides the theoretical properties of this algorithm, we also provided experiments demonstrating the transfer learning capabilities of our method in the context of American option pricing.

One possible future research direction is to apply our transfer learning method to real world datasets, where the similarity between the datasets is a priori not easy to quantify. From the theoretical side we would like to extend our method to regression models that do not necessarily permit a closed-form solution.

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

## A  Game Theoretic Interpretation of Regret-Optimal Algorithm

We now offer a game-theoretic interpretation of the regret functional, as defined in (6). This interpretation views the user as a central planner whose objective is to organize a system of individual agents, representing the pretrained KRR models, to maximize the singular goal of identifying a parameter optimizing (5). We now interpret the roles and interactions of the central planner and each agent as defined by our regret-optimization problem.

**The Central Planner:** The purpose of the central planner is to avoid the situation whereby the structure of the different datasets is ignored in the model selection problem, by merging them into a single dataset $\mathcal{D} \stackrel{\text{def.}}{=} \cup_{i=1}^{N} \mathcal{D}_i$ and thereby reducing the problem to (2). Though intuitively simple, such merging approach can be particularly disadvantageous in the presence of heterogeneity among the datasets, since a dataset that is potentially very different from the focal dataset $\mathcal{D}_1$ would be equally influential in the regression problem (2). Instead, the user, acting as a central planner, organizes the model selection problem through a cooperative objective

$$l(\Theta, w; \mathcal{D}) \stackrel{\text{def.}}{=} \sum_{i=1}^{N} w_i \sum_{j=1}^{|\mathcal{D}_i|} (f_{\theta^w}(x_j^i) - y_j^i)^2,$$

$$\text{with } \theta^w \stackrel{\text{def.}}{=} \sum_{i=1}^{N} w_i\, \theta_i, \tag{24}$$

where $\Theta \stackrel{\text{def.}}{=} (\theta_1^\top, \cdots, \theta_N^\top)^\top \in \mathbb{R}^{Np \times 1}$. Here, the central planner *ascribes* the influence of each agent on the model selection problem through the weight vector $w = (w_1, \cdots, w_N) \in [0, 1]^N$ with $\sum_{i=1}^{N} w_i = 1$. As can be seen in (24), this influence is two-fold: on the one hand, it impacts the cooperative objective $l$ in the aggregation of the SSE of each player; on the other hand, it affects the proportion of each player's preferred parameter choice entering into the collectively selected parameter $\theta^w$. The choice of $w$ is approached in different ways in the literature, e.g. by bagging reminiscent of mean-field games (Carmona et al., 2018), by data-driven weighting procedures (Baxter, 2000), Bayesian aggregation (Xue et al., 2007; Rothfuss et al., 2021; Pavasovic et al., 2022), or with dictionaries (Argyriou et al., 2006).

**The Agents:** When acting in isolation, each agent's preferred model is determined by optimizing a version of (2) using only their dataset. Under mild conditions[7] on $f$, this corresponds to an *individual* parameter selection

$$\theta_i^\star \in \text{argmin}_{\theta \in \Theta} \sum_{(x,y) \in \mathcal{D}_i} (f_\theta(x) - y)^2 + \kappa \, \|\theta\|_2, \tag{25}$$

for $i = 1, \ldots, N$, which we assume to be fixed and known to all agents prior to their *collective* parameter selection. Taking into account the other agents, each agent *acts* by specifying an iterative deterministic algorithm of the form (4), with the intent of maximizing their influence on the joint model selection problem (24). Thus, the $i^{th}$ player wants the jointly selected model, specified by $\theta^w(T) = \sum_{n=1}^{N} w_i\, \theta_i(T)$, to be as close to $\theta_i^\star$ as possible, thereby maximally encoding the characteristics of the $i^{th}$ dataset into the selected model $f_{\theta^w(T)}$. The accumulated *regret* which the $i^{th}$ player incurs by deploying an algorithm $\theta_i^{0\cdots T}$ that deviates from their preferred selection $\theta_i^*$ is measured by

$$\sum_{t=0}^{T-1} \underbrace{\lambda \, \|\theta_i(t+1) - \theta_i^\star\|_2^2}_{\text{Preference Strength}} + \underbrace{\beta \, \|\Delta\theta_i(t)\|_2^2}_{\text{Algorithm Stability}}, \tag{26}$$

where $\lambda > 0$ is a hyperparameter quantifying the player's *attachment* towards their preferred model choice $\theta_i^\star$ and the hyperparameter $\beta > 0$ quantifies the the stability of the algorithm during the iterations, where $\Delta\theta_i(t) \stackrel{\text{def.}}{=} \theta_i(t+1) - \theta_i(t)$.

The central planner then organizes the action $\Theta^{0\cdots T} \stackrel{\text{def.}}{=} \{\Theta(t)\}_{t=0}^{T}$ of the system of $N$ players to reach the objective (24) while encoding their individual regrets (26), where

$$\Theta(t) = (\theta_1(t)^\top, \ldots, \theta_N(t)^\top)^\top \in \mathbb{R}^{Np \times 1}.$$

---

[7]E.g. coercivity and lower semi-continuity in $\theta$.

This is achieved by coupling the individual agent's regret functionals (26) through the *systemic regret functional*

$$\mathcal{R}(\Theta^{0\cdots T}) \stackrel{\text{def.}}{=} \underbrace{l(\Theta(T), w; \mathcal{D})}_{\text{Cooperative Objective}} - l^\star + \sum_{t=0}^{T-1} \big( \underbrace{\lambda \left\| \Theta(t+1) - \Theta^\star \right\|_2^2}_{\text{Preference Strength}} + \underbrace{\beta \left\| \Delta\Theta(t) \right\|_2^2}_{\text{Algo. Stability}} \big), \tag{27}$$

where

$$\Theta^\star \stackrel{\text{def.}}{=} (\theta_1^{\star\top}, \cdots, \theta_N^{\star\top})^\top \in \mathbb{R}^{Np \times 1} \tag{28}$$

encodes the individual preferences of the players,

$$\Delta\Theta(t) \stackrel{\text{def.}}{=} \Theta(t+1) - \Theta(t) \tag{29}$$

quantifies the integrative updates of any candidate optimizing sequence $\Theta^{0\cdots T}$, and the *ideal terminal loss* is

$$l^\star \stackrel{\text{def.}}{=} \underbrace{\min_{\Xi \in \mathbb{R}^{Np \times 1}} l(\Xi, w; \mathcal{D})}_{\text{Cooperative Sub-optimality}} .$$

The term $(l(\cdot, \cdot; \cdot) - l^\star)$ measures the central planner's regret in failing to optimize (24). The algorithm stability term in (27) appears in forward-backward proximal splitting algorithms with quadratic objectives (e.g. Combettes & Pesquet (2016); Briceño Arias & Roldán (2022)) and several standard proximal algorithms (see (Bertsekas, 2015, Section 5.1)).

The systemic regret functional (27) acts as a performance criterion for any optimization algorithm, played by $N$ players, initialized at an arbitrary $\Theta(0)$, and running for $T$ iterations. In what follows, we take the initial condition of the algorithm as $\Theta(0) = \Theta^\star$ for simplicity, but other choices are possible. In other words, the regret of each player is initialized at zero, as they all start from their individually preferred parameter $\theta_i^\star$, and starts to accumulate as the system moves away from $\Theta^\star$ towards the optimizer of (27).

## B    Proofs and Technical Results

**Remark 2.** *We keep our notation light and use $u^\top v$ to implement the Euclidean inner-product between any two vectors $u, v \in \mathbb{R}^k$ for any $k \in \mathbb{N}_+$. We use $|\cdot|$ to denote the Euclidean norm of a vector or Fröbenius norm of a matrix, whichever is applicable. For two square matrices $A$ and $B$, $A \leq B$ ($A < B$, resp.) means that $B - A$ is positive semi-definite (positive definite, resp.); see (Lax, 2007, page 146 - Equation (7)) for details.*

Lemmas 1 and 2 give some elementary properties of matrix algebra, which will be used repeatedly in the subsequent proofs.

**Lemma 1.** *The Euclidean norm $|\cdot|$ of matrices has the following properties:*

i). *For any matrices $A \in \mathbb{R}^{n_1 \times n_2}$ and $B \in \mathbb{R}^{n_2 \times n_3}$, we have that $|AB| \leq |A| \cdot |B|$;*

ii). *For any matrices $A, B \in \mathbb{R}^{n_1 \times n_2}$, we have $|A + B| \leq |A| + |B|$.*

*Proof.* Denote $A = (A_{ij})_{1 \leq i \leq n_1, 1 \leq j \leq n_2}$ and $B = (B_{jk})_{1 \leq j \leq n_2, 1 \leq k \leq n_3}$. By the Cauchy-Schwarz inequality, we have for each column vector $B_{\cdot k}$ of $B$ that

$$|AB_{\cdot k}|^2 = \sum_{i=1}^{n_1} \big( \sum_{j=1}^{n_2} A_{ij} B_{jk} \big)^2 \leq \sum_{i=1}^{n_1} \big( \sum_{j=1}^{n_2} A_{ij}^2 \sum_{j=1}^{n_2} B_{jk}^2 \big) = \big( \sum_{i=1}^{n_1} \sum_{j=1}^{n_2} A_{ij}^2 \big) \sum_{j=1}^{n_2} B_{jk}^2 = |A|^2 \sum_{j=1}^{n_2} B_{jk}^2.$$

It then follows that

$$|AB|^2 = |(AB_{\cdot 1}, AB_{\cdot 2}, ..., AB_{\cdot n_3})|^2 = |AB_{\cdot 1}|^2 + |AB_{\cdot 2}|^2 + \cdots + |AB_{\cdot n_3}|^2$$

$$\leq |A|^2 \sum_{j=1}^{n_2} \sum_{k=1}^{n_3} B_{jk}^2 = |A|^2 |B|^2,$$

and assertion i) is proved.

For $A, B \in \mathbb{R}^{n_1 \times n_1}$, we have

$$|A + B|^2 = \sum_{i=1}^{n_1} \sum_{j=1}^{n_2} (A_{ij} + B_{ij})^2 = \sum_{i=1}^{n_1} \sum_{j=1}^{n_2} A_{ij}^2 + \sum_{i=1}^{n_1} \sum_{j=1}^{n_2} B_{ij}^2 + 2 \sum_{i=1}^{n_1} \sum_{j=1}^{n_2} A_{ij} B_{ij}$$
$$\leq |A|^2 + |B|^2 + 2|A|^2 |B|^2 = (|A| + |B|)^2,$$

which proves assertion ii). $\qquad\qquad\qquad\qquad\qquad\qquad\qquad\qquad\qquad\qquad\qquad\qquad\quad$ $\square$

**Lemma 2.** *Let $w_1, w_2, \ldots, w_n$ be positive numbers such that $\sum_{k=1}^{n} w_k = 1$. For any $n$ matrices $A_1, \ldots, A_n \in \mathbb{R}^{n_1 \times n_2}$, it satisfies that $\left| \sum_{k=1}^{n} w_k A_k \right|^2 \leq \sum_{k=1}^{n} w_k |A_k|^2$; particularly, when $w_1 = w_2 = \cdots = w_n = 1/n$, we have $\left| \sum_{k=1}^{n} A_k \right|^2 \leq n \sum_{k=1}^{n} |A_k|^2$ .*

*Proof.* Denote $A_k = (A_{k,ij})_{1 \leq i \leq n_1, 1 \leq j \leq n_2}$, $k = 1, \ldots, n$. By the Jensen's inequality, we have

$$(w_1 A_{1,ij} + \cdots + w_n A_{n,ij})^2 \leq w_1 A_{1,ij}^2 + \cdots + w_n A_{n,ij}^2.$$

It then follows that

$$|w_1 A_1 + \cdots + w_n A_n|^2 = \sum_{i=1}^{n_1} \sum_{j=1}^{n_2} (w_1 A_{1,ij} + \cdots + w_n A_{n,ij})^2 \leq \sum_{i=1}^{n_1} \sum_{j=1}^{n_2} w_1 A_{1,ij}^2 + \cdots + w_n A_{n,ij}^2$$
$$= w_1 \sum_{i=1}^{n_1} \sum_{j=1}^{n_2} A_{1,ij}^2 + \cdots + w_n \sum_{i=1}^{n_1} \sum_{j=1}^{n_2} A_{n,ij}^2$$
$$= w_1 |A_1|^2 + \cdots + w_n |A_n|^2.$$

$$\square$$

We begin with the relatively short proof of Theorem 1 establishing the optimality of the weights obtained in Algorithm 1.

*Proof of Theorem 1.* For $i = 1, \ldots, N$, consider the probability measure

$$\bar{\mathbb{P}}^\eta \stackrel{\text{def.}}{=} \sum_{i=1}^{N} \frac{[s^1 + \eta - s^i]_+}{\sum_{k=1}^{N} [s^1 + \eta - s^k]_+} \delta_{\theta_i^\star}$$

on the discrete space $\{\theta_i^\star\}_{i=1}^{N}$ topologized by the discrete metric. Any minimizer of (12) is a minimizer of the following problem, and vice-versa

$$\min_{\mathbb{P}_w \in \mathcal{P}(\{\theta_i^\star\}_{i=1}^N)} \frac{1}{|\mathcal{D}_1|} \sum_{j=1}^{|\mathcal{D}_1|} \left\| f_\theta(u_j^1) - y_j^1 \right\|_2^2 + \eta \, \mathrm{D_{KL}} \left( \mathbb{P}_w \| \bar{\mathbb{P}}^\eta \right)$$

$$= \min_{\mathbb{P}_w \in \mathcal{P}(\{\theta_i^\star\}_{i=1}^N)} \mathbb{E}_{\theta \sim \mathbb{P}_w} \left[ \frac{1}{|\mathcal{D}_1|} \sum_{j=1}^{|\mathcal{D}_1|} \left\| f_\theta(u_j^1) - y_j^1 \right\|_2^2 \right] + \eta \, \mathrm{D_{KL}} \left( \mathbb{P}_w \| \bar{\mathbb{P}}^\eta \right) + \iota(\mathbb{P}_w \ll \bar{P}^\eta)$$

$$= \min_{\mathbb{P}_w \in \mathcal{P}(\{\theta_i^\star\}_{i=1}^N)} \mathbb{E}_{\theta \sim \mathbb{P}_w} \left[ \sum_{i=1}^{N} I_{\theta = \theta_i^\star} s_i \right] + \eta \, \mathrm{D_{KL}} \left( \mathbb{P}_w \| \bar{\mathbb{P}}^\eta \right) + \iota(\mathbb{P}_w \ll \bar{P}^\eta), \qquad (30)$$

where the map $\iota(\mathbb{P}_w \ll \bar{P}^\eta)$ is an "indicator function in the sense of convex analysis"; that is, $\iota(\mathbb{P}_w \ll \bar{P}^\eta) = 0$ if $\mathbb{P}_w$ is absolutely continuous with respect to $\bar{P}^\eta$ and $\infty$ otherwise. We may now apply (Wang et al., 2020, Proposition 1)[8] to (30) deducing that it admits a minimizer $\mathbb{P}_{w^\star} = \sum_{i=1}^N w_i^\star \delta_{\theta_i^\star}$ and its Radon-Nikodym derivative with respect to the measure $\bar{\mathbb{P}}^\eta$ is

$$\frac{d\mathbb{P}_{w^\star}}{\bar{\mathbb{P}}^\eta}(\theta_i^\star) = \frac{e^{\sum_{l=1}^N I_{\theta_l^\star = \theta_i^\star} s_i/\eta}}{\mathbb{E}_{\bar{\mathbb{P}}^\eta}\left[\exp\left(\sum_{i=1}^N I_{\theta = \theta_i^\star} s_i\right)\right]} \tag{31}$$

$$= \frac{e^{s_i/\eta}}{\sum_{k=1}^N \frac{e^{s_k/\eta}[s^1 + \eta - s^k]_+}{\sum_{m=1}^N [s^1 + \eta - s^m]_+}}$$

$$= \left(\sum_{m=1}^N [s^1 + \eta - s^m]_+\right) \frac{e^{s_i/\eta}}{\sum_{k=1}^N e^{s_k/\eta}[s^1 + \eta - s^k]_+} \tag{32}$$

for $i = 1, \ldots, N$. Since $\mathbb{P}_{w^\star}$ is a minimizer of (30) then we may deduce the expression for the optimal weights $w^\star$ by first noting that

$$\mathbb{P}_{w^\star} \stackrel{\text{def.}}{=} \sum_{i=1}^N w_i^\star \delta_{\theta_i^\star} = \sum_{i=1}^N \frac{d\mathbb{P}_{w^\star}}{d\bar{\mathbb{P}}^\eta}(\theta_i^\star) \left(\frac{[s^1 + \eta - s^i]_+}{\sum_{j=1}^N [s^1 + \eta - s^j]_+}\right) \delta_{\theta_i^\star}. \tag{33}$$

Therefore, (31)- (32) and (33) imply that, for each $i = 1, \ldots, N$, we have

$$w_i^\star = \frac{d\mathbb{P}_{w^\star}}{d\bar{\mathbb{P}}^\eta}(\theta_i^\star) \left(\frac{[s^1 + \eta - s^i]_+}{\sum_{j=1}^N [s^1 + \eta - s^j]_+}\right)$$

$$= \left(\sum_{m=1}^N [s^1 + \eta - s^m]_+\right) \frac{e^{s_i/\eta}}{\sum_{k=1}^N e^{s_k/\eta}[s^1 + \eta - s^k]_+} \left(\frac{[s^1 + \eta - s^i]_+}{\sum_{j=1}^N [s^1 + \eta - s^j]_+}\right)$$

$$= \frac{e^{s_i/\eta}[s^1 + \eta - s^i]_+}{\sum_{k=1}^N e^{s_k/\eta}[s^1 + \eta - s^k]_+}.$$

$\square$

Next we now prove our main result, namely Theorem 2. The result is first rewritten, in the optimal control-theoretic notation introduced in Section 2.1.

**Theorem 8** (The Regret-Optimal Algorithm - Control Theoretic Form). *Suppose that Assumptions 1 and 2 hold. Then, the optimal control problem* (6) *and* (9) *admits a unique solution for $t = 0, 1, ..., T - 1$ such that*

$$\boldsymbol{\alpha}(t) = -\left[(\lambda + \beta)I_{Np} + P(t+1)\right]^{-1}\left[(\lambda I_{Np} + P(t+1))\Theta(t) - \lambda\Theta^\star + S(t+1)\right], \tag{34}$$

*and the resulting cost is*

$$L^\star(\boldsymbol{\alpha}; \mathcal{D}) = \Theta^{\star\top} P(0)\Theta^\star + 2S^\top(0)\Theta^\star + r(0). \tag{35}$$

*The $P(t)$, $S(t)$, and $r(t)$ in* (34) *and* (35) *are determined by the backward equation system on $t = 1, \ldots, T$:*

$$\begin{cases} P(t) = \beta I_{Np} - \beta^2[(\lambda + \beta)I_{Np} + P(t+1)]^{-1}, \\ P(T) = [w_1^\star I_p, \cdots, w_N^\star I_p]^\top \left(\sum_{i=1}^N w_i^\star \sum_{j=1}^{|\mathcal{D}_i|} u_j^i u_j^{i\top}\right)[w_1^\star I_p, \cdots, w_N^\star I_p], \end{cases} \tag{36}$$

$$\begin{cases} S(t) = \beta\left[(\lambda + \beta)I_{Np} + P(t+1)\right]^{-1}(S(t+1) - \lambda\Theta^\star), \\ S(T) = -[w_1^\star I_p, \cdots, w_N^\star I_p]^\top \sum_{i=1}^N w_i^\star \sum_{j=1}^{|\mathcal{D}_i|} u_j^i y_j^i, \end{cases} \tag{37}$$

---

[8]This result is an extension of (Dai Pra et al., 1996, Proposition 2.3) which allows us to vary the influence of the regulative-entropy term in (30).

$$\begin{cases} r(t) = -(S(t+1) - \lambda\Theta^\star)^\top [(\lambda + \beta)I_{Np} + P(t+1)]^{-1} \cdot (S(t+1) - \lambda\Theta^\star) \\ \qquad + \lambda\|\Theta^\star\|_2^2 + r(t+1), \\ r(T) = \sum_{i=1}^N w_i^\star \sum_{j=1}^{|\mathcal{D}_i|} |y_j^i|^2. \end{cases} \tag{38}$$

*Proof of Theorem 8 (and therefore of Theorem 2).* Let $V(t,\Theta)$ be the value function associated with the optimal control problem (6) and (9). By the dynamic programming principle, $V(t,\Theta)$ satisfies the equation

$$V(t,\Theta) = \min_{\boldsymbol{\alpha}} \left[ \lambda|\Theta + \boldsymbol{\alpha} - \Theta^\star|^2 + \beta|\boldsymbol{\alpha}|^2 + V(t+1, \Theta + \boldsymbol{\alpha}) \right], \quad t = 0, 1, ..., T-1, \tag{39}$$

$$V(T, \Theta) = l(\theta^{w^\star}; \mathcal{D}). \tag{40}$$

The loss function can be written as a linear quadratic function of $\Theta = [\theta_1^\top, \cdots, \theta_N^\top]^\top$ such that

$$\begin{aligned} & l(\theta^{w^\star}; \mathcal{D}) \\ &= \sum_{i=1}^N w_i^\star \sum_{j=1}^{|\mathcal{D}_i|} \left[ u_j^{i\top} \sum_{k=1}^N w_k^\star \theta_k - y_j^i \right]^2 \\ &= \sum_{i=1}^N w_i^\star \sum_{j=1}^{|\mathcal{D}_i|} \left[ \sum_{k=1}^N w_k^{\star 2} \theta_k^\top u_j^i u_j^{i\top} \theta_k - 2 \sum_{k=1}^N w_k^\star y_j^i u_j^{i\top} \theta_k + |y_j^i|^2 \right] \\ &= \left( \sum_{k=1}^N w_k^\star \theta_k^\top \right) \left( \sum_{i=1}^N w_i^\star \sum_{j=1}^{|\mathcal{D}_i|} u_j^i u_j^{i\top} \right) \left( \sum_{k=1}^N w_k^\star \theta_k \right) - 2 \sum_{i=1}^N w_i^\star \sum_{j=1}^{|\mathcal{D}_i|} y_j^i u_j^{i\top} \sum_{k=1}^N w_k^\star \theta_k + \sum_{i=1}^N w_i^\star \sum_{j=1}^{|\mathcal{D}_i|} |y_j^i|^2 \\ &= \Theta^\top [w_1^\star I_p, \cdots, w_N^\star I_p]^\top \left( \sum_{i=1}^N w_i^\star \sum_{j=1}^{|\mathcal{D}_i|} u_j^i u_j^{i\top} \right) [w_1^\star I_p, \cdots, w_N^\star I_p]\Theta \\ & \quad - 2 \sum_{i=1}^N w_i^\star \sum_{j=1}^{|\mathcal{D}_i|} y_j^i u_j^{i\top} [w_1^\star I_p, \cdots, w_N^\star I_p]\Theta + \sum_{i=1}^N w_i^\star \sum_{j=1}^{|\mathcal{D}_i|} |y_j^i|^2. \end{aligned} \tag{41}$$

We claim that $V(t,\Theta)$ takes the linear quadratic form

$$V(t,\Theta) = \Theta^\top P(t)\Theta + 2S^\top(t)\Theta + r(t). \tag{42}$$

To see this, note from (40) and (41) that $V(T,\Theta)$ takes the linear quadratic form of (42). By solving the dynamic programming equation (39) at $t = T-1$ for the optimal $\boldsymbol{\alpha}$ and substituting it back into the equation, we obtain $V(T-1, \Theta)$ also taking the linear quadratic form of (42). By backward induction we have that $V(t,\Theta)$ takes the form of (42) for all $t = 0, ..., T$.

**Remark 3.** *Here $r(t)$ is a constant term independent of the "state variable" $\Theta$, which we note exists because the cost $L^\star$ defined by (6) contains terms independent of $\Theta(t)$.*

Combining (40), (41), and (42) verifies that $P(T)$ and $S(T)$ satisfy the terminal condition as given by (36) and (37). We substitute (42) into (39) and reorganize the terms to get

$$\begin{aligned} & \Theta^\top P(t)\Theta + 2S^\top(t)\Theta + r(t) \\ &= \min_{\boldsymbol{\alpha}} \Big\{ \boldsymbol{\alpha}^\top [(\lambda + \beta)I_{Np} + P(t+1)]\boldsymbol{\alpha} + 2[\Theta^\top P(t+1) + S^\top(t+1) + \lambda(\Theta^\top - \Theta^{\star\top})]\boldsymbol{\alpha} \\ & \qquad + \Theta^\top (\lambda I_{Np} + P(t+1))\Theta + 2(S^\top(t+1) - \lambda\Theta^\star)\Theta + \lambda|\Theta^\star|^2 + r(t+1) \Big\}. \end{aligned} \tag{43}$$

By the first order condition[9] for the right-hand side of (43) with respect to $\boldsymbol{\alpha}$, we have

$$0 = [(\lambda + \beta)I_{Np} + P(t+1)]\boldsymbol{\alpha} + [(\lambda I_{Np} + P(t+1))\Theta + S(t+1) - \lambda\Theta^\star]. \tag{44}$$

---

[9]NB, we can argue this way, since the problem is convexified upon fixing $w^\star$. In fact, this is the control-theoretic motivation for using Algorithm 1 to decouple the optimization of $w$ and of $\theta$. from one another, and not treating them as a single (non-convex if left coupled) control problem.

By Assumption 2, we have $P(T) + (\lambda + 1)I_{Np} > 0$, and therefore from (44) we obtain the optimal $\boldsymbol{\alpha}$ taking the form of (34).

We further substitute $\boldsymbol{\alpha}$ of the form (34) into the right side of (43) to obtain

$$
\begin{aligned}
&\Theta^\top P(t)\Theta + 2S^\top(t)\Theta + r(t) \\
={}& - \left[\Theta^\top(\lambda I_{Np} + P(t+1)) + S^\top(t+1) - \lambda\Theta^{\star\top}\right] \cdot \left[(\lambda + \beta)I_{Np} + P(t+1)\right]^{-1} \cdot \\
&\left[(\lambda I_{Np} + P(t+1))\Theta + S(t+1) - \lambda\Theta^\star\right] + \Theta^\top(\lambda I_{Np} + P(t+1))\Theta \\
&+ 2[S^\top(t+1) - \lambda\Theta^{\star\top}]\Theta + \lambda|\Theta^\star|^2 + r(t+1).
\end{aligned}
\tag{45}
$$

Reorganizing the terms on right side of (45), we have

$$
\begin{aligned}
&\Theta^\top P(t)\Theta + 2S^\top(t)\Theta + r(t) \\
={}& - \Theta^\top(\lambda I_{Np} + P(t+1))[(\lambda + \beta)I_{Np} + P(t+1)]^{-1}(\lambda I_{Np} + P(t+1))\Theta \\
&+ \Theta^\top(\lambda I_{Np} + P(t+1))\Theta + 2[S^\top(t+1) - \lambda\Theta^{\star\top}]\Theta \\
&- 2[S^\top(t+1) - \lambda\Theta^{\star\top}][(\lambda + \beta)I_{Np} + P(t+1)]^{-1}(\lambda I_{Np} + P(t+1))\Theta \\
&- (S^\top(t+1) - \lambda\Theta^{\star\top})[(\lambda + \beta)I_{Np} + P(t+1)]^{-1}(S(t+1) - \lambda\Theta^\star) \\
&+ \lambda|\Theta^\star|^2 + r(t+1).
\end{aligned}
\tag{46}
$$

Matching the coefficients of the quadratic terms of $\Theta$ on both sides of (46), we obtain

$$
\begin{aligned}
P(t) ={}& - (\lambda I_{Np} + P(t+1))[(\lambda + \beta)I_{Np} + P(t+1)]^{-1}(\lambda I_{Np} + P(t+1)) \\
&+ \lambda I_{Np} + P(t+1).
\end{aligned}
\tag{47}
$$

The right side of (47) further simplifies to

$$
\begin{aligned}
&- (\lambda I_{Np} + P(t+1))[(\lambda + \beta)I_{Np} + P(t+1)]^{-1}(\lambda I_{Np} + P(t+1)) + \lambda I_{Np} + P(t+1) \\
={}& - [(\lambda + \beta)I_{Np} + P(t+1) - \beta I_{Np}][(\lambda + \beta)I_{Np} + P(t+1)]^{-1}(\lambda I_{Np} + P(t+1)) \\
&+ \lambda I_{Np} + P(t+1) \\
={}& - (\lambda I_{Np} + P(t+1)) + \beta[(\lambda + \beta)I_{Np} + P(t+1)]^{-1}(\lambda I_{Np} + P(t+1)) + \lambda I_{Np} + P(t+1) \\
={}& \beta[(\lambda + \beta)I_{Np} + P(t+1)]^{-1}(\lambda I_{Np} + P(t+1)) \\
={}& \beta[(\lambda + \beta)I_{Np} + P(t+1)]^{-1}[(\lambda + \beta)I_{Np} + P(t+1) - \beta I_{Np}] \\
={}& \beta I_{Np} - \beta^2[(\lambda + 1)I_{Np} + P(t+1)]^{-1}.
\end{aligned}
\tag{48}
$$

Combining (47) and (48) gives (36) for $t = T - 1$.

Matching the coefficients of the linear terms of $\Theta$ on both sides of (46) and taking the transpose, we obtain

$$
\begin{aligned}
S(t) ={}& - (\lambda I_{Np} + P(t+1))[(\lambda + \beta)I_{Np} + P(t+1)]^{-1}(S(t+1) - \lambda\Theta^\star) \\
&+ S(t+1) - \lambda\Theta^\star.
\end{aligned}
\tag{49}
$$

The right side of (49) further simplifies to

$$
\begin{aligned}
&- (\lambda I_{Np} + P(t+1))[(\lambda + \beta)I_{Np} + P(t+1)]^{-1}(S(t+1) - \lambda\Theta^\star) + S(t+1) - \lambda\Theta^\star \\
={}& - [(\lambda + \beta)I_{Np} + P(t+1) - \beta I_{Np}][(\lambda + \beta)I_{Np} + P(t+1)]^{-1}(S(t+1) - \lambda\Theta^\star) \\
&+ S(t+1) - \lambda\Theta^\star \\
={}& - (S(t+1) - \lambda\Theta^\star) + \beta[(\lambda + \beta)I_{Np} + P(t+1)]^{-1}(S(t+1) - \lambda\Theta^\star) + S(t+1) - \lambda\Theta^\star \\
={}& \beta[(\lambda + \beta)I_{Np} + P(t+1)]^{-1}(S(t+1) - \lambda\Theta^\star).
\end{aligned}
\tag{50}
$$

Combining (49) and (50) gives (37) for $t = T - 1$.

Matching the terms independent of $\Theta$ on both sides of (46) gives

$$r(t) = -\left(S(t+1) - \lambda\Theta^\star\right)^\top [(\lambda+\beta)I_{Np} + P(t+1)]^{-1} \cdot (S(t+1) - \lambda\Theta^\star)$$
$$+ \lambda|\Theta^\star|^2 + r(t+1),$$

which is (38) for $t = T - 1$.

Since $P(T) + (\lambda+\beta)I_{Np} > \beta I_{Np}$, we have

$$P(T-1) = \beta I_{Np} - \beta^2[(\lambda+\beta)I_{Np} + P(T)]^{-1} > 0.$$

By backward induction on $t$, we can show that $\boldsymbol{\alpha}$ takes the form of (34) for all $t = 0, 1, ..., T-1$, and $P$ and $S$ satisfy the system (36) and (37). $\qquad\square$

**Corollary 1.** *Under Assumptions 1 and 2, the solution $P$ and $S$ of the system (36)-(37) satisfies that for all $t = 1, 2, ..., T$,*

$$|[(\lambda+\beta)I_{Np} + P(t)]^{-1}| \leq (\lambda+\beta)^{-1}, \tag{51}$$
$$|S(t)| \leq C \cdot N, \tag{52}$$

*where $C > 0$ is a constant depending on $\lambda$, $\beta$, $(w_1^\star, ..., w_N^\star)$, $\Theta^\star$, $K_x$, $K_y$ and $T$.*

*Proof of Corollary 1.* Let $\|\cdot\|$ be the spectral norm of a matrix such that $\|A\| \overset{\text{def.}}{=} \sup_{x\neq 0} |Ax|/|x|$. Since $P(t)$ is positive semi-definite, we have $(\lambda+\beta)I_{Np} + P(t) \geq (\lambda+\beta)I_{Np}$ and

$$0 \leq [(\lambda+\beta)I_{Np} + P(t)]^{-1} \leq (\lambda+\beta)^{-1}I_{Np},$$

which, by the Courant-Fisher Theorem (see Horn & Johnson (2012)), implies that $\|[(\lambda+\beta)I_{Np} + P(t)]^{-1}\| \leq \|(\lambda+\beta)^{-1}I_{Np}\| = (\lambda+\beta)^{-1}$. Since all norms of finite dimensional normed spaces are equivalent, and in particular, $|\cdot| \leq \|\cdot\|$ on the space of $N_p \times N_p$-matrices, then we find that

$$\left|[(\lambda+\beta)I_{Np} + P(t)]^{-1}\right| \leq \|[(\lambda+\beta)I_{Np} + P(t)]^{-1}\|,$$

and thus $\left|[(\lambda+\beta)I_{Np} + P(t)]^{-1}\right| \leq (\lambda+\beta)^{-1}$, which proves (51).

Next, we prove (52). By Lemmas 1 and 2, and (17), we obtain

$$|S(T)|^2 = \left(\sum_{i=1}^N w_i^{\star 2}\right)\left|\sum_{i=1}^N w_i^\star \sum_{j=1}^{|\mathcal{D}_i|} u_j^i y_j^i\right|^2 \leq \left(\sum_{i=1}^N w_i^{\star 2}\right)\sum_{i=1}^N w_i^\star \left|\sum_{j=1}^{|\mathcal{D}_i|} u_j^i y_j^i\right|^2$$

$$\leq \left(\sum_{i=1}^N w_i^{\star 2}\right)\sum_{i=1}^N w_i^\star|\mathcal{D}_i|\sum_{j=1}^{|\mathcal{D}_i|}|u_j^i y_j^i|^2 \leq \left(\sum_{i=1}^N w_i^{\star 2}\right)\sum_{i=1}^N w_i^\star|\mathcal{D}_i|^2 K_x^2 K_y^2$$

$$\leq \left(\sum_{i=1}^N w_i^{\star 2}\right)\sum_{i=1}^N w_i^\star N^2 K_x^2 K_y^2, \tag{53}$$

and therefore the estimate (52) holds for $t = T$.

Assume by induction that (52) holds for $t = u$. By (37), Lemma 1, and the triangle inequality, we have

$$|S(u-1)| \leq \beta\left|[(\lambda+\beta)I_{Np} + P(u)]^{-1}\right| \cdot \left|S(u) - \lambda\Theta^\star\right|$$
$$\leq \beta\left|[(\lambda+\beta)I_{Np} + P(u)]^{-1}\right| \cdot (|S(u)| + \lambda|\Theta^\star|).$$

The estimate (52) for $t = u - 1$ then follows from (51) and the induction hypothesis. Thus, by induction we have shown that (51) holds for all $t = 1, 2, ..., T$. $\qquad\square$

We proceed to prove Theorem 3 about the complexity of the regret-optimal algorithm. To facilitate the proof, we introduce the following Lemma 3 that summarizes complexities of matrix operations.

**Lemma 3.** *Suppose all elementary arithmetic operations (e.g. addition, subtraction, multiplication) of two real numbers are of constant (i.e. $\mathcal{O}(1)$) complexity. Then elementary matrix computations have the following complexities:*

- *(i) Adding two $n_1 \times n_2$ matrices has a complexity of $O(n_1 n_2)$.*

- *(ii) Multiplying an $n_1 \times n_2$ matrices by a scalar has a complexity of $O(n_1 n_2)$.*

- *(iii) Multiplying an $n_1 \times n_2$ matrix with an $n_2 \times n_3$ matrix has a complexity of $O(n_1 n_2 n_3)$. In particular, multiplying two $n \times n$ matrices has a complexity of $\mathcal{O}(n^{2.373})$.*

*Proof.* Adding two $n_1 \times n_2$ matrices involves adding each entry of one matrix with the corresponding entry of the other matrix, and each entry addition has a complexity of $\mathcal{O}(1)$. Hence the total complexity is $\mathcal{O}(n_1 n_2)$.

Multiplying an $n_1 \times n_2$ matrix by a scalar involves multiplying each entry by the scalar, and the scalar multiplication of each entry has a complexity of $\mathcal{O}(1)$. Hence the total complexity is $O(n_1 n_2)$.

Multiplication of an $n_1 \times n_2$ matrix and an $n_2 \times n_3$ matrix requires multiplying the $n_1$ row vectors of the first matrix by the $n_3$ column vectors of the second matrix, which needs a total of $n_1 n_3$ multiplications. Multiplication of two $n_2$-dimensional vectors has a complexity of $O(n_2)$. Therefore the total complexity is $O(n_1 n_2 n_3)$. When multiplying two matrices of $n \times n$-dimension with an optimized CW-like algorithm the complexity is $\mathcal{O}(n^{2.373})$. $\square$

*Proof of Theorem 3.*

**Complexity of Computing $P(T)$ is $\mathcal{O}(N^2 p^3)$**

Since $u_j^i \in \mathbb{R}^{p \times 1}$, computing each $u_j^i u_j^{i\top}$ has $\mathcal{O}(p^2)$ complexity, by Lemma 3-(iii). Multiplying the $p \times p$ matrix $u_j^i u_j^{i\top}$ by the scalar $w_i^\star$ has a complexity of $\mathcal{O}(p^2)$, by Lemma 3-(iii). Then computing the $p \times p$ matrix $\sum_{i=1}^{N} w_i^\star \sum_{j=1}^{S_i} u_j^i u_j^{i\top}$ by adding the $\bar{N}$ matrices of dimension $p \times p$ has complexity $\mathcal{O}(\bar{N} p^2)$, by Lemma 3-(i). Since $[w_1^\star I_p, \cdots, w_N^\star I_p]^\top$ is $Np \times p$-dimensional, then computing the $np \times p$-dimensional product $[w_1^\star I_p, \cdots, w_N^\star I_p]^\top \sum_{i=1}^{N} w_i^\star \sum_{j=1}^{S_i} u_j^i u_j^{i\top}$ has $\mathcal{O}(Np^3)$ complexity, by Lemma 3-(iii). Likewise, the product

$$\left( [w_1^\star I_p, \cdots, w_N^\star I_p]^\top \sum_{i=1}^{N} w_i^\star \sum_{j=1}^{S_i} u_j^i u_j^{i\top} \right) [w_1^\star I_p, \cdots, w_N^\star I_p]$$

has a complexity of $\mathcal{O}(N^2 p^3)$. Therefore, computing $P(T)$ itself is of $\mathcal{O}(\bar{N} p^2 + N^2 p^3)$ complexity. Assumption (17) implies that $\bar{N} \leq N^2$; hence, the complexity of computing $P(T)$ is $\mathcal{O}(N^2 p^3)$.

**Complexity of Computing $S(T)$ is $\mathcal{O}(Np \max\{N, p\})$**

Since $u_j^i \in \mathbb{R}^{p \times 1}$ and $y_j^i \in \mathbb{R}$, computing the product $u_j^i y_j^i$ has complexity $\mathcal{O}(p)$, by Lemma 3-(iii). And multiplying the $p \times 1$ dimensional vector $u_j^i y_j^i$ by the scalar $w_i^\star$ has complexity $\mathcal{O}(p)$, by Lemma 3-(ii). Then the complexity of computing $\sum_{i=1}^{N} w_i^\star \sum_{j=1}^{S_i} u_j^i y_j^i$ by adding $\bar{N}$ vectors of dimension $p$ is $\mathcal{O}(\bar{N} p)$, by Lemma 3-(i). Since $-[w_1^\star I_p, \cdots, w_N^\star I_p]^\top$ is an $Np \times p$-dimensional matrix and $\sum_{i=1}^{N} w_i^\star \sum_{j=1}^{S_i} u_j^i y_j^i$ is a $p \times 1$-dimensional vector, then computing their $Np \times 1$ dimensional product has $\mathcal{O}(Np^2)$ complexity, according to Lemma 3-(iii). Therefore, the computing $S(T)$ has a complexity of $\mathcal{O}(\bar{N} p + NP^2)$. Since (17) implies that $\bar{N} \leq N^2$ then, computing $S(T)$ had a computational cost of $\mathcal{O}(Np \max\{N, p\})$.

**Complexity of Computing $P(t)$ and $S(t)$ for $t = T - 1, \ldots, 1$ is $\mathcal{O}(T(Np)^{2.373})$**

The complexity of computing $(\lambda + \beta)I_{Np} + P(t+1)$ for each $t$ is $\mathcal{O}(Np)$, since we only need to add scalars on the matrix's *diagonal*. By Le Gall (2014), computing the matrix inverse $[(\lambda + \beta)I_{Np} + P(t+1)]^{-1}$ with a CW-like algorithm has a complexity of $\mathcal{O}((Np)^{2.373})$. Since the addition $\beta I_{Np} - \beta^2 [(\lambda + \beta)I_{Np} + P(t+1)]^{-1}$ only requires us to multiply all entries of $[(\lambda + \beta)I_{Np} + P(t+1)]^{-1}$ by $-\beta^2$ and add along the diagonal, then it has a complexity of $\mathcal{O}((Np)^2 + Np)$, by Lemma 3-(i) and (ii). Performing the matrix addition

$S(t + 1) - \lambda\Theta^\star$ has complexity $\mathcal{O}(Np)$ by Lemma 3-(i). Computing the $Np \times 1$-dimensional product $[(\lambda + 1)I_{Np} + P(t+1)]^{-1}(S(t+1) - \lambda\Theta^\star)$ by multiplying an $Np \times Np$ matrix by an $Np \times 1$ matrix has complexity $\mathcal{O}(N^2p^2)$, according to Lemma 3-(iii). Therefore, computing each $P(t)$ and $S(t)$ for $t = T-1, \ldots, 1$, has complexity $\mathcal{O}((Np)^{2.373})$. Since there are $T-1$ such matrices to compute, then the total cost of computing every $P(T-1), \ldots, P(1)$ and $S(T-1), ..., S(1)$ is $\mathcal{O}(T(Np)^{2.373})$.

**The Cost of Computing Each $\Theta(t)$ is $\mathcal{O}(T N^2 p^2)$**

To compute $\Theta(t)$ for all $t = 1, \ldots, T$, starting from $\Theta(0) = \Theta^\star$, we need to compute $\mathbf{a}(t)$ given by (34) for $t = 0, 1, \ldots, T-1$. Hence we need to quantify the complexity of computing $\mathbf{a}(t)$. By (36), we can obtain $-[(\lambda + \beta)I_{Np} + P(t+1)]^{-1}$ by computing $\beta^{-2}(P(t) - \beta I_{Np})$, which involves adding a scalar to the diagonal of an $Np \times Np$ matrix, and then multiplying the matrix by a scalar. The cost of computing the product $(\lambda I_{Np} + P(t+1))\Theta(t)$ is $\mathcal{O}(N^2p^2)$ and the cost of computing all sums in $(\lambda I_{Np} + P(t+1))\Theta(t) - \lambda\Theta^\star + S(t+1)$ are of lower order complexity. Finally, the cost of computing the matrix-product between $-[(\lambda + \beta)I_{Np} + P(t+1)]^{-1}$ and $[(\lambda I_{Np} + P(t+1))\Theta(t) - \lambda\Theta^\star + S(t+1)]$ is $\mathcal{O}(N^2p^2)$, by Lemma 3-(iii). Therefore, the cost of computing each $\mathbf{a}(t)$ is $\mathcal{O}(N^2p^2)$, and therefore the cost of computing $\Theta(t)$ for all $t = 1, ..., t = T$ from $\Theta(0) = \Theta^\star$ is $\mathcal{O}(TN^2p^2)$.

**Tallying up Complexities: The Complexity of Computing $(\Theta(t))_{t=0}^T$ is $\mathcal{O}(N^2p^3 + T (Np)^{2.373})$.**

Looking over all computations involved, we deduce that the cost of computing the sequence $(\Theta(t))_{t=1}^T$ is (in order) dominated by the computational complexity of computing the sequence $(P(t))_{t=1}^{T-1}$ and $P(T)$; whence it is $\mathcal{O}(N^2p^3 + T (Np)^{2.373})$. $\qquad\square$

The following Lemma 4, Proposition 1, and Corollary 2 are devoted to proving Theorem 4, the adversarial robustness of Algorithm 2.

We introduce the linear map $M(\cdot) : \mathbb{R}^{Np \times Np} \to \mathbb{R}^{Np \times Np}$ such that $M(Q) = (\lambda + \beta)I_{Np} + Q$ for any $Q \in \mathbb{R}^{Np \times Np}$.

**Lemma 4.** *Given any two data sets $\mathcal{D} = \cup_{i=1}^N \mathcal{D}_i = \cup_{i=1}^N \cup_{j=1}^{|\mathcal{D}_i|} \{(x_j^i, y_j^i)\}$ and $\widetilde{\mathcal{D}} = \cup_{i=1}^N \widetilde{\mathcal{D}}_i = \cup_{i=1}^N \cup_{j=1}^{|\mathcal{D}_i|} \{(\tilde{x}_j^i, \tilde{y}_j^i)\}$ satisfying Assumption 1, let $(P, S)$ and $(\widetilde{P}, \widetilde{S})$ be the solutions of (36)-(37) corresponding to $\mathcal{D}$ and $\widetilde{\mathcal{D}}$, respectively. Then, for all $t = 1, ..., T$, we have*

$$\big|P(t) - \widetilde{P}(t)\big| \leq C \cdot \Big[\sum_{i=1}^N w_i^\star |\mathcal{D}_i| \sum_{j=1}^{|\mathcal{D}_i|} |u_j^i - \tilde{u}_j^i|^2\Big]^{1/2}, \tag{54}$$

$$\big|[M(P(t))]^{-1} - [M(\widetilde{P}(t))]^{-1}\big| \leq C \cdot \Big[\sum_{i=1}^N w_i^\star |\mathcal{D}_i| \sum_{j=1}^{|\mathcal{D}_i|} |u_j^i - \tilde{u}_j^i|^2\Big]^{1/2}, \tag{55}$$

$$|S(t) - \widetilde{S}(t)| \leq C \cdot \Big[\sum_{i=1}^N w_i^\star |\mathcal{D}_i| \sum_{j=1}^{|\mathcal{D}_i|} |u_j^i - \tilde{u}_j^i|^2 + |y_j^i - \tilde{y}_j^i|^2\Big]^{1/2}, \tag{56}$$

*where $C > 0$ is a constant depending on $\lambda$, $\beta$, $N$, $\bar{N}$, $(w_1^\star, ..., w_N^\star)$, $\Theta^\star$, $K_x$, $K_y$ and $T$.*

*Proof.* We let $C$ be a constant depending on $\lambda$, $\beta$, $N$, $\bar{N}$, $\Theta^\star$, $(w_1^\star, ..., w_N^\star)$, $K_x$, $K_y$, and $T$, and allow $C$ to vary from place to place throughout the proof. We employ backward induction starting from $t = T$ to prove (55), (54), and (56).

From (36), the terminal condition $|P(T) - \widetilde{P}(T)|^2$ can be written as

$$\big|P(T) - \widetilde{P}(T)\big|^2 = \Big|[w_1^\star I_p, \cdots, w_N^\star I_p]^\top \Big[\sum_{i=1}^N w_i^\star \sum_{j=1}^{|\mathcal{D}_i|} (u_j^i u_j^{i\top} - \tilde{u}_j^i \tilde{u}_j^{i\top})\Big][w_1^\star I_p, \cdots, w_N^\star I_p]\Big|^2$$

$$= \sum_{k,l=1}^N w_k^{\star 2} w_l^{\star 2} \Big|\sum_{i=1}^N w_i^\star \sum_{j=1}^{|\mathcal{D}_i|} (u_j^i u_j^{i\top} - \tilde{u}_j^i \tilde{u}_j^{i\top})\Big|^2.$$

By Lemmas 1 and 2, we have

$$
\begin{aligned}
\left|P(T) - \widetilde{P}(T)\right|^2 &\le \sum_{k,l=1}^{N} w_k^{\star 2} w_l^{\star 2} \sum_{i=1}^{N} w_i^\star \Big| \sum_{j=1}^{|\mathcal{D}_i|} (u_j^i u_j^{i\top} - \tilde{u}_j^i \tilde{u}_j^{i\top}) \Big|^2 \\
&\le \sum_{k,l=1}^{N} w_k^{\star 2} w_l^{\star 2} \sum_{i=1}^{N} w_i^\star |\mathcal{D}_i| \sum_{j=1}^{|\mathcal{D}_i|} \left| u_j^i u_j^{i\top} - \tilde{u}_j^i \tilde{u}_j^{i\top} \right|^2 \\
&= \sum_{k,l=1}^{N} w_k^{\star 2} w_l^{\star 2} \sum_{i=1}^{N} w_i^\star |\mathcal{D}_i| \sum_{j=1}^{|\mathcal{D}_i|} \left| (u_j^i - \tilde{u}_j^i) u_j^{i\top} + \tilde{u}_j^i (u_j^{i\top} - \tilde{u}_j^{i\top}) \right|^2 \\
&\le \sum_{k,l=1}^{N} w_k^{\star 2} w_l^{\star 2} \sum_{i=1}^{N} w_i^\star |\mathcal{D}_i| \sum_{j=1}^{|\mathcal{D}_i|} 2\left( \left| u_j^i - \tilde{u}_j^i \right|^2 \cdot \left| u_j^i \right|^2 + \left| \tilde{u}_j^i \right|^2 \cdot \left| u_j^i - \tilde{u}_j^i \right|^2 \right) \\
&\le \sum_{k,l=1}^{N} w_k^{\star 2} w_l^{\star 2} \sum_{i=1}^{N} w_i^\star |\mathcal{D}_i| \sum_{j=1}^{|\mathcal{D}_i|} 4 K_x^2 \left| u_j^i - \tilde{u}_j^i \right|^2. \quad (57)
\end{aligned}
$$

Therefore (54) holds for $t = T$.

Assume by induction that (54) holds for $t = u$. We show that (54) also holds for $t = u - 1$. From (36), we have

$$
\begin{aligned}
P(u-1) - \widetilde{P}(u-1) &= [M(\widetilde{P}(u))]^{-1} - [M(P(u))]^{-1} \\
&= [M(P(u))]^{-1} (P(u) - \widetilde{P}(u)) [M(\widetilde{P}(u))]^{-1},
\end{aligned}
$$

and by Lemma 1 we further have

$$
\left| P(u-1) - \widetilde{P}(u-1) \right| \le \left| [M(P(u))]^{-1} \right| \cdot \left| P(u) - \widetilde{P}(u) \right| \cdot \left| [M(\widetilde{P}(u))]^{-1} \right|. \quad (58)
$$

By Corollary 1 and the induction hypothesis for $t = u$, (58) implies that (54) holds for $t = u - 1$. We have shown by induction that (54) holds for all $t = 1, ..., T$.

Since

$$
[M(P(t))]^{-1} - [M(\widetilde{P}(t))]^{-1} = [M(P(t))]^{-1} (\widetilde{P}(t) - P(t)) [M(\widetilde{P}(t))]^{-1},
$$

by Lemma 1 we have

$$
\left| [M(P(t))]^{-1} - [M(\widetilde{P}(t))]^{-1} \right| \le \left| [M(P(t))]^{-1} \right| \cdot \left| \widetilde{P}(t) - P(t) \right| \cdot \left| [M(\widetilde{P}(t))]^{-1} \right|. \quad (59)
$$

The estimate (55) then follows from (59), (54), and Corollary 1.

Now we establish the estimate (56). By (37), the terminal condition $S(T) - \widetilde{S}(T)$ satisfies

$$
\begin{aligned}
|S(T) - \widetilde{S}(T)|^2 &= \Big| [w_1^\star, \cdots, w_N^\star]^\top \sum_{i=1}^{N} w_i^\star \sum_{j=1}^{|\mathcal{D}_i|} (u_j^i y_j^i - \tilde{u}_j^i \tilde{y}_j^i) \Big|^2 \\
&= \Big( \sum_{i=1}^{N} w_i^{\star 2} \Big) \Big| \sum_{i=1}^{N} w_i^\star \sum_{j=1}^{|\mathcal{D}_i|} (u_j^i y_j^i - \tilde{u}_j^i \tilde{y}_j^i) \Big|^2.
\end{aligned}
$$

By Lemmas 1 and 2, we obtain

$$
|S(T) - \widetilde{S}(T)|^2 \le \Big( \sum_{i=1}^{N} w_i^{\star 2} \Big) \sum_{i=1}^{N} w_i^\star \Big| \sum_{j=1}^{|\mathcal{D}_i|} (u_j^i y_j^i - \tilde{u}_j^i \tilde{y}_j^i) \Big|^2
$$

$$\leq \Big( \sum_{i=1}^{N} w_i^{\star 2} \Big) \sum_{i=1}^{N} w_i^{\star} |\mathcal{D}_i| \sum_{j=1}^{|\mathcal{D}_i|} \big| u_j^i y_j^i - \tilde{u}_j^i \tilde{y}_j^i \big|^2$$

$$= \Big( \sum_{i=1}^{N} w_i^{\star 2} \Big) \sum_{i=1}^{N} w_i^{\star} |\mathcal{D}_i| \sum_{j=1}^{|\mathcal{D}_i|} \big| (u_j^i - \tilde{u}_j^i) y_j^i + \tilde{u}_j^i (y_j^i - \tilde{y}_j^i) \big|^2$$

$$\leq \Big( \sum_{i=1}^{N} w_i^{\star 2} \Big) \sum_{i=1}^{N} w_i^{\star} |\mathcal{D}_i| \cdot 2 \sum_{j=1}^{|\mathcal{D}_i|} \big[ \big| (u_j^i - \tilde{u}_j^i) y_j^i \big|^2 + \big| \tilde{u}_j^i (y_j^i - \tilde{y}_j^i) \big|^2 \big]$$

$$\leq \Big( \sum_{i=1}^{N} w_i^{\star 2} \Big) \sum_{i=1}^{N} w_i^{\star} |\mathcal{D}_i| \cdot 2 \sum_{j=1}^{|\mathcal{D}_i|} \big[ |u_j^i - \tilde{u}_j^i|^2 \cdot |y_j^i|^2 + |\tilde{u}_j^i|^2 \cdot |y_j^i - \tilde{y}_j^i|^2 \big]$$

$$\leq \Big( \sum_{i=1}^{N} w_i^{\star 2} \Big) \sum_{i=1}^{N} w_i^{\star} |\mathcal{D}_i| \cdot 2 \sum_{j=1}^{|\mathcal{D}_i|} \big[ K_y^2 \cdot |u_j^i - \tilde{u}_j^i|^2 + K_x^2 \cdot |y_j^i - \tilde{y}_j^i|^2 \big]$$

$$\leq \Big( \sum_{i=1}^{N} w_i^{\star 2} \Big) 2(K_x^2 + K_y^2) \sum_{i=1}^{N} w_i^{\star} |\mathcal{D}_i| \sum_{j=1}^{|\mathcal{D}_i|} \big[ |u_j^i - \tilde{u}_j^i|^2 + |y_j^i - \tilde{y}_j^i|^2 \big], \tag{60}$$

and thus (56) holds for $t = T$. Assume by induction that (56) holds for $t = u$, we show that (56) also holds for $t = u - 1$. From (37), the difference $S(t) - \widetilde{S}(t)$ can be written as

$$S(u - 1) - \widetilde{S}(u - 1) = [M(\widetilde{P}(u))]^{-1} [S(u) - \widetilde{S}(u)] + \big[ (M(P(u)))^{-1} - (M(\widetilde{P}(u)))^{-1} \big] (S(u) - \lambda \Theta^{\star}),$$

and satisfies by Lemma 1 and Corollary 1 that

$$|S(u - 1) - \widetilde{S}(u - 1)| \leq \big| [M(\widetilde{P}(u))]^{-1} \big| \cdot |S(u) - \widetilde{S}(u)| + \big| (M(P(u)))^{-1} - (M(\widetilde{P}(u)))^{-1} \big| \cdot (|S(u)| + \lambda |\Theta^{\star}|). \tag{61}$$

By Corollary 1, (55), and the induction hypothesis, (61) implies that (56) holds for $t = u - 1$. We have shown by induction that (56) holds for all $t = 1, ..., T$. $\qquad\square$

**Proposition 1.** *For two arbitrary data sets* $\mathcal{D} = \cup_{i=1}^{N} \mathcal{D}_i = \cup_{i=1}^{N} \cup_{j=1}^{|\mathcal{D}_i|} \{(x_j^i, y_j^i)\}$ *and* $\widetilde{\mathcal{D}} = \cup_{i=1}^{N} \widetilde{\mathcal{D}}_i = \cup_{i=1}^{N} \cup_{j=1}^{|\mathcal{D}_i|} \{(\tilde{x}_j^i, \tilde{y}_j^i)\}$ *satisfying Assumption 1, let* $\boldsymbol{\alpha} = (\alpha_1^{\top}, \cdots, \alpha_N^{\top})^{\top}$ *and* $\widetilde{\boldsymbol{\alpha}} = (\widetilde{\alpha}_1^{\top}, \cdots, \widetilde{\alpha}_N^{\top})^{\top}$ *be the optimal controls for* (6) *and* (9) *corresponding to* $\mathcal{D}$ *and* $\widetilde{\mathcal{D}}$, *respectively. Then, we have*

$$\sum_{t=0}^{T-1} |\boldsymbol{\alpha}(t) - \widetilde{\boldsymbol{\alpha}}(t)| \leq C \Big[ \sum_{i=1}^{N} w_i^{\star 2} |\mathcal{D}_i| \sum_{j=1}^{|\mathcal{D}_i|} |u_j^i - \tilde{u}_j^i|^2 + |y_j^i - \tilde{y}_j^i|^2 \Big]^{1/2}, \tag{62}$$

*where* $C > 0$ *is a constant depending on* $\lambda$, $\beta$, $N$, $\bar{N}$, $\Theta^{\star}$, $(w_1^{\star}, \cdots, w_N^{\star})$, $K_x$, $K_y$, *and* $T$.

*Proof of Proposition 1.* Let $C$ be a constant depending on $\lambda$, $\beta$, $N$, $\bar{N}$, $\Theta^{\star}$, $(w_1^{\star}, \cdots, w_N^{\star})$, $K_x$, $K_y$, and $T$, and $C$ is allowed to vary from place to place throughout the proof. By (34) and (10), the optimal control $\boldsymbol{\alpha}(t)$ along the optimal trajectory can be written as

$$\boldsymbol{\alpha}(t) = [(M(P(t+1)))^{-1} - I_{Np}] \Big[ \Theta^{\star} + \sum_{u=0}^{t-1} \boldsymbol{\alpha}(u) \Big] + [M(P(t+1))]^{-1} (\lambda \Theta^{\star} - S(t+1)),$$

and the difference $\boldsymbol{\alpha}(t) - \widetilde{\boldsymbol{\alpha}}(t)$ can be written as

$$\boldsymbol{\alpha}(t) - \widetilde{\boldsymbol{\alpha}}(t) = \big[ (M(P(t+1)))^{-1} - (M(\widetilde{P}(t+1)))^{-1} \big] \Big[ \Theta^{\star} + \sum_{u=0}^{t-1} \boldsymbol{\alpha}(u) + \lambda \Theta^{\star} - S(t+1) \Big]$$

$$+ \left[ (M(\widetilde{P}(t+1)))^{-1} - I_{Np} \right] \sum_{u=0}^{t-1} (\boldsymbol{\alpha}(u) - \widetilde{\boldsymbol{\alpha}}(u))$$

$$+ (M(\widetilde{P}(t+1)))^{-1}(S(t+1) - \widetilde{S}(t+1)). \tag{63}$$

For $t = 0$, the difference $\boldsymbol{\alpha}(0) - \widetilde{\boldsymbol{\alpha}}(0)$ can be written as

$$\boldsymbol{\alpha}(0) - \widetilde{\boldsymbol{\alpha}}(0) = \left[ (M(P(1)))^{-1} - (M(\widetilde{P}(1)))^{-1} \right] \left[ (\lambda + 1)\Theta^\star - S(1) \right]$$
$$- [M(\widetilde{P}(1))]^{-1}(S(1) - \widetilde{S}(1)),$$

and satisfies by Lemma 1 that

$$\left| \boldsymbol{\alpha}(0) - \widetilde{\boldsymbol{\alpha}}(0) \right| \leq \left| (M(P(1)))^{-1} - (M(\widetilde{P}(1)))^{-1} \right| \cdot \left[ (\lambda + 1)|\Theta^\star| + |S(1)| \right]$$
$$+ \left| [M(\widetilde{P}(1))]^{-1} \right| \cdot \left| S(1) - \widetilde{S}(1) \right|.$$

It then follows from Corollary 1, (55), and (56) that

$$\left| \boldsymbol{\alpha}(0) - \widetilde{\boldsymbol{\alpha}}(0) \right| \leq C \cdot \left[ \sum_{i=1}^{N} w_i^\star |\mathcal{D}_i| \sum_{j=1}^{|\mathcal{D}_i|} |u_j^i - \tilde{u}_j^i|^2 + |y_j^i - \tilde{y}_j^i|^2 \right]^{1/2}.$$

Assume by induction that

$$\sum_{u=0}^{t-1} \left| \boldsymbol{\alpha}(u) - \widetilde{\boldsymbol{\alpha}}(u) \right| \leq C \cdot \left[ \sum_{i=1}^{N} w_i^\star |\mathcal{D}_i| \sum_{j=1}^{|\mathcal{D}_i|} |u_j^i - \tilde{u}_j^i|^2 + |y_j^i - \tilde{y}_j^i|^2 \right]^{1/2}.$$

By Lemma 1, (63) implies that

$$\left| \boldsymbol{\alpha}(t) - \widetilde{\boldsymbol{\alpha}}(t) \right|$$

$$\leq \left| (M(P(t+1)))^{-1} - (M(\widetilde{P}(t+1)))^{-1} \right| \cdot \left[ |\Theta^\star| + \sum_{u=0}^{t-1} |\boldsymbol{\alpha}(u)| + \lambda|\Theta^\star| + |S(t+1)| \right]$$

$$+ \left| (M(\widetilde{P}(t+1)))^{-1} - I_{Np} \right| \sum_{u=0}^{t-1} |\boldsymbol{\alpha}(u) - \widetilde{\boldsymbol{\alpha}}(u)| + \left| (M(\widetilde{P}(t+1)))^{-1} \right| \cdot |S(t+1) - \widetilde{S}(t+1)|.$$

By (55), (56), and the induction hypothesis, the above inequality implies that that

$$\left| \boldsymbol{\alpha}(t) - \widetilde{\boldsymbol{\alpha}}(t) \right| \leq C \left[ \sum_{i=1}^{N} w_i^\star |\mathcal{D}_i| \sum_{j=1}^{|\mathcal{D}_i|} |u_j^i - \tilde{u}_j^i|^2 + |y_j^i - \tilde{y}_j^i|^2 \right]^{1/2},$$

and therefore

$$\sum_{u=0}^{t} \left| \boldsymbol{\alpha}(u) - \widetilde{\boldsymbol{\alpha}}(u) \right| \leq C \cdot \left[ \sum_{i=1}^{N} w_i^\star |\mathcal{D}_i| \sum_{j=1}^{|\mathcal{D}_i|} |u_j^i - \tilde{u}_j^i|^2 + |y_j^i - \tilde{y}_j^i|^2 \right]^{1/2}.$$

The estimate (62) is then proved by induction. $\qquad \square$

**Corollary 2.** *Under the hypothesis of Proposition 1, the costs (6) under the optimal controls* $\mathbf{a}$ *and* $\widetilde{\mathbf{a}}$ *satisfy the estimate*

$$\left| L^\star(\boldsymbol{\alpha}; \mathcal{D}) - L^\star(\widetilde{\boldsymbol{\alpha}}; \widetilde{\mathcal{D}}) \right| \leq C \cdot \left[ \sum_{i=1}^{N} w_i^\star |\mathcal{D}_i| \sum_{j=1}^{|\mathcal{D}_i|} |u_j^i - \tilde{x}_j^i|^2 + |y_j^i - \tilde{y}_j^i|^2 \right]^{1/2}. \tag{64}$$

*where* $C > 0$ *is a constant depending on* $\lambda$, $\beta$, $N$, $\bar{N}$, $\Theta^\star$, $(w_1^\star, \cdots, w_N^\star)$, $K_x$, $K_y$, *and* $T$.

*Proof of Corollary 2.* Throughout the proof $C$ is a constant depending on $\lambda$, $\beta$, $\bar{N}$, $\Theta^\star$, $(w_1^\star, \cdots, w_N^\star)$, $K_x$, $K_y$ and $T$, and may vary from place to place. By (42) and (9), the difference of the costs can be written as

$$L^\star(\boldsymbol{\alpha}; \mathcal{D}) - L^\star(\widetilde{\boldsymbol{\alpha}}; \widetilde{\mathcal{D}}) = \Theta^\top(0)(P(0) - \widetilde{P}(0))\Theta(0) + 2(S(0) - \widetilde{S}(0))^\top \Theta(0) + r(0) - \widetilde{r}(0),$$
$$= \Theta^{\star\top}(P(0) - \widetilde{P}(0))\Theta^\star + 2(S(0) - \widetilde{S}(0))^\top \Theta^\star + r(0) - \widetilde{r}(0).$$

By the triangular inequality, the difference satisfies

$$\left| L^\star(\boldsymbol{\alpha}; \mathcal{D}) - L^\star(\widetilde{\boldsymbol{\alpha}}; \widetilde{\mathcal{D}}) \right| \le |P(0) - \widetilde{P}(0)| \cdot |\Theta^\star|^2 + 2|S(0) - \widetilde{S}(0)| \cdot |\Theta^\star| + |r(0) - \widetilde{r}(0)|. \tag{65}$$

We claim that $r(0) - \widetilde{r}(0)$ satisfies the estimate

$$|r(0) - \widetilde{r}(0)| \le C \cdot \Big[ \sum_{i=1}^{N} w_i^\star |\mathcal{D}_i| \sum_{j=1}^{|\mathcal{D}_i|} |u_j^i - \tilde{u}_j^i|^2 + |y_j^i - \tilde{y}_j^i|^2 \Big]^{1/2}. \tag{66}$$

Then the desired estimate (64) is established by (65), (54), (56) and (66).

Now we prove the claim (66). From (38), the difference $r - \widetilde{r}$ satisfies

$$\begin{aligned} r(t) - \widetilde{r}(t) = {} & r(t+1) - \widetilde{r}(t+1) - (S(t+1) - \widetilde{S}(t+1))\big\{ (M(P(t+1)))^{-1}\big[ S(t+1) - 2\lambda\Theta^\star \big] \\ & + (M(\widetilde{P}(t+1)))^{-1}\widetilde{S}(t+1) \big\} \\ & - \widetilde{S}^\top(t+1)\big[ (M(P(t+1)))^{-1} - (M(\widetilde{P}(t+1)))^{-1} \big]\big[ S(t+1) - 2\lambda\Theta^\star \big] \\ & - \Theta^{\star\top}\lambda\big[ (M(P(t+1)))^{-1} - (M(\widetilde{P}(t+1)))^{-1} \big]\lambda\Theta^\star, \end{aligned}$$
$$r(T) - \widetilde{r}(T) = \sum_{i=1}^{N} w_i^\star \sum_{j=1}^{|\mathcal{D}_i|} (|y_j^i|^2 - |\widetilde{y}_j^i|^2). \tag{67}$$

Inductively, $r(0) - \widetilde{r}(0)$ can be written in terms of $r(T) - \widetilde{r}(T)$ as

$$\begin{aligned} & r(0) - \widetilde{r}(0) \\ = {} & r(T) - \widetilde{r}(T) - \sum_{t=1}^{T} \Big\{ (S(t) - \widetilde{S}(t))\big[ (M(P(t)))^{-1}\big( S(t) - 2\lambda\Theta^\star \big) + (M(\widetilde{P}(t)))^{-1}\widetilde{S}(t) \big] \\ & + \widetilde{S}^\top(t)\big[ (M(P(t)))^{-1} - (M(\widetilde{P}(t)))^{-1} \big]\big[ S(t) - 2\lambda\Theta^\star \big] + \Theta^{\star\top}\lambda\big[ (M(P(t)))^{-1} - (M(\widetilde{P}(t)))^{-1} \big]\lambda\Theta^\star \Big\}. \end{aligned}$$

By Lemma 1, we have

$$\begin{aligned} & |r(0) - \widetilde{r}(0)| \\ \le {} & |r(T) - \widetilde{r}(T)| + \sum_{t=1}^{T} \Big\{ |S(t) - \widetilde{S}(t)| \cdot \Big[ \big|(M(P(t)))^{-1}\big| \cdot \big( |S(t)| + 2\lambda|\Theta^\star| \big) + \big|(M(\widetilde{P}(t)))^{-1}\big| \cdot |\widetilde{S}(t)| \Big] \\ & + \big|(M(P(t)))^{-1} - (M(\widetilde{P}(t)))^{-1}\big| \cdot \big( |\widetilde{S}(t)| \cdot |S(t) - 2\lambda\Theta^\star| + \lambda|\Theta^\star|^2 \big) \Big\}. \end{aligned} \tag{68}$$

By (67), Lemma 2, and the Cauchy-Schwarz inequality, the terminal condition satisfies

$$\begin{aligned} |r(T) - \widetilde{r}(T)| = {} & \Big| \sum_{i=1}^{N} w_i^\star \sum_{j=1}^{|\mathcal{D}_i|} (y_j^i - \widetilde{y}_j^i)(y_j^i + \widetilde{y}_j^i) \Big| \le \sum_{i=1}^{N} w_i^\star \sum_{j=1}^{|\mathcal{D}_i|} |(y_j^i - \widetilde{y}_j^i)(y_j^i + \widetilde{y}_j^i)| \\ \le {} & \sum_{i=1}^{N} w_i^\star \sum_{j=1}^{|\mathcal{D}_i|} |y_j^i + \widetilde{y}_j^i| \cdot |y_j^i - \widetilde{y}_j^i| \le 2K_y \sum_{i=1}^{N} w_i^\star \sum_{j=1}^{|\mathcal{D}_i|} |y_j^i - \widetilde{y}_j^i| \\ \le {} & 2K_y \Big( \sum_{i=1}^{N} w_i^\star \sum_{j=1}^{|\mathcal{D}_i|} |y_j^i - \widetilde{y}_j^i|^2 \Big)^{1/2}. \end{aligned} \tag{69}$$

The estimate (66) then follows from (68), (69), (55), (56), and (52). □

We may now derive the proof of Theorem 4 as a direct consequence of Corollary 2.

*Proof of Theorem 4.* Fix $\widetilde{\mathcal{D}} \in \mathbb{D}^{p,\varepsilon}$ and let $I \subseteq \{(i,j) : i = 1, \ldots, N, j = 1, \ldots, |\mathcal{D}_i|\}$ consist of all indices $(i,j)$ for which $\{(i,j) : u_j^i \neq \widetilde{u}_j^i$ and $y_j^i \neq \widetilde{y}_j^i\}$. By Corollary 2, there is a constant $C > 0$ which is independent of $\mathcal{D}$ such that

$$
\begin{aligned}
\left|L^\star(\boldsymbol{\alpha}; \mathcal{D}) - L^\star(\widetilde{\boldsymbol{\alpha}}; \widetilde{\mathcal{D}})\right|^2 \leq & \widetilde{C} \sum_{i=1}^N w_i^\star |\mathcal{D}_i| \sum_{j=1}^{|\mathcal{D}_i|} |u_j^i - \widetilde{u}_j^i|^2 + |y_j^i - \widetilde{y}_j^i|^2 \\
\leq & \widetilde{C} \cdot \sum_{(i,j) \in I} w_i^\star |\mathcal{D}_i| (|u_j^i - \widetilde{u}_j^i|^2 + |y_j^i - \widetilde{y}_j^i|^2) \\
\leq & \widetilde{C} \cdot \sum_{(i,j) \in I} w_i^\star |\mathcal{D}_i| \cdot \max\{|u_j^i - \widetilde{u}_j^i|^2, |y_j^i - \widetilde{y}_j^i|^2\} \\
= & \widetilde{C} \cdot \sum_{(i,j) \in I} w_i^\star |\mathcal{D}_i| \cdot \max\{|u_j^i - \widetilde{u}_j^i|, |y_j^i - \widetilde{y}_j^i|\}^2 \\
\leq & \widetilde{C} \cdot \sum_{(i,j) \in I} w_i^\star |\mathcal{D}_i| \cdot 2\varepsilon^2 \\
\leq & \widetilde{C} \# I\, p\, 2\varepsilon^2,
\end{aligned}
$$

where we have set $\widetilde{C} \overset{\text{def.}}{=} C^{1/2}$. $\qquad\square$

The computational complexity of the matrix-valued function $P(\cdot)$, as determined by (36), does not primarily stem from the high dimensionality of the output matrices, but rather from their inherent lack of symmetries. As we will see later, such symmetry will allow us to efficiently encode $P(\cdot)$ into low-dimensional structures, therefore completely mitigating the high-dimensionality effect.

We therefore introduce the following control, which is optimal for a "symmetrized version" of the control problem (34) where $w^\star = (1/N, \ldots, 1/N)$. This advantages this surrogate problem is that the associated $P$ function, which we denote by $\widehat{P}$, outputs highly symmetric[10] Toeplitz matrices which are determined by exactly three factors. This allows us to encode the entire control problem into a control problem on $\mathbb{R}^{\tilde{O}(P)}$ whose dimension is independent of $N$. Furthermore, the optimal controls of both problems can be related by stability estimates, depending only on the deviation of $(1/N, \ldots, 1/N)$ from $w^\star$, which allows us to infer the degree of sub-optimality of our $\widehat{P}$. Therefore, we consider the control

$$
\widehat{\boldsymbol{\alpha}}(t) = -\left[(\lambda + \beta)I_{Np} + \widehat{P}(t+1)\right]^{-1}\left[(\lambda I_{Np} + \widehat{P}(t+1))\Theta(t) - \lambda \Theta_{(N)}^\star + \widehat{S}(t+1)\right]. \tag{70}
$$

We will show that the algorithm under control (70) enjoys lower computational complexity in comparison with the regret-optimal algorithm (34), whose computational complexity is summarized through Theorem 3. Furthermore, we will then establish the near-regret optimality of (70), stated in Theorem 5.

The associated matrix-valued functions $\widehat{P}$ and $\widehat{S}$ are uniquely determined by

$$
\begin{cases}
\widehat{P}(t) = \beta I_{Np} - \beta^2[(\lambda + \beta)I_{Np} + \widehat{P}(t+1)]^{-1}, \\
\widehat{P}(T) = [(1/N)I_p, \cdots, (1/N)I_p]^\top \left(\sum_{i=1}^N (1/N)\sum_{j=1}^{|\mathcal{D}_i|} u_j^i u_j^{i\top}\right)[(1/N)I_p, \cdots, (1/N)I_p],
\end{cases} \tag{71}
$$

$$
\begin{cases}
\widehat{S}(t) = \beta[(\lambda + \beta)I_{Np} + \widehat{P}(t+1)]^{-1}\left(\widehat{S}(t+1) - \lambda \Theta_{(N)}^\star\right), \\
\widehat{S}(T) = -[(1/N)I_p, \cdots, (1/N)I_p]^\top \sum_{i=1}^N (1/N)\sum_{j=1}^{|\mathcal{D}_i|} u_j^i y_j^i,
\end{cases} \tag{72}
$$

where $[I_p, \cdots, I_p]$ denotes the $Np \times p$-dimensional matrix formed by concatenating $N$ copies of the $p \times p$-dimensional identity matrix $\frac{1}{N}I_p$. The parameter $\Theta_{(N)}^\star$ is defined as

$$
\Theta_{(N)}^\star \overset{\text{def.}}{=} [\theta_{(N)}^{\star\top}, \cdots, \theta_{(N)}^{\star\top}]^\top, \quad \theta_{(N)}^\star \overset{\text{def.}}{=} \frac{1}{N}\sum_{i=1}^N \theta_i^\star.
$$

---

[10]As formalized in Proposition 4.

**Remark 4.** *According to* (7) *in Assumption* 2, $\widehat{P}(T) \geq 0$ *and therefore* (71) *admits a unique solution.*

**Remark 5.** *Similar to Corollary* 1, *we have* $\left|[(\lambda + \beta)I_{Np} + \widehat{P}(t)]^{-1}\right| \leq (\lambda + \beta)^{-1}$. *By replacing* $(w_1^\star, ..., w_N^\star)$ *in the proof of Corollary* 1 *with* $(1/N, ..., 1/N)$, *we obtain* $|\widehat{S}(t)| \leq C \cdot N^{1/2}$ *for all* $t = 1, ..., T$, *where* $C > 0$ *is a constant depending on* $\lambda$, $\beta$, $\Theta^\star$, $K_x$, $K_y$ *and* $T$.

**Lemma 5.** *Let* $P$ *and* $\widehat{P}$ *be the solutions of the systems* (36) *and* (71), *respectively. Then they satisfy that for all* $t = 1, 2, ..., T$,

$$\left|P(t) - \widehat{P}(t)\right| \leq C \cdot \bar{N} N \max_{1 \leq i,k,l \leq N} \left|\frac{1}{N^3} - w_k^\star w_l^\star w_i^\star\right|, \tag{73}$$

$$\left|(M(P(t)))^{-1} - (M(\widehat{P}(t)))^{-1}\right| \leq C \cdot \bar{N} N \max_{1 \leq i,k,l \leq N} \left|\frac{1}{N^3} - w_k^\star w_l^\star w_i^\star\right|, \tag{74}$$

*where* $C > 0$ *is a constant depending on* $\lambda$, $\beta$, $K_x$.

*Proof of Lemma 5.* By (71), (36), and Lemmas 1 and 2, the difference $\widehat{P}(T) - P(T)$ satisfies

$$|\widehat{P}(T) - P(T)|^2 = \sum_{k,l=1}^{N} \left|\frac{1}{N^2} \sum_{k=1}^{N} \frac{1}{N} \sum_{j=1}^{|\mathcal{D}_i|} u_j^i u_j^{i\top} - w_k^\star w_l^\star \sum_{i=1}^{N} w_i^\star \sum_{j=1}^{|\mathcal{D}_i|} u_j^i u_j^{i\top}\right|^2$$

$$\leq N^2 \cdot \max_{1 \leq k,l,i \leq N} \left|\frac{1}{N^3} - w_k^\star w_l^\star w_i^\star\right|^2 \cdot \left|\sum_{i=1}^{N} \sum_{j=1}^{|\mathcal{D}_i|} u_j^i u_j^{i\top}\right|^2$$

$$\leq N^2 \cdot \max_{1 \leq k,l,i \leq N} \left|\frac{1}{N^3} - w_k^\star w_l^\star w_i^\star\right|^2 \cdot \bar{N} \sum_{i=1}^{N} \sum_{j=1}^{|\mathcal{D}_i|} |u_j^i|^4$$

and

$$|\widehat{P}(T) - P(T)| \leq \bar{N} N K_x^2 \cdot \max_{1 \leq k,l,i \leq N} \left|\frac{1}{N^3} - w_k^\star w_l^\star w_i^\star\right|. \tag{75}$$

Therefore (73) holds for $t = T$.

Assume by induction that (73) holds for $t = u$. We show that (73) also holds for $t = u - 1$. From (71) and (36), we have

$$\widehat{P}(u-1) - P(u-1) = [M(P(u))]^{-1} - [M(\widehat{P}(u))]^{-1}$$
$$= [M(P(u))]^{-1}(\widehat{P}(u) - P(u))[M(\widehat{P}(u))]^{-1}. \tag{76}$$

By Lemmas 1 and 2, we further have

$$|\widehat{P}(u-1) - P(u-1)| = |[M(P(u))]^{-1} - [M(\widehat{P}(u))]^{-1}|$$
$$\leq |[M(P(u))]^{-1}| \cdot |\widehat{P}(u) - P(u)| \cdot |[M(\widehat{P}(u))]^{-1}|. \tag{77}$$

By (77), Corollary 1, Remark 5, and the induction hypothesis, we have that (73) holds for $t = u - 1$. By induction we have shown that (73) holds for all $1 \leq t \leq T$.

From (76), we have

$$\left|(M(P(t)))^{-1} - (M(\widehat{P}(t)))^{-1}\right| \leq \left|(M(P(t)))^{-1}\right| \cdot \left|\widehat{P}(t) - P(t)\right| \cdot \left|(M(\widehat{P}(t)))^{-1}\right|,$$

which, together with (73), Corollary 1, and Remark 5, establishes (74). $\square$

**Lemma 6.** *Let* $S$ *and* $\widehat{S}$ *be the solutions of* (37) *and* (72), *respectively. Then they satisfy that for all* $t = 1, ..., T$,

$$|S(t) - \widehat{S}(t)| \leq C \cdot \left\{\bar{N} N^{3/2} \max_{1 \leq i,k,l \leq N} \left|\frac{1}{N^3} - w_k^\star w_l^\star w_i^\star\right|\right.$$

$$+ \bar{N}N^{1/2} \max_{1 \leq k,l \leq N} \left| \frac{1}{N^2} - w_k^\star w_l^\star \right| + |\Theta^\star - \Theta_{(N)}^\star| \Big\}, \tag{78}$$

where $C > 0$ is a constant depending on $\lambda$, $\beta$, $\Theta^\star$, $K_x$, $K_y$, and $T$.

*Proof of Lemma 6.* By (37), (72), and and Lemmas 1 and 2, the terminal condition $S(T) - \widehat{S}(T)$ satisfies

$$\left| S(T) - \widehat{S}(T) \right|^2 = \sum_{k=1}^{N} \left| \frac{1}{N} \sum_{i=1}^{N} \frac{1}{N} \sum_{j=1}^{|\mathcal{D}_i|} u_j^i y_j^i - w_k^\star \sum_{i=1}^{N} w_i^\star \sum_{j=1}^{|\mathcal{D}_i|} u_j^i y_j^i \right|^2$$

$$\leq N \cdot \max_{1 \leq k,l \leq N} \left| \frac{1}{N^2} - w_k^\star w_l^\star \right|^2 \cdot \left| \sum_{i=1}^{N} \sum_{j=1}^{|\mathcal{D}_i|} u_j^i y_j^i \right|^2$$

$$\leq N \cdot \max_{1 \leq k,l \leq N} \left| \frac{1}{N^2} - w_k^\star w_l^\star \right|^2 \cdot \bar{N} \sum_{i=1}^{N} \sum_{j=1}^{|\mathcal{D}_i|} |u_j^i|^2 \cdot |y_j^i|^2,$$

which implies

$$\left| S(T) - \widehat{S}(T) \right| \leq \bar{N}N^{1/2} \max_{1 \leq k,l \leq N} \left| \frac{1}{N^2} - w_k^\star w_l^\star \right| \cdot K_x K_y. \tag{79}$$

Therefore (78) holds for $t = T$.

Assume by induction that (78) holds for $t = u$, we show that (78) holds for $t = u - 1$. From (37) and (72), the difference $S(u-1) - \widehat{S}(u-1)$ can be written as

$$S(u-1) - \widehat{S}(u-1) = [M(P(u))]^{-1}[S(u) - \widehat{S}(u) - \lambda(\Theta^\star - \Theta_{(N)}^\star)]$$
$$- \left[ (M(\widehat{P}(u)))^{-1} - (M(P(u)))^{-1} \right] (\widehat{S}(u) - \lambda\Theta_{(N)}^\star), \tag{80}$$

and satisfies by Lemma 1 that

$$|S(u-1) - \widehat{S}(u-1)| \leq |[M(P(u))]^{-1}| \cdot \left[ |S(u) - \widehat{S}(u)| + \lambda|\Theta^\star - \Theta_{(N)}^\star| \right]$$
$$+ \left| (M(\widehat{P}(u)))^{-1} - (M(P(u)))^{-1} \right| \cdot \left( |\widehat{S}(u)| + \lambda|\Theta_{(N)}^\star| \right). \tag{81}$$

By (81), (74), Remark 5, and the induction hypothesis, we have that (78) holds for $t = u - 1$. By induction we have shown that (78) holds for all $t = 1, ..., T$.

$$\square$$

We now obtain Theorem 5; which is a direct consequence of the following Propositions 2 and 3.

**Proposition 2.** *The controls (34) and (70) satisfy $|\widehat{\boldsymbol{\alpha}}(t)| \leq C \cdot N^{1/2}$ for all $t = 0, 1, ..., T-1$, and*

$$\sum_{t=0}^{T-1} |\boldsymbol{\alpha}(t) - \widehat{\boldsymbol{\alpha}}(t)| \leq C \cdot \Big\{ \bar{N}N^{3/2} \max_{1 \leq i,k,l \leq N} \left| \frac{1}{N^3} - w_k^\star w_l^\star w_i^\star \right|$$

$$+ \bar{N}N^{1/2} \cdot \max_{1 \leq k,l \leq N} \left| \frac{1}{N^2} - w_k^\star w_l^\star \right| + |\Theta^\star - \Theta_{(N)}^\star| \Big\}, \tag{82}$$

where $C > 0$ is a constant depending on $\lambda$, $\beta$, $\Theta^\star$, $K_x$, $K_y$, and $T$.

*Proof of Proposition 2.* We let $C$ be a constant depending on $\lambda$, $\beta$, $\Theta^\star$, $K_x$, $K_y$, and $T$, and $C$ is allowed to vary from place to place throughout the proof. By (34), (70), and (10), we can write $\boldsymbol{\alpha}$ and $\widehat{\boldsymbol{\alpha}}$ as

$$\boldsymbol{\alpha}(t) = [(M(P(t+1)))^{-1} - I_{Np}] \Big[ \Theta^\star + \sum_{u=0}^{t-1} \boldsymbol{\alpha}(u) \Big] + [M(P(t+1))]^{-1}(\lambda\Theta^\star - S(t+1)),$$

$$\widehat{\boldsymbol{\alpha}}(t) = [(M(\widehat{P}(t+1)))^{-1} - I_{Np}] \Big[\Theta^\star + \sum_{u=0}^{t-1} \widehat{\boldsymbol{\alpha}}(u)\Big] + [M(\widehat{P}(t+1))]^{-1}(\lambda\Theta^\star_{(N)} - \widehat{S}(t+1)).$$

For $t = 0$, we have

$$|\widehat{\boldsymbol{\alpha}}(0)| \le |(M(\widehat{P}(1)))^{-1} - I_{Np}| \cdot |\Theta^\star| + \big|[M(\widehat{P}(1))]^{-1}\big|(\lambda|\Theta^\star_{(N)}| + |\widehat{S}(1)|),$$

which implies $|\widehat{\boldsymbol{\alpha}}(0)| \le CN^{1/2}$. Assume by induction that $|\widehat{\boldsymbol{\alpha}}(u)| \le CN^{1/2}$ for $u = 0, ..., t-1$, then

$$\widehat{\boldsymbol{\alpha}}(t) \le \big|(M(\widehat{P}(t+1)))^{-1} - I_{Np}\big|\Big[|\Theta^\star| + \sum_{u=0}^{t-1} |\widehat{\boldsymbol{\alpha}}(u)|\Big]$$
$$+ \big|[M(\widehat{P}(t+1))]^{-1}\big|(\lambda|\Theta^\star_{(N)}| + |\widehat{S}(t+1)|)$$

gives that $|\widehat{\boldsymbol{\alpha}}(t)| \le CN^{1/2}$. By induction we have shown that $|\widehat{\boldsymbol{\alpha}}(t)| \le CN^{1/2}$ for all $t = 0, 1, ..., T-1$. We further write the difference $\boldsymbol{\alpha} - \widehat{\boldsymbol{\alpha}}$ as

$$\boldsymbol{\alpha}(t) - \widehat{\boldsymbol{\alpha}}(t)$$
$$= \big[(M(P(t+1)))^{-1} - (M(\widehat{P}(t+1)))^{-1}\big]\Big[(\lambda+1)\Theta^\star + \sum_{u=0}^{t-1} \boldsymbol{\alpha}(u) - S(t+1)\Big]$$
$$+ \big[(M(\widehat{P}(t+1)))^{-1} - I_{Np}\big]\sum_{u=0}^{t-1}(\boldsymbol{\alpha}(u) - \widehat{\boldsymbol{\alpha}}(u)) + [M(\widehat{P}(t+1))]^{-1}(\widehat{S}(t+1) - S(t+1))$$
$$+ \lambda[M(\widehat{P}(t+1))]^{-1}(\Theta^\star - \Theta^\star_{(N)}). \tag{83}$$

For $t = 0$, the difference

$$\boldsymbol{\alpha}(0) - \widehat{\boldsymbol{\alpha}}(0) = \big[(M(P(1)))^{-1} - (M(\widehat{P}(1)))^{-1}\big]\big[(\lambda+1)\Theta^\star_{(N)} - \widehat{S}(1)\big]$$
$$+ [M(P(1))]^{-1}[(\lambda+1)(\Theta^\star - \Theta^\star_{(N)}) + \widehat{S}(1) - S(1)]$$

satisfies

$$\big|\boldsymbol{\alpha}(0) - \widehat{\boldsymbol{\alpha}}(0)\big| \le C \cdot \Big\{\bar{N}N^{3/2} \max_{1\le i,k,l\le N}\Big|\frac{1}{N^3} - w_k^\star w_l^\star w_i^\star\Big|$$
$$+ \bar{N}N^{1/2} \max_{1\le k,l\le N}\Big|\frac{1}{N^2} - w_k^\star w_l^\star\Big| + |\Theta^\star - \Theta^\star_{(N)}|\Big\},$$

due to (74), (78), Lemma 1, Corollary 1 and Remark 5. Assume by induction for all $0 \le u \le t-1$, $\boldsymbol{\alpha}(u-1) - \widehat{\boldsymbol{\alpha}}(u-1)$ satisfies

$$\big|\boldsymbol{\alpha}(u-1) - \widehat{\boldsymbol{\alpha}}(u-1)\big| \le C \cdot \Big\{\bar{N}N^{3/2} \max_{1\le i,k,l\le N}\Big|\frac{1}{N^3} - w_k^\star w_l^\star w_i^\star\Big|$$
$$+ \bar{N}N^{1/2} \max_{1\le k,l\le N}\Big|\frac{1}{N^2} - w_k^\star w_l^\star\Big| + |\Theta^\star - \Theta^\star_{(N)}|\Big\}.$$

By (83) we have

$$|\boldsymbol{\alpha}(t) - \widehat{\boldsymbol{\alpha}}(t)|$$
$$\le \big|(M(P(t+1)))^{-1} - (M(\widehat{P}(t+1)))^{-1}\big| \cdot \Big[\sum_{u=0}^{t-1}|\boldsymbol{\alpha}(u)| + |(\lambda+1)\Theta^\star| + |S(t+1)|\Big]$$
$$+ \big|(M(\widehat{P}(t+1)))^{-1} - I_{Np}\big|\sum_{u=0}^{t-1}|\boldsymbol{\alpha}(u) - \widehat{\boldsymbol{\alpha}}(u)| + \big|(M(\widehat{P}(t+1)))^{-1}\big| \cdot |S(t+1) - \widehat{S}(t+1)|,$$

$$+ \left|[M(\widehat{P}(t+1))]^{-1}\right| \cdot |\lambda| \cdot \left|\Theta^\star - \Theta^\star_{(N)}\right|.$$

By the induction hypothesis, (74), (78), and Remark 5, the above inequality implies that

$$
\begin{aligned}
|\boldsymbol{\alpha}(t) - \widehat{\boldsymbol{\alpha}}(t)| \leq & C \cdot \Big\{ \bar{N} N^{3/2} \max_{1 \leq i,k,l \leq N} \left| \frac{1}{N^3} - w_k^\star w_l^\star w_i^\star \right| \\
& + \bar{N} N^{1/2} \max_{1 \leq k,l \leq N} \left| \frac{1}{N^2} - w_k^\star w_l^\star \right| + |\Theta^\star - \Theta^\star_{(N)}| \Big\}.
\end{aligned}
\tag{84}
$$

We have shown by induction that (84) holds for all $t = 0, 1, ..., T-1$, and therefore (82) follows. $\qquad\square$

**Proposition 3.** *Under the hypothesis of Proposition 2, the costs* (6) *under the controls* (34) *and* (70), *respectively, satisfy the estimate*

$$
\begin{aligned}
\left|L^\star(\boldsymbol{\alpha}; \mathcal{D}) - L^\star(\widehat{\boldsymbol{\alpha}}; \mathcal{D})\right| \leq & C \cdot \Big\{ \bar{N} N^3 \max_{1 \leq i,k,l \leq N} \left| \frac{1}{N^3} - w_k^\star w_l^\star w_i^\star \right| + \bar{N} N^{3/2} \max_{1 \leq k,l \leq N} \left| \frac{1}{N^2} - w_k^\star w_l^\star \right| \\
& + \bar{N} \max_{1 \leq i \leq N} \left| \frac{1}{N} - w_i^\star \right| + N|\Theta^\star - \Theta^\star_{(N)}| \Big\},
\end{aligned}
\tag{85}
$$

*where $C > 0$ is a constant depending on $\lambda$, $\beta$, $\Theta^\star$, $(w_1^\star, ..., w_N^\star)$, $K_x$, $K_y$ and $T$.*

*Proof of Proposition 3 .* Let $L^\star(\widehat{\boldsymbol{\alpha}}; \overline{1/N}, \mathcal{D})$, $L^\star(\widehat{\boldsymbol{\alpha}}; w^\star, \mathcal{D})$, and $L^\star(\boldsymbol{\alpha}; w^\star, \mathcal{D})$ be as defined in Lemmas 7 and 8. The difference $L^\star(\boldsymbol{\alpha}; \mathcal{D}) - L^\star(\widehat{\boldsymbol{\alpha}}; \mathcal{D})$ can be decomposed as

$$L^\star(\boldsymbol{\alpha}; \mathcal{D}) - L^\star(\widehat{\boldsymbol{\alpha}}; \mathcal{D}) = L^\star(\boldsymbol{\alpha}; w^\star, \mathcal{D}) - L^\star(\widehat{\boldsymbol{\alpha}}; \overline{1/N}, \mathcal{D}) + L^\star(\widehat{\boldsymbol{\alpha}}; \overline{1/N}, \mathcal{D}) - L^\star(\widehat{\boldsymbol{\alpha}}; w^\star, \mathcal{D}),$$

and satisfies

$$|L^\star(\boldsymbol{\alpha}; \mathcal{D}) - L^\star(\widehat{\boldsymbol{\alpha}}; \mathcal{D})| \leq |L^\star(\boldsymbol{\alpha}; w^\star, \mathcal{D}) - L^\star(\widehat{\boldsymbol{\alpha}}; \overline{1/N}, \mathcal{D})| + |L^\star(\widehat{\boldsymbol{\alpha}}; \overline{1/N}, \mathcal{D}) - L^\star(\widehat{\boldsymbol{\alpha}}; w^\star, \mathcal{D})|.$$

The desired estimate (85) then follows from the above inequality and (86) and (88). $\qquad\square$

**Lemma 7.** *Let $L^\star(\widehat{\boldsymbol{\alpha}}; w^\star, \mathcal{D})$ be* (6) *with information sharing weights $w^\star = (w_1^\star, ..., w_N^\star)$ and $\Theta(t+1) - \Theta(t) = \widehat{\boldsymbol{\alpha}}(t)$, and let $L^\star(\widehat{\boldsymbol{\alpha}}; \overline{1/N}, \mathcal{D})$ be* (6) *with information sharing weights $\overline{1/N} = (1/N, ..., 1/N)$ and $\Theta(t+1) - \Theta(t) = \widehat{\boldsymbol{\alpha}}(t)$. Then it satisfies*

$$|L^\star(\widehat{\boldsymbol{\alpha}}; \overline{1/N}, \mathcal{D}) - L^\star(\widehat{\boldsymbol{\alpha}}; w^\star, \mathcal{D})| \leq C \cdot \Big\{ \bar{N} \max_{1 \leq i \leq N} \left| \frac{1}{N} - w_i^\star \right| + N^{1/2} |\Theta^\star - \Theta^\star_{(N)}| \Big\},
\tag{86}$$

*where $C > 0$ is a constant depending on $\lambda$, $\beta$, $\Theta^\star$, $(w_1^\star, \cdots, w_N^\star)$, $K_x$, $K_y$ and $T$.*

*Proof.* By (6) and (41), we have

$$
\begin{aligned}
L^\star(\widehat{\boldsymbol{\alpha}}; \overline{1/N}, \mathcal{D}) - L^\star(\widehat{\boldsymbol{\alpha}}; w^\star, \mathcal{D}) = & \sum_{t=0}^{T-1} \left( |\Theta(t+1) - \Theta^*|^2 - |\Theta(t+1) - \Theta^*_{(N)}|^2 \right) \\
& + l(\Theta(T), \overline{1/N}; \mathcal{D}) - l(\Theta(T), w^\star; \mathcal{D}).
\end{aligned}
$$

We have that

$$
\begin{aligned}
& \left| \sum_{t=0}^{T-1} \left( |\Theta(t+1) - \Theta^*|^2 - |\Theta(t+1) - \Theta^*_{(N)}|^2 \right) \right| \\
= & \left| \sum_{t=0}^{T-1} (\Theta^\star_{(N)} - \Theta^\star)^\top (2\Theta(t+1) - \Theta^\star - \Theta^\star_{(N)}) \right| \\
\leq & \sum_{t=0}^{T-1} |\Theta^\star_{(N)} - \Theta^\star| \cdot |2\Theta(t+1) - \Theta^\star - \Theta^\star_{(N)}| \leq C N^{1/2} \cdot |\Theta^\star_{(N)} - \Theta^\star|,
\end{aligned}
$$

and

$$l(\Theta(T), \overline{1/N}; \mathcal{D}) - l(\Theta(T), w^\star; \mathcal{D})$$

$$= \sum_{i=1}^{N} w_i^\star \sum_{j=1}^{|\mathcal{D}_i|} \left[ u_j^{i\top} \sum_{k=1}^{N} w_k^\star \theta_k - y_j^i \right]^2 - \sum_{i=1}^{N} \frac{1}{N} \sum_{j=1}^{|\mathcal{D}_i|} \left[ u_j^{i\top} \sum_{k=1}^{N} \frac{1}{N} \theta_k - y_j^i \right]^2$$

$$= \sum_{i=1}^{N} (w_i^\star - 1/N) \sum_{j=1}^{|\mathcal{D}_i|} \left[ u_j^{i\top} \sum_{k=1}^{N} w_k^\star \theta_k - y_j^i \right]^2$$

$$+ \sum_{i=1}^{N} \frac{1}{N} \sum_{j=1}^{|\mathcal{D}_i|} \left[ u_j^{i\top} \sum_{k=1}^{N} (w_k^\star - 1/N)\theta_k \right] \left[ u_j^{i\top} \sum_{k=1}^{N} (w_k^\star + 1/N)\theta_k - 2y_j^i \right]^2,$$

which implies that

$$|l(\Theta(T), \overline{1/N}; \mathcal{D}) - l(\Theta(T), w^\star; \mathcal{D})| \leq C\bar{N} \max_{1 \leq i \leq N} |1/N - w_i^\star|. \tag{87}$$

The estimate (86) then follows from (87) and (87). $\qquad\square$

**Lemma 8.** *Let $L^\star(\alpha; w^\star, \mathcal{D})$ be (6) with information sharing weights $w^\star = (w_1^\star, ..., w_N^\star)$ and $\Theta(t+1) - \Theta(t) = \alpha(t)$, and let $L^\star(\widehat{\alpha}; \overline{1/N}, \mathcal{D})$ be (6) with information sharing weights $\overline{1/N} = (1/N, ..., 1/N)$ and $\Theta(t+1) - \Theta(t) = \widehat{\alpha}(t)$. Then it satisfies*

$$|L^\star(\alpha; w^\star, \mathcal{D}) - L^\star(\widehat{\alpha}; \overline{1/N}, \mathcal{D})| \leq C \cdot \left\{ \bar{N} N^3 \cdot \max_{1 \leq i,k,l \leq N} \left| \frac{1}{N^3} - w_k^\star w_l^\star w_i^\star \right| \right.$$

$$\left. + \bar{N} N^{3/2} \max_{1 \leq k,l \leq N} \left| \frac{1}{N^2} - w_k^\star w_l^\star \right| + \bar{N} \cdot \max_{1 \leq i \leq N} \left| \frac{1}{N} - w_i^\star \right| + N|\Theta^\star - \Theta_{(N)}^\star| \right\}, \tag{88}$$

*where $C$ is a constant depending on $\lambda$, $\beta$, $\Theta^\star$, $(w_1^\star, \cdots, w_N^\star)$, $K_x$, $K_y$ and $T$.*

*Proof.* By the similar argument for (42), we have

$$L^\star(\widehat{\alpha}; \overline{1/N}, \mathcal{D}) = \Theta^{\star\top} \widehat{P}(0)\Theta^\star + 2\widehat{S}(0)^\top \Theta^\star + \widehat{r}(0). \tag{89}$$

By (42) and (89), the difference $L^\star(\alpha; w^\star, \mathcal{D}) - L^\star(\widehat{\alpha}; \overline{1/N}, \mathcal{D})$ can be written as

$$L^\star(\alpha; w^\star, \mathcal{D}) - L^\star(\widehat{\alpha}; \overline{1/N}, \mathcal{D}) = \Theta^{\star\top} (P(0) - \widehat{P}(0))\Theta^\star + 2(S(0) - \widehat{S}(0))^\top \Theta^\star + r(0) - \widehat{r}(0),$$

and by Lemma 1, the difference satisfies

$$\left| L^\star(\alpha; w^\star, \mathcal{D}) - L^\star(\widehat{\alpha}; \overline{1/N}, \mathcal{D}) \right| \leq |\Theta^\star|^2 \cdot |P(0) - \widehat{P}(0)| + 2|S(0) - \widehat{S}(0)| \cdot |\Theta^\star|$$

$$+ |r(0) - \widehat{r}(0)|. \tag{90}$$

We claim that $r(0) - \widehat{r}(0)$ has the estimate

$$|r(0) - \widehat{r}(0)| \leq C \cdot \left\{ \bar{N} N^3 \cdot \max_{1 \leq i,k,l \leq N} \left| \frac{1}{N^3} - w_k^\star w_l^\star w_i^\star \right| + \bar{N} N^{3/2} \max_{1 \leq k,l \leq N} \left| \frac{1}{N^2} - w_k^\star w_l^\star \right| \right.$$

$$\left. + \bar{N} \cdot \max_{1 \leq i \leq N} \left| \frac{1}{N} - w_i^\star \right| + N \cdot |\Theta^\star - \Theta_{(N)}^\star| \right\}, \tag{91}$$

where $C$ is a constant depending on $\lambda$, $\beta$, $\Theta^\star$, $(w_1^\star, \cdots, w_N^\star)$, $K_x$, $K_y$ and $T$. Then the desired estimate (88) is established by (90), (73), (78) and (91).

Now we prove the claim (91). From (45) and (41), we obtain the equation that $\widehat{r}$ satisfies on $0 \leq t \leq T$:

$$\begin{cases} \widehat{r}(t) = -(\widehat{S}(t+1) - \lambda\Theta_{(N)}^\star)^\top [(\lambda + \beta)I_{Np} + \widehat{P}(t+1)]^{-1} \cdot (\widehat{S}(t+1) - \lambda\Theta_{(N)}^\star) \\ \qquad + \lambda\|\Theta_{(N)}^\star\|_2^2 + \widehat{r}(t+1), \\ \widehat{r}(T) = \sum_{i=1}^{N} (1/N) \sum_{j=1}^{|\mathcal{D}_i|} |y_j^i|^2. \end{cases} \tag{92}$$

Then $r - \widehat{r}$ satisfies the equation on $0 \leq t \leq T$

$$
\begin{aligned}
& r(t) - \widehat{r}(t) \\
={}& r(t+1) - \widehat{r}(t+1) + \lambda(\Theta^\star - \Theta^\star_{(N)})(\Theta^\star + \Theta^\star_{(N)}) \\
& - (S(t+1) - \lambda\Theta^\star)^\top \big[(M(P(t+1)))^{-1} - (M(\widehat{P}(t+1)))^{-1}\big](S(t+1) - \lambda\Theta^\star) \\
& - \big[S(t+1) - \widehat{S}(t+1) - \lambda(\Theta^\star - \Theta^\star_{(N)})\big](M(\widehat{P}(t+1)))^{-1} \cdot \\
& \quad \big[S(t+1) + \widehat{S}(t+1) - \lambda(\Theta^\star + \Theta^\star_{(N)})\big],
\end{aligned}
$$

$$
r(T) - \widehat{r}(T) = \sum_{i=1}^{N}(w_i^\star - 1/N)\sum_{j=1}^{|\mathcal{D}_i|}|y_j^i|^2. \tag{93}
$$

Inductively, we have

$$
\begin{aligned}
r(0) - \widehat{r}(0) ={}& r(T) - \widehat{r}(T) + T\lambda(\Theta^\star - \Theta^\star_{(N)})(\Theta^\star + \Theta^\star_{(N)}) \\
& - \sum_{t=1}^{T}(S(t) - \lambda\Theta^\star)^\top\big[(M(P(t)))^{-1} - (M(\widehat{P}(t)))^{-1}\big](S(t) - \lambda\Theta^\star) \\
& - \sum_{t=1}^{T}\big[S(t) - \widehat{S}(t) - \lambda(\Theta^\star - \Theta^\star_{(N)})\big](M(\widehat{P}(t)))^{-1}\big[S(t) + \widehat{S}(t) - \lambda(\Theta^\star + \Theta^\star_{(N)})\big].
\end{aligned}
$$

By Lemma 1, we have

$$
\begin{aligned}
|r(0) - \widehat{r}(0)| \leq{}& |r(T) - \widehat{r}(T)| + T\lambda|\Theta^\star - \Theta^\star_{(N)}| \cdot |\Theta^\star + \Theta^\star_{(N)}| \\
& + \sum_{t=1}^{T}\big|(M(P(t)))^{-1} - (M(\widehat{P}(t)))^{-1}\big| \cdot \big|S(t) - \lambda\Theta^\star\big|^2 \\
& + \sum_{t=1}^{T}\big[|S(t) - \widehat{S}(t)| + \lambda|\Theta^\star - \Theta^\star_{(N)}|\big] \cdot \big|(M(\widehat{P}(t)))^{-1}\big| \cdot \big|S(t) + \widehat{S}(t) - \lambda(\Theta^\star + \Theta^\star_{(N)})\big|. \tag{94}
\end{aligned}
$$

Then the estimate (91) follows from (94), (74), (78), and Remark 5. $\qquad\square$

*Proof of Theorem 5.* Under Assumptions 1 and 2, the existence of the regret-optimal algorithm $\theta$. is implied by Theorem 8. The first claim follows from Proposition 2 upon noting that the constant "$C_w$" therein is bounded as a function of $w$; whence we take the constant $C$ hidden by $\mathcal{O}$ to be $C \stackrel{\text{def.}}{=} \sup_{w \in [0,1]^N \sum_{i=1}^{N} w_i = 1} C_w$. Likewise, the second statement follows directly from Corollary 2 upon defining the constant hidden by $\mathcal{O}$ as before. $\qquad\square$

Further investigation of the "symmetrized system" (71) reveals a symmetric form of $\widehat{P}$, which suggests that $\widehat{P}$ can be decomposed into homogeneous submatrices of lower dimensions. Specifically, if we decompose the high dimensional $Np \times Np$ matrix $\widehat{P}$ into $N \times N$ submatrices of dimensions $p \times p$ each, and either exchange two submatrices symmetrically positioned with respect to the diagonal of $\widehat{P}$ or exchange two submatrices on the diagonal, the system (71) is invariant. This suggests that all the submatrices on the diagonal are homogeneous and all the off-diagonal submatrices are homogeneous.

**Proposition 4** (Symmetries in $\widehat{P}$). *The solution of* (71) *has the following submatrix decomposition*

$$
\widehat{P} = \begin{bmatrix} \pi_1 & \pi_2 & \cdots & \pi_2 \\ \pi_2 & \pi_1 & \cdots & \pi_2 \\ \vdots & \vdots & \ddots & \vdots \\ \pi_2 & \pi_2 & \cdots & \pi_1 \end{bmatrix} \in \mathbb{R}^{Np \times Np}, \quad \pi_1 \in \mathbb{R}^{p \times p}, \quad \pi_2 \in \mathbb{R}^{p \times p}, \tag{95}
$$

*and the submatrices $\pi_1$ and $\pi_2$ are uniquely determined by the system*

$$\begin{cases} \pi_1(t) = \beta I_p - \beta^2 \gamma_1(t+1), \\ \pi_2(t) = -\beta^2 \gamma_2(t+1), \quad 1 \le t \le T-1, \\ \pi_1(T) = \pi_2(T) = (1/N^3) \sum_{i=1}^{N} \sum_{j=1}^{|\mathcal{D}_i|} u_j^i u_j^{i\top}, \end{cases} \tag{96}$$

*where $\gamma_1(\cdot)$ and $\gamma_2(\cdot)$ are defined as* (18) *and* (19).

We argue as in (Huang & Yang, 2021, Lemmata 1) in deriving the following submatrix decomposition of $\widehat{P}$.

*Proof of Proposition 4.* We decompose the $Np \times Np$ identity matrix into $N \times N$ submatrices $I = (I_{ij})_{1 \le i,j \le N}$ with each $I_{ij} \in \mathbb{R}^{p \times p}$. For each $1 \le i, j \le N$, by $E_{ij}$ we denote the $Np \times Np$ elementary matrix by exchanging the $i$th and $j$th rows of submatrices of the $Np \times Np$ identity matrix $I = (I_{ij})_{1 \le i,j \le N}$. Note that $E_{ij} = E_{ij}^{-1} = E_{ij}^T$. Multiplying both sides of (36) from the left by $E_{ij}$ and from the right by $E_{ij}$, we obtain

$$\begin{cases} E_{ij}\widehat{P}(t)E_{ij} = \beta I_{Np} - \beta^2 [(\lambda + \beta)I_{Np} + E_{ij}\widehat{P}(t+1)E_{ij}]^{-1}, \\ E_{ij}\widehat{P}(T)E_{ij} = [(1/N)I_p, \cdots, (1/N)I_p]^\top \left( \sum_{i=1}^{N} \frac{1}{N} \sum_{j=1}^{|\mathcal{D}_i|} u_j^i u_j^{i\top} \right) [(1/N)I_p, \cdots, (1/N)I_p]. \end{cases} \tag{97}$$

Comparing (97) and (71) reveals that $\widehat{P}(t)$ and $E_{ij}\widehat{P}(t)E_{ij}$ satisfy the same ODE for all $1 \le i, j \le N$. This observation implies that

$$\widehat{P}_{ii} = \widehat{P}_{jj}, \quad \widehat{P}_{ij} = \widehat{P}_{ji}, \quad \widehat{P}_{ik} = \widehat{P}_{jk}, \quad \widehat{P}_{ki} = \widehat{P}_{kj}, \quad \forall 1 \le i, j \le N, \quad \forall k \ne i, j,$$

and the first assertion of the proposition is proved.

Since $\widehat{P}$ takes the form (95), it is straight forward to verify that $[(\lambda + \beta)I_{Np} + \widehat{P}]^{-1}$ takes the form

$$[(\lambda + \beta)I_{Np} + \widehat{P}]^{-1} = \begin{bmatrix} \gamma_1 & \gamma_2 & \cdots & \gamma_2 \\ \gamma_2 & \gamma_1 & \cdots & \gamma_2 \\ \vdots & \vdots & \ddots & \vdots \\ \gamma_2 & \gamma_2 & \cdots & \gamma_1 \end{bmatrix}, \tag{98}$$

with $\gamma_1$ and $\gamma_2$ given by (18) and (19).

We substitute (95) and (98) into (71) to obtain the equations (96). $\qquad\square$

Upon substituting $\widehat{P}$ into (72), we find that the $Np \times 1$ matrix $\widehat{S}$ can be decomposed into homogeneous submatrices of dimensions $p \times 1$ each.

**Proposition 5.** *The solution of* (72) *can be decomposed into submatrices*

$$\widehat{S} = (\pi_3^\top, \cdots, \pi_3^\top)^\top \in \mathbb{R}^{Np \times 1}, \quad \pi_3 \in \mathbb{R}^{p \times 1}, \tag{99}$$

*and $\pi_3$ satisfies*

$$\begin{cases} \pi_3(t) = \beta [\gamma_1(t+1) + (N-1)\gamma_2(t+1)](\pi_3(t+1) - \lambda \theta_{(N)}^\star), \\ \pi_3(T) = -(1/N^2) \sum_{i=1}^{N} \sum_{j=1}^{|\mathcal{D}_i|} u_j^i y_j^i. \end{cases} \tag{100}$$

*Proof of Proposition 5.* We multiply both sides of (72) from the left by $E_{ij}$ defined in the proof of Proposition 4,

$$\begin{aligned} E_{ij}\widehat{S}(t) &= E_{ij}\beta[(\lambda + \beta)I_{Np} + \widehat{P}(t+1)]^{-1}(\widehat{S}(t+1) - \lambda\Theta_{(N)}^\star) \\ &= E_{ij}\beta[(\lambda + \beta)I_{Np} + \widehat{P}(t+1)]^{-1}E_{ij}E_{ij}(\widehat{S}(t+1) - \lambda\Theta_{(N)}^\star) \\ &= \beta[(\lambda + \beta)I_{Np} + E_{ij}\widehat{P}(t+1)E_{ij}]^{-1}(E_{ij}\widehat{S}(t+1) - \lambda\Theta_{(N)}^\star) \end{aligned} \tag{101}$$

and

$$E_{ij}\widehat{S}(T) = - E_{ij}[(1/N)I_p, \cdots, (1/N)I_p]^\top \sum_{i=1}^{N} \frac{1}{N} \sum_{j=1}^{|\mathcal{D}_i|} u_j^i y_j^i$$

$$= - [(1/N)I_p, \cdots, (1/N)I_p]^\top \sum_{i=1}^{N} \frac{1}{N} \sum_{j=1}^{|\mathcal{D}_i|} u_j^i y_j^i. \tag{102}$$

From (101) and (101), we have that $E_{ij}\widehat{S}$ also satisfies (72) for all $1 \leq i, j \leq N$. This proves that all submatrices of $\widehat{S} = (\widehat{S}_i)_{1 \leq i \leq N}$ are equal on $0 \leq t \leq T$. We further substitute (95), (98) and $\widehat{S} = [\pi_3^\top, \cdots, \pi_3^\top]^\top$ into (72) to obtain (100). □

**Corollary 3.** *Fix a data set $\mathcal{D}$ and assume that (17) holds. Then the computational complexity of computing the sequence $(\theta(t))_{t=0}^{T}$ by (70) is $\mathcal{O}(Tp^2 \max\{N, p^{0.373}\})$.*

*Proof of Corollary 3.*
**Complexity of computing $\pi_1(T)$, $\pi_2(T)$, and $\pi_3(T)$ is $O(N^2 p^2)$**
Computing $u_j^i u_j^{i\top}$ has complexity $O(p^2)$ by Lemma 3-(iii) and computing the sum $\sum_{i=1}^{N} \sum_{j=1}^{|\mathcal{D}_i|} u_j^i u_j^{i\top}$ has a complexity of $O(N^2 p^2)$ by Lemma 3-(i). Moreover, multiplying the $p \times p$ matrix $\sum_{i=1}^{N} \sum_{j=1}^{|\mathcal{D}_i|} u_j^i u_j^{i\top}$ by the scalar $1/N^3$ has complexity $O(p^2)$. Hence the computing $\pi_1(T)$ and $\pi_2(T)$ has complexity $\mathcal{O}(N^2 p^2)$. Computing the product $u_j^i y_j^i$ has a complexity of $O(p)$ by Lemma 3-(iii), and summing $\sum_{i=1}^{N} \sum_{j=1}^{|\mathcal{D}_i|} u_j^i y_j^i$ has complexity $O(N^2 p)$ by Lemma 3-(i). Multiplying the $p \times 1$ vector $\sum_{i=1}^{N} \sum_{j=1}^{|\mathcal{D}_i|} u_j^i y_j^i$ by the scalar $1/N^2$ has a complexity of $O(p)$. Therefore computing $\pi_3(T)$ has complexity $O(N^2 p)$, and the complexity of computing $\pi_1(T)$, $\pi_2(T)$, and $\pi_3(T)$ is $\mathcal{O}(N^2 p)$.

**Complexity of Computing $\gamma_1(T)$ and $\gamma_2(T)$ is $O(p^{2.373})$**
With $\pi_1(T)$ and $\pi_2(T)$ obtained, we can compute $\gamma_1(T)$ and $\gamma_2(T)$ by (18) and (19). From (18) and (19), computing $\gamma_1(\cdot)$ and $\gamma_2(\cdot)$ from $\pi_1(\cdot)$ and $\pi_2(\cdot)$ involves multiplication and addition of $p \times p$ matrices, scalar multiplication of $p \times p$-dimensional matrices, multiplication of a $p \times p$ matrix and a $p \times 1$ vector, and inversion of a $p \times p$ matrix. By Lemma 3, multiplication of two $p \times p$ matrices has the dominant complexity of $\mathcal{O}(p^{2.373})$ among the above matrix operations. Moreover, inverting a $p \times p$ matrix has complexity $\mathcal{O}(p^{2.373})$. Hence, computing $\gamma_1(T)$ and $\gamma_2(T)$ has a complexity of $O(p^{2.373})$.

**Complexity of computing $\pi_1(t)$, $\pi_2(t)$, $\pi_3(t)$, $\gamma_1(t)$ and $\gamma_2(t)$ for $t = T-1$, $T-2$, ..., 1 has complexity $\mathcal{O}(Tp^{2.373})$.**
Similar to computing $\gamma_1(T)$ and $\gamma_2(T)$, computing $\gamma_1(t)$ and $\gamma_2(t)$ from $\pi_1(t+1)$ and $\pi_2(t+1)$ has complexity $\mathcal{O}(p^{2.373})$. According to (96), computing $\pi_1(t)$ and $\pi_2(t)$ from $\gamma_1(t+1)$ and $\gamma_2(t+1)$ involves scalar multiplication and addition along diagonal of $p \times p$ matrices, and therefore has complexity $O(p^2)$ by Lemma 3-(i) and (ii), According to (100), computing $\pi_3(t)$ from $\gamma_1(t+1)$, $\gamma_2(t+1)$, and $\pi_3(t+1)$ involves scalar multiplication and addition of $p \times p$ matrices, scalar multiplication and addition of $p \times 1$ matrices, and multiplication of a $p \times p$ matrix with a $p \times 1$ matrix. By Lemma 3, the dominant complexity of these operations is $\mathcal{O}(p^2)$. Summing up the above argument, computing $\pi_1(t)$, $\pi_2(t)$, $\pi_3(t)$, $\gamma_1(t)$ and $\gamma_2(t)$ for each $t$ has a complexity of $\mathcal{O}(p^{2.373})$, and therefore has a complexity of $\mathcal{O}(Tp^{2.373})$ for all $t = T-1$, $T-2$,..., 1.

**The Cost of Computing Each $\Theta(t)$ is $O(Tp^2 \max\{N, p^{0.373}\})$.**
By (9), the cost of updating from $\Theta(t)$ to $\Theta(t+1)$ is equal to the complexity of computing $\widehat{\alpha}(t) = (\widehat{\alpha}_1^\top(t), \cdots \widehat{\alpha}_N^\top(t))^\top$ for $t = 0, 1, ..., T-1$.

By (20), $\widehat{\alpha}_i(t)$ for all $i = 1, ..., N$ share the common term

$$- \big\{\gamma_1(t+1)\pi_2(t+1) + \gamma_2(t+1)[\lambda I_p + \pi_1(t+1) + (N-2)\pi_2(t+1)]\big\} \sum_{k=1}^{N} \theta_k(t)$$

$$- \big[\gamma_1(t+1) + (N-1)\gamma_2(t+1)\big](\pi_3(t+1) - \lambda\theta_{(N)}^\star),$$

and computing the common term involves matrix multiplication, addition, and scalar multiplication of $p \times p$ matrices, matrix addition and scalar multiplication of $p \times 1$ matrices, and multiplication of a $p \times p$ matrix and a $p \times 1$ matrix. By Lemma 3, these operations have complexity $O(Np + p^{2.373})$, where $Np$ is due to adding the $N$ $p$-dimensional vectors $\sum_{k=1}^{N} \theta_k(t)$, and $p^{2.373}$ is due to multiplication of two $p \times p$ matrices by a CW-like algorithm.

Computing the individual terms

$$-(\gamma_1(t+1) - \gamma_2(t+1))(\lambda I_p + \pi_1(t+1) - \pi_2(t+1))\theta_i(t)$$

for $\widehat{\alpha}_i(t)$, $i = 1, ..., N$ involves addition, scalar multiplication, and matrix multiplication of $p \times p$ matrices, and multiplication of a $p \times p$ matrix and a $p \times 1$ matrix. By Lemma 3, these operations have complexity of $\mathcal{O}(p^{2.373} + Np)$, where $p^{2.373}$ is due to multiplication of two $p \times p$ matrices by a CW-like algorithm and $Np$ is due to multiplying a $p \times p$ matrix by the $p$-dimensional vector $\theta_i(t)$ for $i = 1, ..., N$. Hence the complexity of computing $\widehat{\boldsymbol{\alpha}}(t)$ for $t = 0, 1, ..., T-1$ is $\mathcal{O}(Tp^2 \max\{N, p^{0.373}\})$.

**The complexity of computing** $(\Theta(t))_{t=0}^{T}$ **is** $\mathcal{O}(Tp^2 \max\{N, p^{0.373}\})$
Looking over all computations involved, we deduce that the complexity of computing the sequence $(\Theta(t))_{t=1}^{T}$ starting from $\Theta(0) = \Theta^{\star}$ has a complexity of $\mathcal{O}(Tp^2 \max\{N, p^{0.373}\})$. $\qquad\square$

We may now deduce Theorem 6 and Theorem 7.

*Proof of Theorem 6.* The result directly follows from Theorem 3 and Corollary 3. $\qquad\square$

*Proof of Theorem 7.* Direct consequence of Propositions 4 and 5. $\qquad\square$

## C Background

In this paper, we consider regret-optimal memoryless training dynamics for **f**inite-rank **k**ernel **r**idge **r**egressors (fKRR) trained from $N$ finite datasets drawn from $N$ (possibly) different data-generating distributions. We are considering a scenario in which these datasets can originate from $N$ distinct tasks or be adversarially generated perturbations of the initial dataset, and we aim for our fKRR to exhibit robustness towards them.

Any continuous *feature map* $\phi : \mathbb{R}^d \to \mathbb{R}^p$, induces a hypothesis of fKRR $f_\theta : \mathbb{R}^d \to \mathbb{R}$ each of which is uniquely determined by a *trainable parameter* $\theta \in \mathbb{R}^p$ via

$$f_\theta(x) = \phi(x)^\top \theta; \tag{103}$$

where all vectors are represented as column-vectors and not row vectors.

The hypothesis class $\mathcal{H}$ possesses a natural reproducing kernel Hilbert space (RKHS) structure with inner-product given for $f_\theta, f_{\tilde{\theta}} \in \mathcal{H}$ by $\langle f_\theta, f_{\tilde{\theta}} \rangle_\phi \overset{\text{def.}}{=} \sum_{k=1}^{p} \theta_k \tilde{\theta}_k$. This RKHS structure suggests an optimal selection criterion to decide which hypothesis in $\mathcal{H}$ best describes a *single* training set $\mathcal{D}_1 \overset{\text{def.}}{=} \{(x_j^1, y_j^1)\}_{j=1}^{S^1}$ of $S^1 \in \mathbb{N}_+$, by solving the penalized least-squares problem

$$\theta_1^{\star} \in \underset{\theta \in \mathbb{R}^P}{\arg\min} \left\| \Phi(x^1)\theta - y^1 \right\|_2^2 + \kappa \|\theta\|_2^2, \tag{104}$$

where

$$\Phi(x^1) \overset{\text{def.}}{=} \left( \phi(x_1^1), \ \phi(x_2^1), \ \cdots, \ \phi(x_{S^1}^1) \right)^\top \in \mathbb{R}^{S^1 \times p}, \ y^1 \overset{\text{def.}}{=} (y_1^1, y_2^1, \cdots, y_{S^1}^1)^\top \in \mathbb{R}^{S^1},$$

and $\kappa > 0$ is a fixed hyperparameter. In this classical case, the Representer Theorem Kimeldorf & Wahba (1970; 1971) states that the optimal choice of $f_\theta$ to the convex optimization problem (104) is parameterized by

$$\theta_1^{\star} = (\Phi^\top(x^1)\Phi(x^1) + \kappa I_p)^{-1}\Phi^\top(x^1)y^1, \tag{105}$$

Alternatively, the optimal parameter choice $\theta_1^\star$ can be arrived at via *gradient descent* since (104) has a strictly convex loss with a Lipschitz gradient. The training of fKRR on a single dataset is very-well understood. Likewise, other aspects of fKRR are well understood; namely, their statistical properties Cheng et al. (2023), their in-sample behaviour in Amini et al. (2022), and even the approximation power and risk of randomly generated fKRR are respectively known Gonon et al. (2023) and Gonon et al. (2020).

Using the fKRR (103), the joint terminal time objective (3) simplifies to

$$
\begin{aligned}
l(\theta_1, \ldots, \theta_N, w; \mathcal{D}) &\stackrel{\text{def.}}{=} \sum_{i=1}^{N} w_i \left( \sum_{j=1}^{S_i} \left| \langle u_j^i, \theta^w \rangle - y_j^i \right|^2 \right) \\
\text{s.t. } \theta^w &\stackrel{\text{def.}}{=} \sum_{i=1}^{N} \theta_i w_i,
\end{aligned}
\tag{106}
$$

where $\langle \cdot, \cdot \rangle$ denotes the Euclidean inner-product, $u_j^i \stackrel{\text{def.}}{=} \phi(x_j^i)$ and, as in (3), $w = (w_1, \cdots, w_N) \in [0,1]^N$ with $\sum_{i=1}^{N} w_i = 1$.

