# OpenReview forum: "Regret-Optimal Federated Transfer Learning for Kernel Regression – with Applications in American Option Pricing"
_TMLR — Rejected by TMLR_

### Review · Reviewer_348V · 2023-11-04

**Summary Of Contributions:**

This paper studies federated transfer learning, and i) provides regret optimal algorithms and heuristics; ii) demonstrates the adversarial robustness of the proposed approach; and iii) applies the proposed approach to American option pricing problems.

**Audience:**

Yes

**Broader Impact Concerns:**

NA.

**Claims And Evidence:**

Yes

**Requested Changes:**

1. Typo at 3 lines after equation (3): 'one the tone hand' --> on the one hand.

**Strengths And Weaknesses:**

**Strength**

(S1) The proposed approach is carefully examined with supportive theoretical derivations.

(S2) An efficient heuristic with near optimal regret is proposed. The efficiency improvement is critical for problem in higher dimensional space.

(S3) The adversarial robustness of the proposed approach is established. This is a useful result in the scenario of federated learning, because it

** Weakness**

(W1) This paper can benefit from demonstrating more on the definition of regret. For example, it is not clear on the reason to consider stability.

---

### Review · Reviewer_nq6v · 2023-12-02

**Summary Of Contributions:**

The paper studies the federated transfer learning that aims to learn a model from multiple datasets. When the model can be represented as a finite-rank kernel regression, the authors provide a regret-optimal algorithm, and investigate the robustness under adversarial data points.  Finally, the authors provide experiments on American option pricing.

**Audience:**

Yes

**Claims And Evidence:**

No

**Requested Changes:**

- This paper has to be re-written in a readable way. Motivation, Problem formulation, Contribution, and State of the Art should be organized in a clean way. Algorithm design idea and analysis novelty should be highlighted.

- The comparison with existing methods should be explained in both theory and experiments.

**Strengths And Weaknesses:**

Strengths:

- The optimization with many datasets is an important problem in federated learning.

- The authors present algorithms that perform information-sharing and optimization, which are crucial for efficiency. The authors also present an accelerated optimization algorithm.

- The authors demonstrate the performance of algorithms in American option pricing.

Weaknesses:

-  The paper is poorly organized. For instance, the introduction mixes motivation, background, problem formulation.

- The writing of this paper has a technical style that may fit the learning theory community, which is not generally readable to a broad community of machine learning.

- The regret optimal algorithm is not formally introduced before 1.1 Contributions. The motivation of developing such algorithms is not clearly emphasized.

- The novelty of proposed algorithms is limited due to the similarity of algorithm ideas with the prior art.

- The optimal regret is not defined in this federated learning setup.

- The learning model is limited to linear kernels.

- Colors in Figures are not distinguishable. There is no comparison with other methods.

---

### Review · Reviewer_u2oB · 2024-01-09

**Summary Of Contributions:**

The paper studies federated transfer learning, a setting in which data sets $D_i$ are distributed according to the row partition model across nodes, and a central planner has access to the same learning model $f_\theta$. The communication model assumed allows for communication between the nodes and the server. The goal is to minimize the error in the learned parameter $\theta$ while maintaining stability of the parameter between successive iterations. This problem is formulated as a regret minimization problem with regularization. The model $f_\theta$ is a finite-rank kernel ridge regressor, which means linear in $\theta$ but nonlinear in inputs via the feature map $\phi$.

By interpreting the regret minimization problem as one of optimal control the paper then applies existing optimal control tools to derive an algorithm for the problem. Additionally, the paper characterizes the adversarial robustness of their developed algorithm and the runtime cost for attaining regret optimality based on how dissimilar the $D_i$ are across nodes. The paper supplements its theory with experiments in the optional stopping problem for American options pricing.

**Audience:**

Yes

**Claims And Evidence:**

Yes

**Requested Changes:**

I am curious to see how the adversarial robustness results compare with those in the textbook by Diakonikolas and Kane.

**Strengths And Weaknesses:**

The paper's algorithm is quite reminiscent of mirror descent from the standard optimization literature. The paper's main contributions are to the problem of communication complexity. Overall the writing style of the paper is quite clear.

---

> ### Author Response · Authors · 2024-01-16
> **Differences Between Mirror Descent and the Regret-Optimal Algorithm**
>
> Dear Reviewer u2oB,
>
> Thank you very much for your positive feedback on our manuscript.
>
>  The paper's algorithm is quite reminiscent of mirror descent from the standard optimization literature.
> The main link between our regret-optimal algorithm (Algorithm 1) and mirror descent is that our algorithm initializes (Algorithm 2) the weights $w$ in the objective function by minimizing a loss function penalized by the KL divergence.  However, the updates in the regret-optimal algorithm, for fixed $w$, do not share any other explicit connection to mirror descent.
>
> For instance, the regret-optimal algorithm produces iterates, $\theta_n$, which depend on all previous iterates $(\theta_m){m\le n}$.  In this way, Algorithm, has long/full memory.  In contrast, mirror descent produces the updated parameter $\theta_n$ only using $\theta_{n-1}$, and in this sense the algorithm is memoryless.  In this way the regret-optimal algorithm leverages the entire path information when making an update but mirror descent (and similar "memoryless algorithms" do not).
>
> one step for memory from the past while Algorithm 1 uses the entire realized trajectory to product an update at any iteration.  We have added this as a remark following our adversarial robustness result, Theorem 4.
>
> Though we were previously not aware of it, both Algorithm 1 and mirror descent exhibit different notions of optimality.  For instance, Theorem 14 in [Lower Bounds for Parallel and Randomized Convex Optimization - Diakonikolas and Guzman] implies that, in certain settings, Mirror descent achieves the minimal number of communication rounds required to optimize the loss function $\ell$ to a given error.  While, in comparison, our main result (Theorem 1) shows that the regret-optima l algorithm (Algorithm 1) optimizes the (path-wise) regret functional $\mathcal{R}$ and not the terminal time objective essentially encoded by $\ell$.  Nevertheless, it is very much worth highlighting this similarity and we have included it in the related works section.
>
> I am curious to see how the adversarial robustness results compare with those in the textbook by Diakonikolas and Kane.
>
> Though the notion of adversarial robustness is only mildly different, due to the Gaussianity assumption, in Theorem 7.15 in [Algorithmic High-Dimensional Robust Statistics, 2023, I. Diakonikolas and D. Kane] it is indeed true that both algorithms achieve nearly the same complexity up to a polylogarithmic factor (though for mildly different notions of ε-poisoned data).  We have added this as a remark following our adversarial robustness result in Theorem 4.

---

### Decision · Action_Editor_pW3U · 2024-04-14

**Recommendation:** Reject

**Comment:**

Before I begin these lengthy decision comments, I would like to encourage the authors to please consider again the review of Reviewer nq6v. Not to rule out the other critiques, but in particular I tend to agree that the paper has organizational issues, issues in presenting the algorithms, and issues with motivation (I discuss this more below). I actually tend to disagree that the paper is written in a more learning theory style. I find that the focus on learning is missing right now (I also comment on this below), with paper instead emphasizing optimization and control.

# Main comments

This paper positions itself to be about transfer learning. The authors take as their objective equation (5), the penalized regret functional, and the paper revolves around showing how to efficiently find the optimal solution to this objective and beneficial properties of the optimal solution (like adversarial robustness).

To my eyes, the paper suffers from at least 4 major issues: lack of motivation for the main objective, lack of motivation and explanation of plausibility of assumptions, presentational issues (already highlighted by one reviewer), and some issues with the technical presentation. I will cover these in turn. My intent is to give the authors constructive criticism so that they can complete a major revision of their work, making it much stronger, after which I suggest the authors either submit a smaller version of their revised work (focusing on fewer aspects) to a conference, or submit the full work to a journal.

## Main objective
It is not clear to me why the objective (5) was chosen. The authors do mention that one term is related to transfer learning, and another term is related to algorithmic stability. Also, there is a game-theoretic interpretation in the appendix. However, ultimately, what is needed is a crisp/concrete theoretical motivation for choosing such an objective. For more context, I would like to point out:
 - Transfer learning is an area for which there are many theoretically principled works: the authors have cited some of these works, like Baxter's 2000 JAIR paper. Other works, like the Argyriou et al. (2006) paper, have had theory established (see [1]). By theory, here I mean actually saying something about generalization, i.e., how well the trained method will perform on new samples drawn from the same distribution as that of the principal regression task's data $\mathcal{D}_1$.
 - There is a rich set of results relating algorithmic stability (both at the single task level and at the meta level) to generalization. Also, in the design of online learning algorithms (which I admit is different from the authors' work, for which the focus is not regret in the online learning/"measuring prediction on new data" sense), it is beneficial for algorithms to satisfy some notion of stability in order to prove regret bounds. However, the regret itself ultimately does not have any explicit algorithmic stability term; rather, algorithmic stability may appear "under the hood" (in an analysis of Follow the Regularized Leader or Online Mirror Descent) when proving regret bounds.
 - In the authors' setting, they have all the data ahead of time. That is, the data does not arrive online. Consequently, I am not able to understand why the authors formulate a notion of regret which takes into account all of the optimization iterates. Taking the analogy of stochastic convex optimization, we may well use online learning algorithms (which, in that case, actually do often proceed by treating the data as arriving online), but ultimately we gauge the performance of the algorithm on a single output of the algorithm: this may be either an online-to-batch conversion (like some type of average of the optimization iterates), or the last iterate. To stress my point more clearly, I am saying let us not confuse (5), which is a criterion for an algorithm to minimize and may well need to depend on all the iterates, from the actual goal which is to ultimately obtain a single predictor that performs well on *new* data drawn from the principal regression task.

[1] Maurer, A. Transfer bounds for linear feature learning. Machine Learning, 75(3):327–350, 2009.

## Assumptions
While I believe Assumption 1 is fair and may not need much additional explanation, Assumption 2 definitely warrants some discussion. I don't have good intuition for why either of equations (7) or (8) are natural to impose. In addition to motivating Assumption 2, it's really important to discuss the plausibility of this assumption. A good way to to do that is by providing examples.

## Organization
The organization of the paper needs to be better. I will just mention some of the issues I ran into, but I suggest the authors have a (non-coauthor) colleague look over the paper to provide more detailed advice.

Several times, the authors reference equations that occur much later in the text (in fact, sometimes these equations are in the appendix). For the reader, such far-forward references are unnatural, as the reader may feel that they were supposed to already be familiar with the equation (or that it will be presented very soon). To give a few examples, Figure 1 (a) references equation (25), the same thing happens near the bottom of page 6, and Theorem 2 references equations (28) and (29). The authors should do a better job guiding the reader. They might consider moving some of these equations earlier, or if those equations absolutely should appear much later and the authors still wish to reference them much earlier (this should be done sparingly), explicitly tell the reader what section the equations appear in so the reader is not confused.

Regarding Algorithm 1, in the text, the authors write:
> "We begin by describing our procedure for determining the optimal mixture weights...".

Where is this description?  The description only seems to be in the captions in Figure 1 (which are somewhat vague). Writing the description in the main text (not only in a caption), with clear reference to the mathematical updates in the algorithm, would alleviate a lot of confusion.

If the game-theoretic interpretation of the algorithm is important, please consider putting it in the main text (another reason to do this is to better motivate your objective (5)  in the main text). Ideally, this interpretation might even lead to some theoretical backing for the choice of your optimization objective (5).

I don't know if this is an organizational issue exactly, but somehow it was not clear how we should choose $T$. My understanding is that we are given various datasets, but $T$ is a free parameter. How should it be tuned (I assume larger is better, but how large is good enough; is there some theory here)?

## Technical presentation

Finally, I mention some issues with the technical presentation.

In Algorithm 1, it is unclear why the weights $w_i$ are computed; they are never used nor returned. Relatedly, I don't see why there is a Step 1 in Figure 1.

Also in Algorithm 1, the scores $s_i$ are never defined in the algorithm. At first, I entertained the possibility that the scores $s_i$ depend on the weights $w_i$, but upon seeing equation (11) and especially equation (13), I now believe that $s_i$ (respectively $s_j$) is a typo that should have been written as $s^i$ (respectively $s^j$).

The most major issue is the statement of Theorem 4. The authors claim that
>"Here $O$ hides a constant depending only on Assumption 1."

However, Assumption 1 only involves $K_x$ and $K_y$, but from Corollary 2, the hidden constant also depends on $\lambda, \beta, N, \bar{N}, \Theta^\star, (w_1^\star, \ldots, w_N^\star )$, and $T$. Also, you are also missing a $\sqrt{p}$ factor. I imagine it is very important to discuss the precise dependence on these various problem/dataset-dependent parameters in order to interpret Theorem 4.

In light of the the various issues, I did not feel it was warranted to also closely look at the part of the paper related to acceleration, nor to closely look at the experimental results.


# Minor
Just after Remark 1, the authors write:
> "Since we will be assuming that we work with floating-point arithmetic, all elementary arithmetic operations (e.g. addition, subtraction, multiplication) are of constant (i.e. O(1)) complexity."

You can easily dispense with this level of detail. This type of thing is taught within first or second year undergrad computer science (the RAM model!), and is generally assumed in any paper.

The notation $\\#$ was never defined. I assume it means the cardinality of a set that follows the operator. Please consider using different notation, as readability becomes different when you write something like $\tilde{C} \\# I p 2 \varepsilon^2$.

# Final comment

I deeply regret how long the review process took for this paper. One thing I will say, based on experience, is that organizational issues, as well as a paper that touches on many different areas, does have a tendency to slow down the submission of reviews (as well as to sometimes lead to lower quality reviews). I already think the paper is better (more focused) by having removed the federated learning aspect (as the authors did in going from their original submission to the current version). I felt that the adversarial robustness angle might have been more of a distraction and would suggest instead pushing more on the motivation of your regret objective from the generalization perspective, especially if (e.g.) targeting a machine learning journal.

**Audience:**

Yes, there is an audience for this paper. I believe the audience would be the transfer learning community. With the current version of the paper, I do not think it will interest the federated learning community.

**Claims And Evidence:**

There appear to be some issues with at least one claim. This is discussed in in the detailed comments that appear in the "Comment" box, in the section titled "Technical Presentation".

---

> ### Author Response · Authors · 2024-04-27
>
> Dear AE,
>
> Thank you for your feedback.  We will incorporate some of these suggestions in a future version, such as the super/subscript typos and the comment that O also hides a constant depending on p; which slipped past us.  Thanks for noticing these typos.
>
> That said, we are a bit unsure about why one should add guarantees on the generalization of the model selected by our algorithm when the paper studies the efficiency of the algorithm itself, in the context of meta-optimization.   Nevertheless, we will keep these ideas in mind.
>
> Best,
> The authors.